# Notes on symmetries in particle physics

**Akash Jain**

Department of Physics & Astronomy, University of Victoria, PO Box 1700 STN CSC, Victoria, BC, V8W 2Y2, Canada

ABSTRACT: These are introductory notes on symmetries in quantum field theory and how they apply to particle physics. The notes cover the fundamentals of group theory, their representations, Lie groups, and Lie algebras, along with an elaborate discussion of the representations of SU(N), Lorentz, and Poincaré groups and their respective algebras. We spend a lot of time on the realisation of these symmetry groups in quantum field theory, as both global and gauge symmetries, as well as their spontaneous breaking and the Higgs mechanism. In the end, we culminate all the lessons from the course to enumerate the symmetries and field content of the Standard Model of particle physics and write down the Standard Model Lagrangian. Special consideration is given to how the weak-force gauge bosons and the matter fields obtain their mass via the Higgs mechanism.

These lecture notes were developed while teaching the graduate course PHYS 509 - Standard Model Phenomenology at the University of Victoria in the spring of 2021. They build upon and extend the hand-written notes of Prof. Adam Ritz who had developed and delivered the course in previous years. That prior work is acknowledged, and included here with permission.

DATE: September 27, 2021

COMMENTS: $86 + 1$ pages, 1 figure, 1 table, LaTeX

---

E-mail: ajainphysics@gmail.com

# 1 | Symmetries in particle physics

Symmetry is one of the foundational pillars on which our modern understanding of physics is built. Symmetries underlying physical systems put strong constraints on the mathematical models we build to describe them, often boiling the freedom down to a handful of free parameters that can be fixed by experiments and observations. By virtue of Noether's theorem [1, 2], symmetries have also been associated with the conserved quantities that remain constant throughout the evolution of a system, such as energy, momentum, angular momentum, charge, and particle number. These play a vital role in characterising the spectrum of physical states a system can exhibit, as well as its time-evolution. On the other hand, in high-energy physics, symmetries have been known to underlie the fundamental forces we see in nature, i.e. weak force, strong force, electromagnetic force, and the gravitational force (albeit a complete description of gravity is still lacking due to our inability to reconcile it with the quantum nature of the universe). Our present understanding of the former three of these fundamental forces is compiled into a unified field-theoretic description, developed during the second-half of the 20th century, known as the *Standard Model of particle physics* [3]. The discovery of a part of the model, known as the *electroweak theory*, furnishing a unified description of weak and electromagnetic forces, was awarded the 1979 Nobel Prize in physics [4].

Even the absence of symmetries has proved to be a powerful organisational tool in physics. The physical phases of matter, including the traditional ones such as solids, liquids, and gases, as well as the more exotic ones such as superfluids, superconductors, and the Bose-Einstein condensate, can often be classified based on symmetries alone (or the absence thereof). Careful consideration of symmetries has also led to the prediction and discovery of entirely new phases of matter previously unknown to physicists, and have even been the subject of the 2016 Nobel Prize in physics [5]. In particle physics, the (spontaneous) breaking of the electroweak symmetry at low-energies is crucial to give mass to the weak-force carriers W and Z bosons, making the weak-force short-ranged. This so-called Higgs mechanism is also responsible for the masses of (almost) all the quarks and leptons making up the entire observable matter in the universe. This discovery led to the 2013 Nobel Prize in physics [6]. The only exception to this general rule are neutrinos, which are left massless by the Higgs mechanism, but have been observed to be massive in nature. This discovery was awarded yet another Nobel Prize in 2015 [7]. Therefore, it is no surprise that a deluge of pre-prints continues to appear every day in theoretical physics, suggesting ingenious ways of exploiting symmetries to learn more about the world around us.

The aim of this course is to set the groundwork for the discussion of symmetries in (quantum) field theory, particularly aimed at the applications in particle physics. As we expect from a theory of fundamental forces in nature, the Standard Model of particle physics is invariant under spacetime symmetries – arbitrary space and time translations, rotations, and the special relativistic Lorentz boosts – together known as the Poincaré transformations. The model also features certain abstract "internal" $SU(3) \times SU(2) \times U(1)$ "gauge" symmetry comprised of $8 + 3 + 1$ independent symmetry transformations; we will learn more about this nomenclature and notation as we navigate through the course. Superficially speaking, the gauge symmetries reflect redundancies in the description of the theory that arise in order to manifestly preserve locality and Lorentz invariance. The 8 transformations making up the $SU(3)$ part of the internal symmetry correspond to the strong force, while the remaining 4 transformations making up the $SU(2) \times U(1)$ part correspond to the electroweak force. Every particle or field occurring in nature furnishes a representation of these symmetries, i.e. transforms in a well-defined manner under these symmetry transformations, in a

way that leaves the physics invariant. At our day-to-day energy scales, the $SU(2) \times U(1)$ symmetry is spontaneously broken down to a single $U(1)$ sub-transformation, breaking down the electroweak force into a short-ranged weak force and a long-ranged electromagnetic force. The Standard Model is not invariant under the spatial-parity (P) and/or time-reversal (T) operation, which are symmetries of the underlying Minkowski spacetime, but is invariant under the combined CPT symmetry including the charge conjugation (C) operation.

In these notes, we will not follow the historical flow of developments leading up to the Standard Model; several excellent references already follow this approach; see e.g. [8]. Rather, we will take a symmetry-oriented perspective where we will use symmetries, their representations, and their breaking pattern to build the Standard Model Lagrangian from the ground-up. The mathematical language for symmetry is *group theory*. The notes start with a formal discussion of the basics of group theory in section 2, specialising to the Abelian unitary group $U(1)$ in section 3, the special unitary group $SU(N)$ in section 4 and the Lorentz and Poincaré groups in section 5, relevant for the Standard Model. Building upon these formal developments, in section 6, we introduce how these symmetry groups can be realised in (quantum) field theory, and eventually write down the Standard Model Lagrangian in section 7. These latter two sections also cover the spontaneous breaking of internal symmetries and the Higgs mechanism, which generates masses for the matter fields and the weak-force gauge bosons. Finally, in section 9, we close off with a outlook and further scope of the course material.

Although we are not aware of any texts that closely follow the structure of this course, there are a plethora of excellent references that individually cover various of its aspects, albeit in much more detail than the scope of these notes. A list of the suggested reading material appears on Prof. Adam Ritz's course website. We will mention the suggested reading material for individual sections as we make our way through the notes.

Throughout these notes we will use the mostly positive metric sign convention $(-1, 1, 1, 1)$. We will *not* set any of the fundamental constants to one for pedagogical reasons.

# 2 | Basic group theory

In this section, we will study the fundamentals of group theory. We will set the stage with some illustrative examples of "rotation" and "reflection" symmetry groups to draw out the essential features of group theory. We will then proceed to a formal discussion of group theory and representation theory. A main feature of this section will be the discussion on Lie groups and Lie algebras that are vital from the perspective of field theory. The discussion in this section will be quite brief and focused to our needs; a more complete treatment can be found in the standard texts geared towards the application of group theory in theoretical physics such as [9–11].

## 2.1 Introductory examples

To facilitate the forthcoming abstract discussion of group theory, let us start with some illustrative examples of symmetries and groups. In particular, we shall be interested in the symmetries of a square and a circle, i.e. the geometric operations that leave the respective objects invariant.

### 2.1.1 Symmetries of a square – dihedral group

Let us start with a square. It has a 4-fold rotational symmetry, i.e. rotations by multiples of $\pi/2$ of the square about its center leave the square invariant:

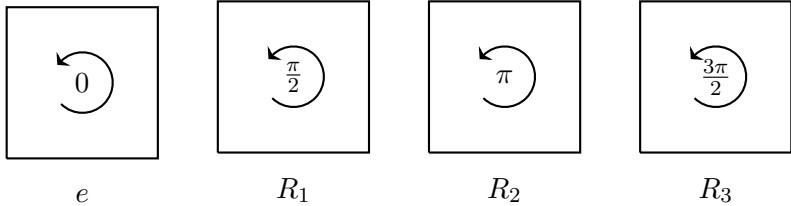

There are three non-trivial rotations by angles $\pi/2$, $\pi$, and $3\pi/2$, along with a trivial rotation by angle 0 or $2\pi$ that amounts to doing nothing. In addition, the square is also invariant under reflections (parity transformations) about the horizontal and vertical axes, and about the two diagonals:

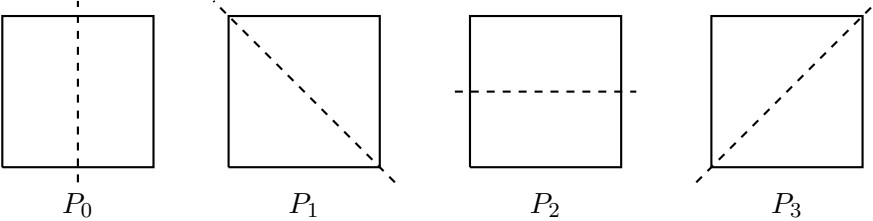

These 8 transformations together form the *dihedral group* $D_4 = \{e, R_1, R_2, R_3, P_0, P_1, P_2, P_3\}$. Note that composing any two symmetry transformations together results in another element of the group.

This can be summarised via the multiplication table

| | $e$ | $R_1$ | $R_2$ | $R_3$ | $P_0$ | $P_1$ | $P_2$ | $P_3$ |
|---|---|---|---|---|---|---|---|---|
| $e$ | $e$ | $R_1$ | $R_2$ | $R_3$ | $P_0$ | $P_1$ | $P_2$ | $P_3$ |
| $R_1$ | $R_1$ | $R_2$ | $R_3$ | $e$ | $P_3$ | $P_0$ | $P_1$ | $P_2$ |
| $R_2$ | $R_2$ | $R_3$ | $e$ | $R_1$ | $P_2$ | $P_3$ | $P_0$ | $P_1$ |
| $R_3$ | $R_3$ | $e$ | $R_1$ | $R_2$ | $P_1$ | $P_2$ | $P_3$ | $P_0$ |
| $P_0$ | $P_0$ | $P_1$ | $P_2$ | $P_3$ | $e$ | $R_1$ | $R_2$ | $R_3$ |
| $P_1$ | $P_1$ | $P_2$ | $P_3$ | $P_0$ | $R_3$ | $e$ | $R_1$ | $R_2$ |
| $P_2$ | $P_2$ | $P_3$ | $P_0$ | $P_1$ | $R_2$ | $R_3$ | $e$ | $R_1$ |
| $P_3$ | $P_3$ | $P_0$ | $P_1$ | $P_2$ | $R_1$ | $R_2$ | $R_3$ | $e$ |

| | $g_1$ | $g_2$ | $\cdots$ |
|---|---|---|---|
| $g_1$ | $g_1 g_1$ | $g_2 g_1$ | $\cdots$ |
| $g_2$ | $g_1 g_2$ | $g_2 g_2$ | $\cdots$ |
| $\vdots$ | $\vdots$ | $\vdots$ | $\ddots$ |

$\implies$

The composition $g_1 g_2$ means the transformation $g_2$ followed by the transformation $g_1$. Note also that the composition does not, in general, commute. For example, $P_1 P_2 \neq P_2 P_1$. However, the composition is associative, i.e. $(g_1 g_2)g_3 = g_1(g_2 g_3)$ for all the elements of the group. Another property that the symmetry group has is that every transformation has an inverse, which can be used to undo the said transformation. To wit

$$e^{-1} = e, \qquad R_1^{-1} = R_3, \qquad R_2^{-1} = R_2, \qquad R_3^{-1} = R_1,$$
$$P_0^{-1} = P_0, \qquad P_1^{-1} = P_1, \qquad P_2^{-1} = P_2, \qquad P_3^{-1} = P_3.$$

The symmetry transformations in the dihedral group can be represented by $2 \times 2$ matrices representing their action on a point $(a, b)$ on the square. We find

$$e = \begin{pmatrix} 1 & 0 \\ 0 & 1 \end{pmatrix}, \qquad R_1 = \begin{pmatrix} 0 & 1 \\ -1 & 0 \end{pmatrix}, \qquad R_2 = \begin{pmatrix} -1 & 0 \\ 0 & -1 \end{pmatrix}, \qquad R_3 = \begin{pmatrix} 0 & -1 \\ 1 & 0 \end{pmatrix},$$
$$P_0 = \begin{pmatrix} -1 & 0 \\ 0 & 1 \end{pmatrix}, \qquad P_1 = \begin{pmatrix} 0 & -1 \\ -1 & 0 \end{pmatrix}, \qquad P_2 = \begin{pmatrix} 1 & 0 \\ 0 & -1 \end{pmatrix}, \qquad P_3 = \begin{pmatrix} 0 & 1 \\ 1 & 0 \end{pmatrix}.$$
$$(2.1)$$

This is called a *representation* of the group. It can be explicitly checked that this matrix representation follows the group multiplication table given above.

It should be noted that the entire group can be generated by multiplying together two fundamental elements: the $\pi/2$ rotation $R_1$ and the reflection about the vertical axis $P_0$. For instance, we have

$$e = (R_1)^4, \quad R_2 = (R_1)^2, \quad R_3 = (R_1)^3, \qquad P_1 = R_1 P_0, \quad P_2 = (R_1)^2 P_0, \quad P_3 = (R_1)^3 P_0. \quad (2.2)$$

The elements $R_1$, $P_0$ can be understood as the generators of the dihedral group.

There are various subsets, called *subgroups*, of the dihedral group $D_4$ that are closed under composition and form a group of their own. Few notable examples are the orientation preserving transformations of a square called the "cyclic group" $\mathbb{Z}_4 = \{e, R_1, R_2, R_3\}$, dihedral group of a line-segment (or of a rectangle) $D_2 = \{e, R_2, P_0, P_2\}$, the cyclic group $\mathbb{Z}_2 = \{e, R_2\}$, and the trivial subset $\{e\}$. The cyclic groups $\mathbb{Z}_4$ and $\mathbb{Z}_2$ can also be represented by the complex numbers

$$e = 1, \qquad R_1 = e^{i\pi/2}, \qquad R_2 = e^{i\pi}, \qquad R_3 = e^{i3\pi/2}. \qquad (2.3)$$

This furnishes a *complex representation* of the said groups and illustrates the idea that groups can have many distinct representations. We note yet another representation of $\mathbb{Z}_4$ and $\mathbb{Z}_2$ as

$$
e = \begin{pmatrix} 1 & 0 & 0 & 0 \\ 0 & 1 & 0 & 0 \\ 0 & 0 & 1 & 0 \\ 0 & 0 & 0 & 1 \end{pmatrix}, \quad
R_1 = \begin{pmatrix} 0 & 0 & 0 & 1 \\ 1 & 0 & 0 & 0 \\ 0 & 1 & 0 & 0 \\ 0 & 0 & 1 & 0 \end{pmatrix}, \quad
R_2 = \begin{pmatrix} 0 & 0 & 1 & 0 \\ 0 & 0 & 0 & 1 \\ 1 & 0 & 0 & 0 \\ 0 & 1 & 0 & 0 \end{pmatrix}, \quad
R_3 = \begin{pmatrix} 0 & 1 & 0 & 0 \\ 0 & 0 & 1 & 0 \\ 0 & 0 & 0 & 1 \\ 1 & 0 & 0 & 0 \end{pmatrix}.
\tag{2.4}
$$

A similar story can be repeated for other $n$-sided regular polygons, such a equilateral triangles, regular pentagons etc., leading to other dihedral groups $D_n$.

### 2.1.2 Symmetries of a circle – orthogonal group

A circle can be obtained as the $n \to \infty$ limit of a regular $n$-sided polygon. Correspondingly, the dihedral group $D_n$ in $n \to \infty$ limit leads to the symmetry group of a circle, known as the 2-dimensional *orthogonal group* O(2). The transformations in this group consist of arbitrary rotations about the center, and reflections about arbitrary diameters:

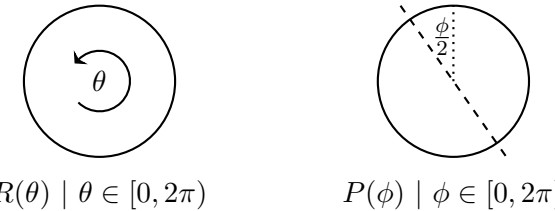

$$R(\theta) \mid \theta \in [0, 2\pi) \qquad\qquad P(\phi) \mid \phi \in [0, 2\pi)$$

The identity element is given by $e = R(0)$ that does nothing. The respective multiplication table can be written down as

| | $R(\theta_2)$ | $P(\phi_2)$ |
|---|---|---|
| $R(\theta_1)$ | $R(\theta_1 + \theta_2)$ | $P(\phi_2 - \theta_1)$ |
| $P(\phi_1)$ | $P(\phi_1 + \theta_2)$ | $R(\phi_1 - \phi_2)$ |

It is clearly seen that the group multiplication does not commute, because $R(\theta)P(\phi) \neq P(\phi)R(\theta)$, however it is associative. The inverse of the rotations $R(\theta)$ are $R(-\theta)$, while the inverse of the reflections $P(\phi)$ are themselves.

The orthogonal group O(2) can be represented by $2 \times 2$ matrices acting on an arbitrary point $(a, b)$ on the circle:

$$
R(\theta) = \begin{pmatrix} \cos\theta & \sin\theta \\ -\sin\theta & \cos\theta \end{pmatrix}, \qquad
P(\phi) = \begin{pmatrix} -\cos\phi & -\sin\phi \\ -\sin\phi & \cos\phi \end{pmatrix}.
\tag{2.5}
$$

This is the set of all $2 \times 2$ orthogonal matrices satisfying $MM^{\mathrm{T}} = \mathbb{1}$; hence the name. In can be checked that this *group representation* respects the multiplication table mentioned above.

The generators of O(2) can be taken to be the infinitesimal rotation $R_\epsilon = R(\epsilon)$ (for $\epsilon \to 0$) and the y-axis reflection $P_0 = P(0)$, leading to

$$
R(\theta) = \lim_{\epsilon \to 0} (R_\epsilon)^{\theta/\epsilon}, \qquad
P(\phi) = \lim_{\epsilon \to 0} (R_\epsilon)^{\phi/\epsilon} P_0.
\tag{2.6}
$$

For infinitesimal transformations, given a matrix representation, the generators can be represented as an exponential $R_\epsilon = \exp(i\epsilon\sigma_2)$, where $\sigma_2$ is the second Pauli matrix. The matrix $\sigma_2$ is called the Lie algebra generator of rotations. In terms of this, we can represent the group elements as $R(\theta) = \exp(i\theta\sigma_2)$ and $P(\phi) = \exp(i\phi\sigma_2)P_0$.

All dihedral groups $D_n$ and cyclic groups $\mathbb{Z}_n$ form a subgroup of the orthogonal group $O(2)$. These are given by

$$D_n = \{R_a = R(2a\pi/n), P_a = P(2a\pi/n) \mid a = 0, 1, \ldots, n-1\},$$
$$\mathbb{Z}_n = \{R_a = R(2a\pi/n) \mid a = 0, 1, \ldots, n-1\}. \tag{2.7}$$

Another interesting subgroup of $O(2)$ comprises of the orientation preserving transformations of a circle $SO(2) = \{R(\theta)\}$, known as the special orthogonal group of dimension 2. In the matrix representation above, $SO(2)$ is the set of all $2 \times 2$ orthogonal matrices with unit determinant $\det M = 1$. There is another representation of $SO(2)$ given by complex numbers

$$R(\theta) = e^{i\theta}, \tag{2.8}$$

formally known as the unitary group $U(1)$. More generally, the unitary group $U(N)$ is the set of all $N \times N$ unitary matrices satisfying $MM^\dagger = \mathbb{1}$.

Similarly, the symmetry group of an $N$-dimensional sphere is called $O(N)$, while its orientation preserving subgroup is called $SO(N)$. The former can be represented by $N \times N$ orthogonal matrices, while the latter by $N \times N$ orthogonal matrices with unit determinant.

## 2.2 Basic definitions

**Group:** A *group* $G$ is a set of elements with a product rule, such that

1. $G$ is *closed* under group multiplication, i.e. $g_1 g_2 \in G$ for all elements $g_1, g_2 \in G$ — combining two symmetry operations is also a symmetry.

2. The group product is *associative*, i.e. $(g_1 g_2)g_3 = g_1(g_2 g_3)$ for all elements $g_1, g_2, g_3 \in G$ — symmetry operations are associative.

3. There exists a unique identity element $e \in G$ such that $eg = ge = g$ for any element $g \in G$ — there exists a trivial symmetry operation where nothing is done.

4. For every element $g \in G$, there exists a unique inverse $g^{-1} \in G$ such that $g^{-1}g = gg^{-1} = e$ — symmetry operations can be inverted to return to the original state.

If $g_1 g_2 = g_2 g_1$ for all $g_1, g_2 \in G$, the group $G$ is called an *Abelian group* with a commutative product. If $g_1 g_2 \neq g_2 g_1$ for some $g_1, g_2 \in G$, the group $G$ is called a *non-Abelian group* with a non-commutative product.

- Example: from our illustrative examples in section 2.1, it can be explicitly checked that $D_4$, $\mathbb{Z}_4$, $O(2)$, and $SO(2)$ are all groups. In particular, $\mathbb{Z}_4$ and $SO(2)$ are Abelian groups, while $D_4$ and $O(2)$ are non-Abelian groups.

- Example: symmetry transformations in a field theory form a group. If a set of invertible field transformations $G = \{a, b, c, \ldots\}$ on the fields $\varphi \to a \circ \varphi$ leave the action invariant, i.e. $S[a \circ \varphi] = S[\varphi]$ etc.: (1) successive symmetry transformations also leave the action invariant, (2) symmetry transformations are associative, (3) the identity element is the trivial transformation $\varphi \to \varphi$, and (4) symmetry transformations have well-defined inverses.

**Group isomorphism:** Two groups $G$ and $G'$ are said to be isomorphic, denoted as $G \cong G'$, if there exists an invertible map $\iota : G \to G'$ such that $\iota(e) = e'$, $\iota(g_1 g_2) = \iota(g_1)\iota(g_2)$, and $\iota(g^{-1}) = \iota(g)^{-1}$, where $g, g_1, g_2 \in G$ are arbitrary elements and $e \in G$ and $e' \in G'$ are the identity elements.

- Example: from our illustrative examples in section 2.1, the groups U(1) and SO(2) are isomorphic, with the isomorphism

$$\iota\left(e^{i\theta} \in \mathrm{U}(1)\right) = \begin{pmatrix} \cos\theta & \sin\theta \\ -\sin\theta & \cos\theta \end{pmatrix} \in \mathrm{SO}(2). \tag{2.9}$$

**Product groups:** Given two groups $G$ and $G'$, their product group is given by the set of ordered pairs $G \times G' = \{(g, g') \mid g \in G, g' \in G'\}$ with the obvious product operation

$$(g_1, g_1')(g_2, g_2') = (g_1 g_2, g_1' g_2'). \tag{2.10}$$

- Example: from our illustrative examples in section 2.1, the dihedral group $\mathrm{D}_2$ is isomorphic to the product group $\mathbb{Z}_2 \times \mathbb{Z}_2$, with the isomorphism $\iota(e) = (e, e)$, $\iota(R_2) = (R_2, R_2)$, $\iota(P_0) = (e, R_2)$, and $\iota(P_2) = (R_2, e)$.
- NB: A subgroup $G$ is always isomorphic to its product with the trivial group $G \times \{e\}$. We will not distinguish between these groups.

**Subgroup:** A subset $H \subset G$ is called a *subgroup* of a group $G$ if it forms a group within itself with the same product rule.

A subgroup $H \subset G$ is called a *normal* or *invariant* subgroup of $G$ if $g^{-1}hg \in H$ for all $h \in H$ and $g \in G$.

- NB: given any group $G$, the trivial group element $\{e\}$ and the group $G$ itself are normal subgroups of $G$.
- NB: every subgroup of an Abelian group is normal.
- NB: given two groups $G$ and $G'$, the product group $G \times G'$ has normal subgroups $G \cong G \times \{e\}$ and $G' \cong \{e\} \times G'$.
- Example: $\mathbb{Z}_4$, $\mathrm{D}_2$, $\mathbb{Z}_2$ are subgroups of $\mathrm{D}_4$, while $\mathrm{D}_n$, $\mathbb{Z}_n$, SO(2) are subgroups of O(2). In particular, the subgroups $\mathbb{Z}_4$ and SO(2) are normal subgroups of $\mathrm{D}_4$ and O(2) respectively.

**Simple and semi-simple groups:** A group $G$ is said to be *simple* if its only normal subgroups are the trivial group $\{e\}$ and $G$ itself. A group is semi-simple if it is a product of simple groups.

- Example: the cyclic groups of prime order $\mathbb{Z}_p$ are simple.
- Example: the group SU(N) of all $N \times N$ unitary matrices with unit determinant is simple.

**Centre:** The *centre* $Z(G) = \{z \in G \mid zg = gz \; \forall g \in G\}$ of a group $G$ is the maximal subset of $G$ that commutes with all elements of $G$.

- NB: the centre of a group is a normal subgroup.

- NB: the centre of an Abelian group is the entire group itself.

- Example: the centre of all the groups $D_4$, $\mathbb{Z}_4$, $O(2)$, $SO(2)$ is the normal subgroup $\mathbb{Z}_2 = \{e, R_2\}$; note that $R_2 = R(\pi)$.

**Cosets:**   Let $G$ be a group and $H \subset G$ be a subgroup. Given an element $g \in G$, a left coset of $H$ in $G$ are given as $gH = \{gh \mid h \in H\}$. Similarly, a right coset of $H$ in $G$ are given as $Hg = \{hg \mid h \in H\}$.

- NB: given a normal subgroup $H \subset G$, the left and right cosets of $H$ in $G$ are the same, i.e. $gH = Hg$ for all $g \in G$, and are simply called cosets.

**Quotient group:**   Given a group $G$ and a normal subgroup $H \subset G$, the quotient group of $H$ in $G$ is defined to be the set of all cosets $G/H = \{gH \mid g \in G\}$. The product rule in the quotient group is given as $(g_1 H)(g_2 H) = (g_1 g_2)H$ for all $g_1, g_2 \in G$. Note that the identity element in $G/H$ is simply $H$ and the inverse elements are $(gH)^{-1} = g^{-1}H$.

- NB: given a product group $G \times G'$ and the normal subgroup $G \cong G \times \{e\}$, the quotient group $(G \times G')/G$ is isomorphic to $G'$. Similarly, $(G \times G')/G'$ is isomorphic to $G$. Note that the converse is not true: $H \times (G/H)$ is not generically isomorphic to $G$.

- Example: $\mathbb{Z}_4$ is a normal subgroup of $D_4$, and $D_4/\mathbb{Z}_4 \cong \mathbb{Z}_2$. However, $D_4 \ncong \mathbb{Z}_4 \times \mathbb{Z}_2$. Similarly, $SO(2)$ is a normal subgroup of $O(2)$, and $O(2)/SO(2) \cong \mathbb{Z}_2$. However, $O(2) \ncong SO(2) \times \mathbb{Z}_2$.

**Order:**   the *order* $|G|$ of a group $G$ is the number of elements in $G$.

- NB: continuous symmetry transformations, such as rotations of a circle, have order infinite and are known as infinite dimensional groups.

- Example: $|D_n| = 2n$, $|\mathbb{Z}_n| = n$, $|O(2)| = \infty$, and $|SO(2)| = \infty$.

- NB: Given groups $G$ and $G'$, we have $|G \times G'| = |G||G'|$. Similarly, given a group $G$ and normal subgroup $H \subset G$, we have $|G/H| = |G|/|H|$. Note that the latter is not well-defined when $H$ is an infinite dimensional group.

## 2.3   Group representations

An important aspect of group theory is that groups can have many *representations*. In addition to giving a tangible meaning to the abstract group elements in terms of matrices and such, in field theory, it also implies that the same symmetry can be realised by various different kinds of fields.

**Representation:**   An $N$-dimensional *matrix representation* $D(G)$ of a group $G$ is a mapping of the elements of $G$ onto a set of $N \times N$ invertible matrices $D : G \to GL(N)$, s.t.

1. $D(e) = \mathbb{1}$, where $\mathbb{1} \in GL(N)$ is the $N \times N$ identity matrix.
2. $D(g_1)D(g_2) = D(g_1 g_2)$ for all $g_1, g_2 \in G$.

A representation is said to be *faithful* if the mapping $D(G)$ is one-to-one and invertible. In particular, for a faithful representation, $D(g) = \mathbb{1} \implies g = e$. A representation $D(G)$ is said to be *unitary* if $D(g)$ is a unitary matrix for all $g \in G$, i.e. $D(g)D(g)^\dagger = \mathbb{1}$.

The representation $D(g) = \mathbb{1}$ for all $g \in G$ is called the *trivial* or *singlet* representation of the group $G$. Except for the order 1 trivial group $G = \{e\}$, the trivial representation is never a faithful representation. The trivial representation is a unitary representation.

For a complex matrix representation $D(G)$, the *conjugate representation* $D^*(G)$ is defined as $D^*(g) = D(g)^*$ for all $g \in G$.

- NB: a faithful representation of a group is isomorphic to the group.
- Given a representation $D(G)$ of a group $G$ and a subgroup $H \subset G$, we can construct a representation $D(H)$ of $H$ by restricting to the elements of $H$ in the map.
- Example: A 2-dimensional real matrix representation of $D_4$ and $\mathbb{Z}_4$ is given in eq. (2.1), while a 1-dimensional complex matrix representation of $\mathbb{Z}_4$ is given in eq. (2.3). Similarly, a 2-dimensional real matrix representation of $O(2)$ and $SO(2)$ is given in eq. (2.5), while a 1-dimensional complex matrix representation of $SO(2)$ is given in eq. (2.8). All of these are faithful representations. All of these are unitary representations.
- NB: in quantum field theory, we will particularly be interested in unitary representations. Given that a physical theory admits a symmetry group $G$, the physical states are expected to transform under a unitary representation of the group $|\psi\rangle \to D(G)|\psi\rangle$ for $g \in G$. This ensures that the inner product $\langle\psi|\psi\rangle$ remains invariant under the symmetry. Similarly, observable Hermitian operators also transform under a unitary representation of the group $\mathcal{O} \to D(G)\mathcal{O}D(G)^{-1}$, so that they remain Hermitian after the transformation.

**Equivalent representations:**  Two representations $D(G)$, $D'(G)$ of a group $G$ are said to be equivalent if there exists an invertible matrix $S$ such that $D'(g) = S^{-1}D(g)S$ for all $g \in G$. We will often not distinguish between equivalent representations of a group because they are related by a mere change of basis.

- NB: equivalent unitary representations $D(G)$, $D'(G)$ of a group $G$ are related via a unitary matrix $U$, i.e. $D'(g) = U^\dagger D(g)U$ for all $g \in G$.

**Direct sum representations:**  Given two representations $D(G)$, $D'(G)$ of a group $G$, we can construct a larger direct sum representation as

$$(D \oplus D')(g) = \begin{pmatrix} D(g) & 0 \\ 0 & D'(g) \end{pmatrix} \quad \forall g \in G \tag{2.11}$$

We can iterate this procedure for any number of representations.

**Irreducible representation:**  A representation $D(G)$ of a group $G$ is said to be *reducible* if it is equivalent to a block upper-triangular representation, i.e. there exists a matrix $S$ such that

$$S^{-1}D(g)S = \begin{pmatrix} D_1(g) & D_{12}(g) \\ 0 & D_2(g) \end{pmatrix} \quad \forall g \in G, \tag{2.12}$$

where the $\dim D_1(g) \neq \dim D(g)$. Note that, while $D_1(G)$, $D_2(G)$ are smaller sub-representations of $G$, the map $D_{12}(G)$ is *not* a representation. A representation is said to be *irreducible*, if it is not reducible.

A representation is said to be *completely reducible* if it is equivalent to a direct sum of irreducible representations.

- NB: an irreducible representation is automatically completely reducible.

- NB: we state without proof an important result that every finite-dimensional unitary representation of a group is completely reducible into a direct sum of irreducible unitary representations. This implies that to study unitary representations of a group, it is sufficient to classify all of its irreducible unitary representations.

- Example: eq. (2.1) furnishes an irreducible representation of both $D_4$ and $\mathbb{Z}_4$, and eq. (2.3) furnishes an irreducible representation of $\mathbb{Z}_4$. While eq. (2.3), being a one-dimensional representation, continues to furnish an irreducible representation of all the subgroups of $\mathbb{Z}_4$ as well, the representations of $D_2$ and $\mathbb{Z}_2$ furnished by eq. (2.1) consist of diagonal matrices and hence are reducible. A similar story applies to the representations of the continuous groups $O(2)$ and $SO(2)$ given in eqs. (2.5) and (2.8).

- The representation of $\mathbb{Z}_4$ given in eq. (2.4) is a reducible representation. Being a unitary representation, it is also a completely reducible representation. Using

$$
S = \begin{pmatrix} 1 & -1 & 1 & 0 \\ 1 & 1 & 0 & -1 \\ 1 & -1 & -1 & 0 \\ 1 & 1 & 0 & 1 \end{pmatrix}, \tag{2.13}
$$

as the equivalence transformation, this representation turns into

$$
e = \begin{pmatrix} 1 & 0 & 0 & 0 \\ 0 & 1 & 0 & 0 \\ 0 & 0 & 1 & 0 \\ 0 & 0 & 0 & 1 \end{pmatrix}, \qquad R_1 = \begin{pmatrix} 1 & 0 & 0 & 0 \\ 0 & -1 & 0 & 0 \\ 0 & 0 & 0 & 1 \\ 0 & 0 & -1 & 0 \end{pmatrix},
$$

$$
R_3 = \begin{pmatrix} 1 & 0 & 0 & 0 \\ 0 & 1 & 0 & 0 \\ 0 & 0 & -1 & 0 \\ 0 & 0 & 0 & -1 \end{pmatrix}, \qquad R_3 = \begin{pmatrix} 1 & 0 & 0 & 0 \\ 0 & -1 & 0 & 0 \\ 0 & 0 & 0 & -1 \\ 0 & 0 & 1 & 0 \end{pmatrix}. \tag{2.14}
$$

This is a direct sum of the trivial representation, another 1-dimensional non-faithful representation $D(e) = D(R_2) = 1$, $D(R_1) = D(R_3) = -1$, and the 2-dimensional faithful representation given in eq. (2.1).

## 2.4   Lie algebras

Let us take a quick detour to study a related topic – Lie algebras. These are quintessential in the study of continuous groups and play a pivotal role in high-energy physics. Their connection to group theory will be established later in section 2.5.

**Lie algebra:** A *Lie algebra* $\mathfrak{g}$ is a vector-space over some field $F$ (real $\mathbb{R}$ or complex numbers $\mathbb{C}$) with a bilinear Lie bracket operation $[\cdot, \cdot] : \mathfrak{g} \times \mathfrak{g} \to \mathfrak{g}$ satisfying

1. *Alternativity*: $[X, X] = 0$ for all $X \in \mathfrak{g}$.

2. *Anti-commutativity*: $[X, Y] = -[Y, X]$ for all $X, Y \in \mathfrak{g}$.

3. *Bilinearity*: $[aX + bY, Z] = a[X, Z] + b[Y, Z]$ for all $X, Y, Z \in \mathfrak{g}$ and $a, b \in F$.

4. *Jacobi identity*: $[X, [Y, Z]] + [Y, [Z, X]] + [Z, [X, Y]] = 0$.

The dimension of a Lie algebra $\dim \mathfrak{g}$ is defined to be the dimension of the vector space. A Lie algebra $\mathfrak{g}$ is said to be *Abelian* if $[X, Y] = 0$ for all $X, Y \in \mathfrak{g}$.

A Lie algebra is said to be Abelian if $[X, Y] = 0$ for all $X, Y \in \mathfrak{g}$. A Lie algebra is said to be non-Abelian if it is not Abelian.

- Example: examples of Lie algebras include the matrix algebras over $F = \mathbb{R}$:

  - $\mathfrak{gl}(N, \mathbb{R})$, $\mathfrak{gl}(N, \mathbb{C})$: real and complex $N \times N$ matrices (dimension $N^2$, $2N^2$).
  - $\mathfrak{sl}(N, \mathbb{R})$, $\mathfrak{sl}(N, \mathbb{C})$: real and complex $N \times N$ traceless matrices, $\operatorname{tr} M = 0$ (dimension $N^2 - 1$, $2N^2 - 2$).
  - $\mathfrak{so}(N)$: imaginary $N \times N$ antisymmetric matrices, $M^{\mathrm{T}} = -M$ (dimension $N(N-1)/2$).
  - $\mathfrak{u}(N)$: complex $N \times N$ Hermitian matrices, $M^\dagger = M$ (dimension $N^2$).
  - $\mathfrak{su}(N)$: complex $N \times N$ traceless Hermitian matrices, $M^\dagger = M$ and $\operatorname{tr} M = 0$ (dimension $N^2 - 1$).

  The respective Lie bracket operation is given by the matrix commutator $[X, Y] = XY - YX$. The Lie algebras $\mathfrak{gl}(N, \mathbb{C})$ and $\mathfrak{sl}(N, \mathbb{C})$ can also be defined over $F = \mathbb{C}$, but have dimensions $N^2$ and $N^2 - 1$ respectively.

**Generators and structure constants:** Given a Lie algebra $\mathfrak{g}$ over a field $F$, we can choose a vector-space basis $T_a$ on $\mathfrak{g}$, known as the *generators* of $\mathfrak{g}$, so that an arbitrary element $X \in \mathfrak{g}$ can be represented as $X = \sum_a \alpha_a T_a$ with $\alpha_a \in F$. In this basis, the entire structure of the Lie algebra can be captured by the *structure constants* $f_{abc}$, defined as

$$[T_a, T_b] = i\hbar \sum_c f_{abc} T_c, \tag{2.15}$$

where $\hbar$ is the reduced Planck's constant.[1] We can choose a particular set of generators orthonormalised such that

$$\operatorname{tr}(T_a T_b) = C\hbar^2 \delta_{ab}, \tag{2.16}$$

for some convenient real number $C$. In such a basis, the structure constants are totally antisymmetric; to wit

$$i\hbar^3 C f_{abc} = \operatorname{tr}([T_a, T_b] T_c) = \operatorname{tr}([T_c, T_a] T_b) = \operatorname{tr}([T_b, T_c] T_a). \tag{2.17}$$

---

[1]The factor of $i\hbar$ is a purely physicist's convention and does not appear in pure maths literature. Physicists also typically set $\hbar$ to 1.

- Example: the generator of $\mathfrak{so}(2)$ is given by the second Pauli matrix

$$J_1 = \hbar \begin{pmatrix} 0 & -i \\ i & 0 \end{pmatrix}, \tag{2.18}$$

  with $f_{111} = 0$ and $C = 2$. Similarly, the generators of $\mathfrak{so}(3)$ are given by

$$J_1 = \hbar \begin{pmatrix} 0 & 0 & 0 \\ 0 & 0 & -i \\ 0 & i & 0 \end{pmatrix}, \qquad J_2 = \hbar \begin{pmatrix} 0 & 0 & i \\ 0 & 0 & 0 \\ -i & 0 & 0 \end{pmatrix}, \qquad J_3 = \hbar \begin{pmatrix} 0 & -i & 0 \\ i & 0 & 0 \\ 0 & 0 & 0 \end{pmatrix}, \tag{2.19}$$

  with $f_{abc} = \epsilon_{abc}$ being the Levi-Civita symbol, and $C = 2$.

- Example: the generators of $\mathfrak{su}(2)$ are given by the Pauli matrices (normalised by $1/2$)

$$T_1 = \frac{\hbar}{2} \begin{pmatrix} 0 & 1 \\ 1 & 0 \end{pmatrix}, \qquad T_2 = \frac{\hbar}{2} \begin{pmatrix} 0 & -i \\ i & 0 \end{pmatrix}, \qquad T_3 = \frac{\hbar}{2} \begin{pmatrix} 1 & 0 \\ 0 & -1 \end{pmatrix}, \tag{2.20}$$

  with again $f_{abc} = \epsilon_{abc}$ and $C = 1/2$. This should be familiar as the angular-momentum algebra in quantum mechanics. Similarly, the generators of $\mathfrak{su}(3)$ are given by the *Gell-Mann matrices* (normalised by $\hbar/2$)

$$T_1 = \frac{\hbar}{2} \begin{pmatrix} 0 & 1 & 0 \\ 1 & 0 & 0 \\ 0 & 0 & 0 \end{pmatrix}, \qquad T_2 = \frac{\hbar}{2} \begin{pmatrix} 0 & -i & 0 \\ i & 0 & 0 \\ 0 & 0 & 0 \end{pmatrix}, \qquad T_3 = \frac{\hbar}{2} \begin{pmatrix} 1 & 0 & 0 \\ 0 & -1 & 0 \\ 0 & 0 & 0 \end{pmatrix},$$

$$T_4 = \frac{\hbar}{2} \begin{pmatrix} 0 & 0 & 1 \\ 0 & 0 & 0 \\ 1 & 0 & 0 \end{pmatrix}, \qquad T_5 = \frac{\hbar}{2} \begin{pmatrix} 0 & 0 & -i \\ 0 & 0 & 0 \\ i & 0 & 0 \end{pmatrix}, \qquad T_6 = \frac{\hbar}{2} \begin{pmatrix} 0 & 0 & 0 \\ 0 & 0 & 1 \\ 0 & 1 & 0 \end{pmatrix},$$

$$T_7 = \frac{\hbar}{2} \begin{pmatrix} 0 & 0 & 0 \\ 0 & 0 & -i \\ 0 & i & 0 \end{pmatrix}, \qquad T_8 = \frac{\hbar}{2\sqrt{3}} \begin{pmatrix} 1 & 0 & 0 \\ 0 & 1 & 0 \\ 0 & 0 & -2 \end{pmatrix}, \tag{2.21}$$

  with the structure constants

$$f_{123} = 1, \qquad f_{147} = f_{165} = f_{246} = f_{257} = f_{345} = f_{376} = \frac{1}{2}, \qquad f_{458} = f_{678} = \frac{\sqrt{3}}{2}, \tag{2.22}$$

  while all others either 0 or determined by the total anti-symmetry of $f_{abc}$, and $C = 1/2$.

- Example: The $\mathfrak{su}(2)$ and $\mathfrak{su}(3)$ algebras can be extended to $\mathfrak{u}(2)$ and $\mathfrak{u}(3)$ respectively by including an additional generator

$$T_0 = \frac{\hbar}{2} \mathbb{1}_{2\times 2} \qquad \text{and} \qquad T_0 = \frac{\hbar}{\sqrt{6}} \mathbb{1}_{3\times 3}, \tag{2.23}$$

  respectively. The additional structure constants $f_{0ab} = 0$, because $T_0$ commutes with all the other generators in the respective algebras, while $C = 1/2$. The $T_0$ generator for generic $\mathfrak{u}(N)$ is given by $\hbar/\sqrt{2N}\,\mathbb{1}_{N\times N}$.

- NB: Note that the Lie algebra $\mathfrak{su}(1)$ is trivial and only contains the element 0, as there are no non-zero $1 \times 1$ traceless Hermitian matrices. The Lie algebra $\mathfrak{u}(1)$, on the other hand, has generator $T_0 = \hbar/\sqrt{2}$, with $f_{000} = 0$ and $C = 1/2$. Note that $\mathfrak{u}(1)$ is just the set of real numbers.

**Lie algebra isomorphism:**   Two Lie algebras $\mathfrak{g}$ and $\mathfrak{g}'$ over a field $F$ are said to be isomorphic, denoted as $\mathfrak{g} \cong \mathfrak{g}'$, if there exists an invertible map $\iota : \mathfrak{g} \to \mathfrak{g}'$ such that $\iota(aX + bY) = a\iota(X) + b\iota(Y)$ and $\iota([X, Y]) = [\iota(X), \iota(Y)]$, where $X, Y \in \mathfrak{g}$ are arbitrary elements and $a, b \in F$.

- NB: given two isomorphic Lie algebras $\mathfrak{g} \cong \mathfrak{g}'$ and the generators $T_a$ of $\mathfrak{g}$, the set $\iota(T_a)$ serves as generators of $\mathfrak{g}'$. In this basis, the structure constants of the isomorphic Lie algebras $\mathfrak{g}$ and $\mathfrak{g}'$ are the same.

- Example: the Lie algebras $\mathfrak{so}(2)$ and $\mathfrak{u}(1)$ are isomorphic, with the isomorphism

$$\iota(\theta \in \mathfrak{u}(1)) = \theta \, J_1 \in \mathfrak{so}(2). \tag{2.24}$$

  The Lie algebras $\mathfrak{so}(3)$ and $\mathfrak{su}(2)$ are also isomorphic, with the isomorphism

$$\iota(\theta_a T_a \in \mathfrak{su}(2)) = \theta_a J_a \in \mathfrak{so}(3). \tag{2.25}$$

  However, unlike $\mathrm{SO}(2) \cong \mathrm{U}(1)$, the groups $\mathrm{SO}(3)$ and $\mathrm{SU}(2)$ are not isomorphic; more on this later in section 2.5.

**Direct sum of Lie algebras:**   Given two Lie algebras $\mathfrak{g}$ and $\mathfrak{g}'$ over a field $F$, the direct sum Lie algebra is given by $\mathfrak{g} \oplus \mathfrak{g}' = \{(X, X') \mid X \in \mathfrak{g}, X' \in \mathfrak{g}'\}$, such that

$$a(X, X') + b(Y, Y') = (aX + bY, aX' + bY') \quad \forall \, a, b \in F, \tag{2.26}$$

with the obvious commutator operation

$$[(X, X'), (Y, Y')] = ([X, Y], [X', Y']). \tag{2.27}$$

- Example: the Lie algebra $\mathfrak{u}(N)$ is isomorphic to the direct sum Lie algebra $\mathfrak{su}(N) \oplus \mathfrak{u}(1)$, with the isomorphism given by

$$\iota\Big((\lambda_a T_a, \hbar\lambda_0) \in \mathfrak{su}(N) \oplus \mathfrak{u}(1)\Big) = \lambda_a T_a + \sqrt{2N}\,\lambda_0 T_0 \in \mathfrak{u}(N). \tag{2.28}$$

  Here $T_a$ with $a = 1, 2, \ldots, N^2 - 1$ are the generators of $\mathfrak{su}(N)$, while the generator $T_0$ is proportional to the identity matrix. It should again be noted that the group $\mathrm{U}(N)$ is not isomorphic to the product group $\mathrm{SU}(N) \times \mathrm{U}(1)$; more on this later in section 2.5.

**Lie subalgebra:**   Given a Lie algebra $\mathfrak{g}$, a subset $\mathfrak{h} \subset \mathfrak{g}$ is called a *Lie subalgebra* of $\mathfrak{g}$ if it is closed under the Lie bracket operation: $[X, Y] \in \mathfrak{h}$ for all $X, Y \in \mathfrak{h}$.

Given a Lie algebra $\mathfrak{g}$, a Lie subalgebra $\mathfrak{h} \subset \mathfrak{g}$ is said to be an *invariant subalgebra* or *ideal* if $[X, Y] \in \mathfrak{h}$ for all $X \in \mathfrak{g}$ and $Y \in \mathfrak{h}$.

- NB: given a Lie algebra $\mathfrak{g}$, the trivial algebra $\{0\}$ and the Lie algebra $\mathfrak{g}$ itself are Lie subalgebras.

- NB: given two Lie algebras $\mathfrak{g}$ and $\mathfrak{g}'$, the direct sum Lie algebra $\mathfrak{g} \oplus \mathfrak{g}'$ has invariant subalgebras $\mathfrak{g} \cong \mathfrak{g} \oplus \{0\}$ and $\mathfrak{g}' \cong \{0\} \oplus \mathfrak{g}'$.

- Example: $\mathfrak{sl}(N, \mathbb{R}) \subset \mathfrak{gl}(N, \mathbb{R})$, $\mathfrak{sl}(N, \mathbb{C}) \subset \mathfrak{gl}(N, \mathbb{C})$, $\mathfrak{so}(N) \subset \mathfrak{sl}(N, \mathbb{R})$, $\mathfrak{u}(N) \subset \mathfrak{gl}(N, \mathbb{C})$, $\mathfrak{su}(N) \subset \mathfrak{sl}(N, \mathbb{C})$, $\mathfrak{su}(N) \subset \mathfrak{u}(N)$ are examples of Lie subalgebras. Moreover, $\mathfrak{sl}(N, \mathbb{R}) \subset \mathfrak{gl}(N, \mathbb{R})$, $\mathfrak{sl}(N, \mathbb{C}) \subset \mathfrak{gl}(N, \mathbb{C})$, and $\mathfrak{su}(N) \subset \mathfrak{u}(N)$ are invariant subalgebras.

- NB: given a Lie algebra $\mathfrak{g}$ over a field $F$ with generators $T_a$, it admits an infinite number of $\mathfrak{u}(1)$ subalgebras spanned by any linear combination of the generators: $\mathfrak{h}(\lambda_a) = \{\alpha \lambda_a T_a \mid \alpha \in \mathbb{R}\}$ for any $\lambda_a \in F$.

- Example: The Lie algebra $\mathfrak{su}(3)$, with generators $T_{1,\ldots,8}$ given in eq. (2.21), has an $\mathfrak{su}(2)$ subalgebra spanned by the generators $T_{1,2,3}$.

**Simple and semi-simple Lie algebra:**   A Lie algebra $\mathfrak{g}$ is said to be *simple* if it does not admit any invariant subalgebras except the trivial algebra $\{0\}$ and $\mathfrak{g}$ itself.[2]   A Lie algebra is said to be *semi-simple* if it is a direct sum of simple Lie algebras.

- Example: $\mathfrak{sl}(N,\mathbb{C})$, $\mathfrak{sl}(N,\mathbb{R})$ and $\mathfrak{su}(N)$ for $N \geq 1$ are simple Lie algebras.

- Example: $\mathfrak{u}(N) \cong \mathfrak{su}(N) \oplus \mathfrak{u}(1)$ for $N \geq 1$ is a semi-simple Lie algebra.

**Centre:**   The centre $Z(\mathfrak{g}) = \{X \in \mathfrak{g} \mid [X,Y] = 0 \ \forall Y \in \mathfrak{g}\}$ of a Lie algebra $\mathfrak{g}$ is defined to the maximal subset of $\mathfrak{g}$ that commutes with all the elements of $\mathfrak{g}$.

- NB: the centre of a Lie algebra is an invariant Lie subalgebra.

- NB: the centre of an Abelian Lie algebra is itself.

**Quotient algebra:**   Given a Lie algebra $\mathfrak{g}$ and an invariant subalgebra $\mathfrak{h}$, the quotient algebra of $\mathfrak{h}$ in $\mathfrak{g}$ is defined as $\mathfrak{g}/\mathfrak{h} = \{X + \mathfrak{h} \mid X \in \mathfrak{g}\}$, where $X + \mathfrak{h} = \{X + Y \mid Y \in \mathfrak{h}\}$ for all $X \in \mathfrak{g}$. The commutator on the quotient algebra is defined as $[X + \mathfrak{h}, Y + \mathfrak{h}] = [X,Y] + \mathfrak{h}$.

- NB: given a direct sum Lie algebra $\mathfrak{g} \oplus \mathfrak{g}'$ and the invariant subalgebra $\mathfrak{g} \cong \mathfrak{g} \oplus \{0\}$, we have that $(\mathfrak{g} \oplus \mathfrak{g}')/\mathfrak{g} \cong \mathfrak{g}'$. Similarly $(\mathfrak{g} \oplus \mathfrak{g}')/\mathfrak{g}' \cong \mathfrak{g}$.

**Lie algebra representation:**   A *matrix representation* $D(\mathfrak{g})$ of a Lie algebra $\mathfrak{g}$ defined over a field $F$ is a linear map onto the set of matrices $D : \mathfrak{g} \to \mathfrak{gl}(N)$ satisfying

1. $D(aX + bY) = aD(X) + bD(Y)$ for all $X, Y \in \mathfrak{g}$ and $a, b \in F$.
2. $[D(X), D(Y)] = D([X,Y])$ for all $X, Y \in \mathfrak{g}$.

A Lie algebra representation is said to be *faithful* if the mapping $D(\mathfrak{g})$ is one-to-one and invertible. In particular, for a faithful representation, $D(X) = 0 \implies X = 0$. A representation $D(\mathfrak{g})$ of a Lie algebra $\mathfrak{g}$ is said to be *Hermitian* if $D(\mathfrak{g})$ is a Hermitian matrix for all $X \in \mathfrak{g}$, i.e. $D(X)^\dagger = D(X)$.

The definitions of equivalent representations, direct sum of representations, and reducible, irreducible, and completely reducible representations for Lie algebras are the same as for groups in section 2.3.

- NB: a faithful representation of a Lie algebra is isomorphic to the Lie algebra.

- NB: we state without proof that a Hermitian Lie algebra representation is completely reducible as a direct sum of irreducible Hermitian Lie algebra representations.

---

[2]Some authors restrict simple Lie algebras and Lie groups to be non-Abelian.

**Adjoint representation:** The *adjoint representation* is a $\dim \mathfrak{g} \times \dim \mathfrak{g}$ representation furnished by the structure constants, with the matrix elements of the generators given by

$$(T_a^{\text{adjoint}})_{bc} = -i\hbar f_{abc}. \tag{2.29}$$

Note that the Jacobi's identity implies

$$\sum_d (f_{bcd}f_{ade} + f_{abd}f_{cde} + f_{cad}f_{bde}) = 0$$

$$\implies \left[T_a^{\text{adjoint}}, T_b^{\text{adjoint}}\right]_{de} = i\hbar \sum_c f_{abc}(T_c^{\text{adjoint}})_{de}. \tag{2.30}$$

- Example: the $\mathfrak{so}(3)$ algebra given in eq. (2.19) is already in the adjoint representation. The adjoint representation for $\mathfrak{su}(2)$ is also given by eq. (2.19). This supports $\mathfrak{so}(3) \cong \mathfrak{su}(2)$.

- Example: the adjoint representation for $\mathfrak{su}(3)$ can be worked out using the structure constants in eq. (2.22).

- Example: The adjoint representation for $\mathfrak{u}(N)$ is not faithful, because $D(J_0) = 0$. Similarly, the adjoint representations for $\mathfrak{so}(2)$ and $\mathfrak{u}(1)$ are not faithful.

## 2.5 Lie groups

After our little excursion into Lie algebras, let us return to group theory. We will now focus on continuous groups, called Lie groups. These play an important role in quantum field theory as the groups of continuous symmetry operations such as rotations. Lie groups will also formalise the relation between Lie algebras and group theory.

**Lie group:** A *Lie group* $G$ is an infinite-dimensional group whose elements $g(\alpha)$ can be smoothly labelled by finite set of continuous parameters $\alpha = \{\alpha_1, \alpha_2, \ldots\}$, such that the group multiplication is a smooth operation

$$g(\alpha)g(\beta) = g(\gamma(\alpha, \beta)), \qquad \gamma_i(\alpha, \beta) \text{ are smooth functions of } \alpha_i \text{ and } \beta_i. \tag{2.31}$$

- Example: the orthogonal group O(2) and the special orthogonal group SO(2) from our examples in section 2.1 are Lie groups, with the smooth mapping furnished by eq. (2.5).

- Example: other examples of matrix Lie groups are

  - $GL(N, \mathbb{R})$, $GL(N, \mathbb{C})$: real and complex $N \times N$ invertible matrices.
  - $SL(N, \mathbb{R})$, $SL(N, \mathbb{C})$: real and complex $N \times N$ matrices with unit determinant, $\det M = 1$.
  - O(N): real $N \times N$ orthogonal matrices, $MM^{\text{T}} = \mathbb{1}$ — the symmetry group of an $N$-dimensional sphere.
  - SO(N): real $N \times N$ orthogonal matrices with unit determinant, $MM^{\text{T}} = \mathbb{1}$, $\det M = 1$ — the symmetry group of rotations of an $N$-dimensional sphere.
  - U(N): complex $N \times N$ unitary matrices, $MM^\dagger = \mathbb{1}$.
  - SU(N): complex $N \times N$ unitary matrices with unit determinant, $MM^\dagger = \mathbb{1}$, $\det M = 1$.

**Lie subgroup:** A subgroup $H \subset G$ is said to be a *Lie subgroup* of $G$ if it forms a Lie group itself.

- Example: $SL(N, \mathbb{R}) \subset GL(N, \mathbb{R})$, $SL(N, \mathbb{C}) \subset GL(N, \mathbb{C})$, $O(N) \subset GL(N, \mathbb{R})$, $SO(N) \subset SL(N, \mathbb{R})$, $U(N) \subset GL(N, \mathbb{C})$, $SU(N) \subset SL(N, \mathbb{C})$, $SO(N) \subset O(N)$, and $SU(N) \subset U(N)$ are all examples of Lie subgroups. Moreover, $SL(N, \mathbb{R}) \subset GL(N, \mathbb{R})$, $SL(N, \mathbb{C}) \subset GL(N, \mathbb{C})$, $SO(N) \subset O(N)$, and $SU(N) \subset U(N)$ are normal/invariant subgroups.

**Connected Lie groups:** A Lie group $G$ is said to be *connected* if every group element can be continuously connected to the identity element, i.e. there exists a parametrisation $g(\alpha)$ of $G$ so that $g(\alpha = 0) = e$ for all $g(\alpha) \in G$.

The *connected part* of a Lie group $G$ is the Lie subgroup of $G$ comprising of all the elements of $G$ that are continuously connected to the identity element. The connected part of a Lie group is a connected Lie group itself.

- $SO(N)$ is the connected part of $O(N)$. This is because an $O(N)$ matrix has either determinant 1 or $-1$, while an $SO(N)$ matrix has determinant 1. The identity matrix has determinant 1, which cannot be flipped to $-1$ using continuous deformations.

- Unlike the orthogonal case, both $SU(N)$ and $U(N)$ are connected Lie groups. This is because a $U(N)$ matrix has determinant $\exp(i\theta)$, and these can be continuously arrived at starting from the identity matrix by moving around the complex unit circle.

**Simple Lie groups:** A Lie group $G$ is said to be simple if is connected and has no connected normal Lie subgroups except the trivial group $\{e\}$ and $G$ itself; see footnote 2.

- NB: a connected Lie group that is simple as a group is automatically a simple Lie group. However, a simple Lie group is not necessarily simple as a group. In particular, a simple Lie group is allowed to have discrete normal subgroups.

- Example: $U(1)$ is a simple Lie group. However, $U(1)$ is not simple as group because it has discrete normal subgroups.

**Exponential parametrisation:** Given a faithful representation $D(G)$ of a Lie group $G$, a group element $g(\alpha)$ arbitrarily close to the identity element $e$ can be represented as

$$D(g(\alpha)) = \mathbb{1} + \frac{i}{\hbar} \sum_a \alpha_a D(T_a) + O(\alpha^2). \tag{2.32}$$

Here $T_a$ are abstract objects known as the generators of $G$, while $D(T_a)$ is a faithful representation of the generators. The connected part of $G$ in the representation $D(G)$ can be generated by multiplying together an infinite number of infinitesimal elements $g(\alpha/k)$ with $k \to \infty$, i.e.

$$D(g(\alpha)) = \lim_{k \to \infty} \left( \mathbb{1} + \frac{i}{\hbar k} \alpha_a D(T_a) \right)^k = \exp\left(i/\hbar \, \alpha_a D(T_a)\right). \tag{2.33}$$

Summation over the index $a$ is understood. This is known as the *exponential parametrisation* of the connected part of the Lie group.

The generators $T_a$ span the Lie algebra $\mathfrak{g} = \{\lambda_a T_a \mid \lambda_a \in \mathbb{R}\}$ associated with the Lie group $G$, with the Lie bracket derived from the commutator operation on the representation space: $[X, Y] = D^{-1}([D(X), D(Y)])$ for $X, Y \in \mathfrak{g}$. Note that $D(G)$ is a faithful representation, so the map $D$ is invertible. Matrix commutators identically satisfy all the properties of a Lie bracket. Closedness of the Lie algebra under commutators follows from the group product

$$\exp\left(\frac{i}{\hbar}\sum_a \alpha_a D(T_a)\right)\exp\left(\frac{i}{\hbar}\sum_a \beta_a D(T_a)\right) = \exp\left(\frac{i}{\hbar}\sum_a \gamma_a D(T_a)\right) \quad \text{for some } \gamma_a$$

$$\implies \gamma_a D(T_a) = \alpha_a D(T_a) + \beta_a D(T_a) + \frac{i}{2\hbar}\alpha_a\beta_b[D(T_a), D(T_b)] + \dots$$

$$\implies [D(T_a), D(T_b)] = i\hbar f_{abc} D(T_c) \quad \text{for some } f_{abc}$$

$$\text{and} \qquad \gamma_a = \alpha_a + \beta_a + \frac{i}{2}f_{bca}\alpha_b\beta_c + \dots. \tag{2.34}$$

The structure constants $f_{abc}$ only depend on the structure of the group and not on the representation employed. The exponential parametrisation of a Lie group $G$ and the relation to the associated Lie algebra $\mathfrak{g}$ can also be established without making reference to a faithful representation, however we will not concern ourselves with these formalities.

- Example: Lie algebras associated with the aforementioned matrix groups are

  - $\mathrm{GL}(N, \mathbb{R}) \to \mathfrak{gl}(N, \mathbb{R})$, $\mathrm{GL}(N, \mathbb{C}) \to \mathfrak{gl}(N, \mathbb{C})$.
  - $\mathrm{SL}(N, \mathbb{R}) \to \mathfrak{sl}(N, \mathbb{R})$, $\mathrm{SL}(N, \mathbb{C}) \to \mathfrak{sl}(N, \mathbb{C})$.
  - $\mathrm{O}(N), \mathrm{SO}(N) \to \mathfrak{so}(N)$.
  - $\mathrm{U}(N) \to \mathfrak{u}(N)$, $\mathrm{SU}(N) \to \mathfrak{su}(N)$.

- Example: note that the Lie algebra for both $\mathrm{O}(N)$ and $\mathrm{SO}(N)$ groups is $\mathfrak{so}(N)$. This is because the connected part of $\mathrm{O}(N)$ is its subgroup $\mathrm{SO}(N)$.

- NB: let $G$, $G'$ be connected Lie groups with associated Lie algebras $\mathfrak{g}$, $\mathfrak{g}'$. Exponentiating the direct sum Lie algebra $\mathfrak{g} \oplus \mathfrak{g}'$ leads to the direct product Lie group $G \times G'$ and vice-versa.

- NB: given a Lie group $G$ with Lie algebra $\mathfrak{g}$, and a Lie subgroup $H \subset G$, the Lie algebra $\mathfrak{h}$ associated with $H$ is a Lie subalgebra of $\mathfrak{g}$.

- NB: A simple Lie algebra generates a *simple Lie group* via exponentiation.

- NB: given a Lie group $G$ and the associated Lie algebra $\mathfrak{g}$, the concepts of equivalent representations, direct sum of representations, and reducible, irreducible, and completely reducible representations of invertibly translate into each other.

- NB: given a Lie group $G$ and the associated Lie algebra $\mathfrak{g}$, a Hermitian representation of a Lie algebra generates a unitary representation of the Lie group and vice-versa.

**Isomorphic Lie algebras vs isomorphic Lie groups:**   Let $G \cong G'$ be isomorphic connected Lie groups, then the associated Lie algebras are also isomorphic $\mathfrak{g} \cong \mathfrak{g}'$. However, the converse is *not* true: given two isomorphic Lie algebras $\mathfrak{g} \cong \mathfrak{g}'$, the connected Lie groups $G$, $G'$ generated via exponentiation need not be isomorphic.

- Example: consider the isomorphic Lie algebras $\mathfrak{su}(2) \cong \mathfrak{so}(3)$ with the isomorphism $\iota : \mathfrak{su}(2) \to \mathfrak{so}(3)$ given in eq. (2.25). Upon exponentiation, this defines a map $\iota : \mathrm{SU}(2) \to \mathrm{SO}(3)$ as

$$\iota\big( \exp(i/\hbar\,\theta_a T_a) \in \mathrm{SU}(2)\big) = \exp(i/\hbar\,\iota(\theta_a T_a)) = \exp(i/\hbar\,\theta_a J_a) \in \mathrm{SO}(3). \tag{2.35}$$

However, this is not an invertible map because multiple entries in $\mathrm{SU}(2)$ are mapped to the same entry in $\mathrm{SO}(3)$. Take, for example, $\theta_a = (0,0,0)$ and $\theta_a = (0,0,2\pi)$. This results in distinct elements $\exp(i/\hbar\,\theta_a T_a) = \mathbb{1}$ and $\exp(i/\hbar\,\theta_a T_a) = \exp(2\pi i/\hbar\,T_3) = -\mathbb{1}$ in $\mathrm{SU}(2)$, being mapped to the same element $\exp(i/\hbar\,\theta_a J_a) = \mathbb{1}$ and $\exp(i/\hbar\,\theta_a J_a) = \exp(2\pi i/\hbar\,J_3) = \mathbb{1}$ in $\mathrm{SO}(3)$. Therefore, the Lie groups $\mathrm{SU}(2)$ and $\mathrm{SO}(3)$ are not isomorphic. In fact, $\mathrm{SU}(2)$ is the double-cover of $\mathrm{SO}(3)$, where every element is covered twice.

  This has important implications in quantum mechanics. Fundamental particles in nature do not transform under unitary representations of the rotation group $\mathrm{SO}(3)$, but of its double-cover $\mathrm{SU}(2)$. This leads to the existence of fermions, which are particles that need to be completely rotated in space twice before returning to their original state. This is because a single $2\pi$ rotation in $\mathrm{SO}(3)$ corresponds to the inversion operator $-\mathbb{1}$ in $\mathrm{SU}(2)$, and a double $4\pi$ rotation is needed to return to the original state.

- Example: consider the isomorphic Lie algebras $\mathfrak{su}(N) \oplus \mathfrak{u}(1) \cong \mathfrak{u}(N)$ with the isomorphism $\iota : \mathfrak{su}(N) \oplus \mathfrak{u}(1) \to \mathfrak{u}(N)$ given in eq. (2.28). Upon exponentiation, this defines a map among the Lie groups $\iota : \mathrm{SU}(N) \times \mathrm{U}(1) \to \mathrm{U}(N)$ given as

$$
\begin{aligned}
\iota\Big( (\exp(i/\hbar\,\lambda_a T_a), \exp(i\lambda_0)) \in \mathrm{SU}(N) \times \mathrm{U}(1)\Big) &= \exp\big( i/\hbar\,\iota\big((\lambda_a T_a, \hbar\lambda_0)\big)\big) \\
&= \exp(i/\hbar\,\lambda_a T_a + i\lambda_0 \mathbb{1}) \\
&= \mathrm{e}^{i\lambda_0} \exp(i/\hbar\,\lambda_a T_a) \in \mathrm{U}(N). \tag{2.36}
\end{aligned}
$$

Or more succinctly, given $(U,\alpha) \in \mathrm{SU}(N) \times \mathrm{U}(1)$, we have $\iota((U,\alpha)) = \alpha U \in \mathrm{U}(N)$. However, this is again not an invertible map. Let $\omega$ be one of the $N$ distinct $N$th roots of identity. Then $(\omega U, \alpha/\omega) \in \mathrm{SU}(N) \times \mathrm{U}(1)$ map to the same element $\alpha U \in \mathrm{U}(N)$ for every $\omega$. Therefore, the Lie groups $\mathrm{SU}(N) \times \mathrm{U}(1)$ and $\mathrm{U}(N)$ are not isomorphic for $N > 1$. In fact, $\mathrm{SU}(N) \times \mathrm{U}(1)$ covers $\mathrm{U}(N)$ a total of $N$-times.

# 3 | U(1) group

U(1) is the simplest Lie group. It is the group of all $1 \times 1$ unitary matrices, which amounts to the set of all unit norm complex numbers. U(1) is an Abelian group, because complex numbers commute among themselves. U(1) is also a simple Lie group. An arbitrary element $\alpha \in$ U(1) can be parametrised as

$$\alpha = \exp(i\theta), \qquad \text{where} \quad \theta \in [0, 2\pi). \tag{3.1}$$

As mentioned in eq. (2.9), U(1) is isomorphic to SO(2), the group of all $2 \times 2$ real orthogonal matrices. Therefore, U(1) can also be understood as the group of all two-dimensional rotations.

The Lie algebra of U(1) is $\mathfrak{u}(1)$, which is the set of all $1 \times 1$ Hermitian matrices or, in other words, the set of all real numbers $\mathbb{R}$. The generator of U(1) can be taken to be $T_0 = \hbar$. The structure constant $f_{000} = 0$ because $[T_0, T_0] = 0$. On the other hand, $\text{tr}(T_0 T_0) = \hbar^2$, so $C = 1$ in this basis.

The irreducible representations of U(1) can be characterised by a real number $q$. To wit, the irreducible representation $D_q$ of U(1) is defined as

$$D_q(\mathrm{e}^{i\theta}) = \mathrm{e}^{iq\theta} \qquad \forall \mathrm{e}^{i\theta} \in \mathrm{U}(1) \quad \text{and} \quad \theta \in \mathbb{R}. \tag{3.2}$$

Correspondingly, the action of $D_q$ on the Lie algebra is given as $D_q(\theta) = q\theta$ for all $\theta \in \mathfrak{u}(1)$. In terms of the generator, $D_q(T_0) = q\hbar$. Note that the conjugate representation to $D_q$ is $D_{-q}$. A complex field $\phi$ is said to transform in the $D_q$ representation of U(1), or to have charge $q$ under U(1), if $\mathrm{e}^{i\theta} \in$ U(1) acts as

$$\phi \to \phi' = \mathrm{e}^{iq\theta}\phi. \tag{3.3}$$

The Abelian Lie group U(1) is different from its non-Abelian cousins, which we will study in the next section, in that its irreducible representations are parametrised by a continuous real parameter $q$. In other words, U(1) charges of fields can be fractional or even irrational. This is a good news for quarks which are observed to have fractional charges under the electromagnetic U(1) symmetry. However, it does beg the question why the electromagnetic charges of fundamental fields in our universe are quantised at all.

# 4 | Special unitary group

In this section we specialise to the special unitary group $\mathrm{SU}(N)$, associated with "internal" symmetries in the Standard Model of particle physics. This is the group comprising of $N \times N$ unit determinant unitary matrices $U \in \mathrm{SU}(N)$ satisfying $UU^\dagger = U^\dagger U = \mathbb{1}$ and $\det U = 1$. The unitarity condition can be represented in the index notation as

$$U_i{}^k U^j{}_k = U_k{}^j U^k{}_i = \delta_i^j, \tag{4.1}$$

where $U_i{}^j$ denotes the components of $U$ and $U^j{}_i = (U_j{}^i)^*$ denotes the components of $(U^\dagger)^{\mathrm{T}}$. The repeated indices are understood to be summed over $i, j, k, \ldots = 1, 2, \ldots, N$. $\mathrm{SU}(N)$ is a Lie group and the associated Lie algebra $\mathfrak{su}(N)$ is the set of all $N \times N$ traceless Hermitian matrices. The dimension of $\mathfrak{su}(N)$ as a vector-space is $N^2 - 1$ and the respective generators are denoted as $T_a$ with $a = 1, 2, \ldots, N^2 - 1$. We study the representations of $\mathrm{SU}(N)$ and introduce the method of Young tableaux to classify all the irreducible representations. Later in the section, we specialise to the specific examples of the first few special unitary groups $\mathrm{SU}(2)$ and $\mathrm{SU}(3)$, which will be helpful in the forthcoming discussion of the Standard Model. Majority of the discussion here is derived from [9].

## 4.1 SU(N) representations

**Fundamental representation:** The identity map $D(U)_i{}^j = U_i{}^j$ for all $U \in \mathrm{SU}(N)$ furnishes an obvious representation of the group $\mathrm{SU}(N)$, known as the *fundamental representation*. An $N$-component complex field $\psi_i = (\psi_1, \ldots, \psi_N)$ is said to transform in the fundamental representation of $\mathrm{SU}(N)$ if an arbitrary group transformation $U \in \mathrm{SU}(N)$ acts as

$$\psi_i \to \psi_i' = U_i{}^j \psi_j. \tag{4.2}$$

The fundamental representation is often denoted by the dimension "$\mathbf{N}$" of the vector $\psi_i$ (for instance, "$\mathbf{3}$" for the fundamental representation of $\mathrm{SU}(3)$). Suppressing the indices, the transformation rule for a fundamental field can be represented as $\psi \to U\psi$.

**Anti-fundamental representation:** The *anti-fundamental representation* is the conjugate representation to the fundamental representation, given as $D(U)^i{}_j = (U_i{}^j)^*$. An $N$-component complex field $\bar\psi^i$ is said to transform in the anti-fundamental representation of $\mathrm{SU}(N)$ if an arbitrary group transformation $U \in \mathrm{SU}(N)$ acts as

$$\bar\psi^i \to \bar\psi'^i = U^i{}_j \bar\psi^j. \tag{4.3}$$

The anti-fundamental representation is often denoted with a bar over the dimension "$\bar{\mathbf{N}}$". Suppressing the indices, the transformation rule for an anti-fundamental field can be represented as $\bar\psi \to \bar\psi U^\dagger$.

- NB: given a fundamental vector field $\psi_i$, the complex conjugate vector $\bar\psi^i = (\psi_i)^*$ transforms in the anti-fundamental representation. Similarly, given an anti-fundamental vector field $\bar\psi^i$, the complex conjugate vector $\psi_i = (\bar\psi^i)^*$ transforms in the fundamental representation.

**Singlet representation:**  The trivial map $D(U) = 1$ for all $U \in \mathrm{SU}(N)$ furnishes the *trivial* or *singlet representation* of the group $\mathrm{SU}(N)$. A complex field $\phi$ is said to transform in the *singlet representation* of $\mathrm{SU}(N)$ if it is invariant under a $U \in \mathrm{SU}(N)$ transformation

$$\phi \to \phi' = \phi. \tag{4.4}$$

The trivial representation is denoted by "**1**".

- NB: given a fundamental field $\psi_i$, we can create a singlet field as $\psi_i \bar{\psi}^i$.

**Adjoint representation:**  A complex field $\Psi_i{}^j$, with $i, j = 1, \ldots, N$ and the trace $\Psi^i{}_i = 0$, is said to transform in the *adjoint representation* of $\mathrm{SU}(N)$ if

$$\Psi_i{}^j \to \Psi_i'{}^j = U_i{}^k U^j{}_l \Psi_k{}^l. \tag{4.5}$$

Suppressing the indices, the transformation rule is given as $\Psi \to U \Psi U^\dagger$. The adjoint representation is denoted by the dimension "$\mathbf{N^2 - 1}$" of the field $\Psi_i{}^j$.

- NB: given a fundamental field $\psi_i$, we can create an adjoint field as $\psi_i \bar{\psi}^j - 1/N\, \delta_i^j \psi_k \bar{\psi}^k$.
- NB: elements of the Lie algebra $\mathfrak{su}(N)$, i.e. $N \times N$ traceless Hermitian matrices, transform in the adjoint representation of $\mathrm{SU}(N)$. Note that

$$X^\dagger = X \implies (UXU^{-1})^\dagger = UXU^{-1}, \qquad \mathrm{tr}\, X = 0 \implies \mathrm{tr}(UXU^{-1}) = 0. \tag{4.6}$$

- NB: the relation to the adjoint representation (2.29) of the Lie algebra $\mathfrak{su}(N)$ can be established as follows: given the $\mathfrak{su}(N)$ generators $T_a$ and the exponential representation $U = \exp(i/\hbar\, \theta^a T_a)$, the transformation rule of $T_a$ is given as

$$T_a \to U T_a U^{-1} = \exp(i/\hbar\, \theta^b T_b^{\mathrm{adjoint}})^c{}_a T_c. \tag{4.7}$$

**Tensor representations:**  An arbitrary tensor field $\Psi_{i_i i_2 \ldots}^{j_1 j_2 \ldots}$ that behaves under an $\mathrm{SU}(N)$ transformation according to

$$\Psi_{i_1 i_2 \ldots}^{j_1 j_2 \ldots} \to \Psi_{i_1 i_2 \ldots}'^{j_1 j_2 \ldots} = (U_{i_1}{}^{k_1} U_{i_2}{}^{k_2} \ldots)(U^{j_1}{}_{l_1} U^{j_2}{}_{l_2} \ldots)\Psi_{k_1 k_2 \ldots}^{l_1 l_2 \ldots}, \tag{4.8}$$

is said to transform in a *tensor representation* of $\mathrm{SU}(N)$. Given two tensor fields $\Psi_{i_1 i_2 \ldots}^{j_1 j_2 \ldots}$ and $\Phi_{i_1 i_2 \ldots}^{j_1 j_2 \ldots}$, we can construct a higher-dimensional *product representation* as

$$(\Psi \otimes \Phi)_{i_1 i_2 \ldots k_1 k_2 \ldots}^{j_1 j_2 \ldots l_1 l_2 \ldots} = \Psi_{i_1 i_2 \ldots}^{j_1 j_2 \ldots} \Phi_{k_1 k_2 \ldots}^{l_1 l_2 \ldots}. \tag{4.9}$$

- NB: Kronecker delta symbol $\delta_i^j$ and Levi-Civita symbols $\epsilon_{i_1 i_2 \ldots i_N}$, $\epsilon^{i_1 i_2 \ldots i_N}$ are $\mathrm{SU}(N)$ invariants

$$\delta_i^j \to U_i{}^k U^j{}_l \delta_k{}^l = \delta_i^j,$$
$$\epsilon_{i_1 i_2 \ldots i_N} \to (U_{i_1}{}^{j_1} U_{i_2}{}^{j_2} \ldots U_{i_N}{}^{j_N})\epsilon_{j_1 j_2 \ldots j_N} = (\det U)\epsilon_{i_1 i_2 \ldots i_N},$$
$$\epsilon^{i_1 i_2 \ldots i_N} \to (U^{i_1}{}_{j_1} U^{i_2}{}_{j_2} \ldots U^{i_N}{}_{j_N})\epsilon^{j_1 j_2 \ldots j_N} = \frac{1}{\det U}\epsilon^{i_1 i_2 \ldots i_N}, \tag{4.10}$$

  with $\det U = 1$. Note that the Kronecker delta symbol is also invariant under the group $\mathrm{U}(N)$, while the Levi-Civita symbols are not, because $\det U$ is not necessarily 1.

**Reducible and irreducible representations:** A tensor field $\Psi^{j_1 j_2 \ldots}_{i_1 i_2 \ldots}$ is said to transform in a *reducible representation* of SU($N$), if it can be decomposed into a direct sum of fields transforming under smaller representations. A tensor field that does not transform under a reducible representation is said to transform in an *irreducible representation.*

The decomposition of a reducible tensor field $\Psi^{j_1 j_2 \ldots}_{i_1 i_2 \ldots}$ into fields transforming under smaller irreducible representations is known as its *tensor decomposition.*

- NB: tensor product of irreducible representations is generically reducible.
- Example: fundamental, anti-fundamental, singlet, and adjoint representations of SU($N$) are all irreducible.
- Example: A tensor field $\Psi_i{}^j$ (not necessarily traceless) is reducible into a singlet field $\Psi_i{}^i$ and an adjoint field $\Psi_i{}^j - 1/N\, \delta_i^j \Psi_k{}^k$. A tensor field $\Psi_{ij}$, on the other hand, is reducible into a symmetric tensor field $\Psi_{(ij)} = (\Psi_{ij} + \Psi_{ji})/2$ and an antisymmetric tensor field $\Psi_{[ij]} = (\Psi_{ij} - \Psi_{ji})/2$. Both of these transform independently under an SU($N$) transformation.

## 4.2 Young Tableaux

**Young tableaux:** We can generate arbitrary irreducible representations of SU($N$) using the method of Young tableaux. Firstly, given a tensor field $\Psi^{j_1 j_2 \ldots}_{i_1 i_2 \ldots}$, we can lower all the indices using the $\epsilon_{i_1 \ldots i_N}$ tensor to obtain the equivalent representation

$$\Psi_{i_1 i_2 \ldots k_1 k_2 \ldots l_1 l_2 \ldots} = \Psi^{j_1 j_2 \ldots}_{i_1 i_2 \ldots} \left( \epsilon_{j_1 k_1 l_1 \ldots} \epsilon_{j_2 k_2 l_2 \ldots} \ldots \right). \tag{4.11}$$

Given this, a typical Young tableaux representing an irreducible tensor field $\Psi_{i_1 j_1 \ldots}$ has the form

| $i_1$ | $j_1$ | $\cdots$ | | |
|---|---|---|---|---|
| $i_2$ | $j_2$ | $\cdots$ | | |
| $\vdots$ | $\vdots$ | $\ddots$ | | |
| | | | | |
| | | | | |

with rules

1. The indices in a given column have to be anti-symmetrised.
2. The indices in a given row have to be symmetrised.
3. A row cannot contain more boxes than the row above.
4. A column cannot contain more than $N$ boxes because of anti-symmetry. A column with $N$ boxes corresponds to an uncontracted factor of $\epsilon_{i_1 \ldots i_N}$ and can be removed.

The dimensions of an irreducible representation can be computed as

$$\dim = \frac{\prod_i (N + n_i)}{\prod_j d_j}, \qquad n_i :$$

| 0 | 1 | 2 | 3 | 4 | 5 |
|---|---|---|---|---|---|
| -1 | 0 | 1 | 2 | 3 | |
| -2 | -1 | 0 | 1 | 2 | |
| -3 | -2 | -1 | | | |
| -4 | | | | | |

$$, \qquad d_i :$$

| 10 | 8 | 7 | 5 | 4 | 1 |
|---|---|---|---|---|---|
| 8 | 6 | 5 | 3 | 2 | |
| 7 | 5 | 4 | 2 | 1 | |
| 4 | 2 | 1 | | | |
| 1 | | | | | |

$$\tag{4.12}$$

$n_i$ for the top-left box is 0; moving right we increase by 1, while moving down we decrease by 1. On the other hand, $d_i$ for a box is 1 plus the number of boxes below and to the right of the box.

- Examples:

Fundamental "**N**" $(\psi_i)$ : $\boxed{i}$ ,  Anti-fundamental "**N̄**" $(\psi_{[i_1...i_{N-1}]} = \epsilon_{ki_1...i_{N-1}}\psi^k)$ : $\begin{array}{c}\boxed{i_1}\\\boxed{i_2}\\\boxed{\vdots}\\\boxed{i_{N-1}}\end{array}$ ,

Singlet "**1**" $(\psi)$ : $\begin{array}{c}\boxed{i_1}\\\boxed{i_2}\\\boxed{\vdots}\\\boxed{i_N}\end{array}$ or null,  Adjoint "**N² − 1**" $(\psi_{j[i_1...i_{N-1}]} = \epsilon_{ki_1...i_{N-1}}\psi_j{}^k)$ : $\begin{array}{cc}\boxed{i_1}&\boxed{j}\\\boxed{i_2}\\\boxed{\vdots}\\\boxed{i_{N-1}}\end{array}$ ,

Symmetric 2-tensor "**N(N + 1)/2**" $(\psi_{(ij)})$ : $\boxed{i}\,\boxed{j}$ ,

Anti-symmetric 2-tensor "**N(N − 1)/2**" $(\psi_{[ij]})$ : $\begin{array}{c}\boxed{i}\\\boxed{j}\end{array}$ .                   (4.13)

**Tensor decomposition:**  Young tableaux can also be used to decompose the tensor product of irreducible representations into the direct sum of irreducible representations. To compute the tensor product of two representations $A$ and $B$, we use the algorithm:

1. Label all the boxes in the first row of $B$ as "$a_1$", in the second row as "$a_2$", in the third row as "$a_3$", and so on.

2. Attach the "$a_1$" boxes from $B$ to $A$ in all possible ways to produce legitimate tableaux. Repeat for "$a_2$" boxes, and so on. Discard any tableaux with repeated labels in a column.

3. For a given tableaux, create a sequence by enlisting all the labels from right to left, from top row to the bottom. Reading this sequence from left to right, discard any tableaux where the number of $a_i$'s is less than the number of $a_{i+1}$'s at any point in the sequence.

4. Tableaux with same structure are inequivalent only if the respective sequences are different.

In the final answer, the product of the dimensions of $A$ and $B$ must equal the sum of the dimensions of the reduced tableaux.

## 4.3  SU(2)

SU(2) is the simplest special unitary group. It is the group of all $2 \times 2$ unit determinant unitary matrices. The generators of the associated Lie algebra $\mathfrak{su}(2)$, i.e. the set of all $2 \times 2$ traceless Hermitian matrices, are given by $T_i = 1/2\,\sigma_i$, where $\sigma_i$ are the three Pauli matrices; see eq. (2.20). It is convenient to work in the basis

$$T_\pm = T_1 \pm iT_2, \qquad T_3, \tag{4.14}$$

with commutation relations

$$[T_+, T_-] = 2\hbar\, T_3, \qquad [T_3, T_\pm] = \pm\hbar\, T_\pm. \tag{4.15}$$

Note that $T_\pm^\dagger = T_\mp$. There is a Casimir operator in the theory $T^2 = T_1^2 + T_2^2 + T_3^2 = \frac{1}{2}\{T_+, T_-\} + T_3^2$ that commutes with all the operators in the algebra, i.e $[T^2, T_a] = 0$. Note that the anti-commutator of the generators is given by $\{T_i, T_j\} = \hbar^2/2\,\delta_{ij}\mathbb{1}$.

The fundamental and anti-fundamental representations of SU(2) are equivalent. This trivially follows using the rules of Young tableaux, as both of these can be related by a contraction with $\epsilon_{ij}$ and are represented by the same tableaux

$$\text{Fundamental “}\mathbf{2}\text{” } (\psi_i): \quad \boxed{i}\,, \qquad \text{Anti-fundamental “}\bar{\mathbf{2}}\text{” } (\epsilon_{ij}\psi^j): \quad \boxed{i}\,. \tag{4.16}$$

In fact, all inequivalent irreducible representations of SU(2) are given by

$$\text{“}\mathbf{1}\text{” } (j=0): \quad \text{null}, \qquad \text{“}\mathbf{2}\text{” } (j=1/2): \quad \square\,, \qquad \text{“}\mathbf{3}\text{” } (j=1): \quad \square\square\,,$$
$$\text{“}\mathbf{4}\text{” } (j=3/2): \quad \square\square\square\,, \qquad \text{“}\mathbf{5}\text{” } (j=2): \quad \square\square\square\square\,, \qquad \dots. \tag{4.17}$$

The numbers in bold are the dimensions of the respective representations, while the "quantum-number" $j = 0, 1/2, 1, 3/2, \dots$ is related to the dimensions as $d = 2j + 1$.

Let us consider a field $\Psi$ transforming under the $(\mathbf{2j+1})$-dimensional irreducible representation of SU(2). Without loss of generality, we can take the possible states of $\Psi$ to be the eigenvectors of the generator $T_3$, labelled by $j$ and the $T_3$-eigenvalue $m$, i.e. $|j, m\rangle$ with

$$T_3\,|j, m\rangle = \hbar m\,|j, m\rangle\,. \tag{4.18}$$

Note that $T_3(T_\pm\,|j, m\rangle) = \hbar(m \pm 1)T_\pm\,|j, m\rangle$. It implies

$$T_\pm\,|j, m\rangle = \hbar\lambda_{j,m}^\pm\,|j, m \pm 1\rangle\,, \qquad T^2\,|j, m\rangle = \hbar^2\lambda(\lambda+1)\,|j, m\rangle\,, \tag{4.19}$$

for some constants $\lambda_{j,m}^\pm$ and $\lambda$. Using the normalisation of states $\langle j, m|j, m\rangle = 1$, and relations $[T_+, T_-] = 2\hbar\, T_3$ and $\{T_+, T_-\} = 2T^2 - 2T_3^2$, we get

$$\lambda_{j,m}^-\lambda_{j,m-1}^+ = \lambda(\lambda+1) - m(m-1), \qquad |\lambda_{j,m}^+|^2 - |\lambda_{j,m}^-|^2 = -2m. \tag{4.20}$$

Since the dimension of the Hilbert space is finite, there must exist a state $|j, m_h\rangle$ such that $T_+\,|j, m_h\rangle = 0$, and similarly $|j, m_\ell\rangle$ such that $T_-\,|j, m_\ell\rangle = 0$. Correspondingly, we find $\lambda_{j,m_h}^+ = \lambda_{j,m_\ell}^- = 0$, leading to $m_h = \lambda$ and $m_\ell = -\lambda$. Since the dimension of the Hilbert space is $2j + 1$, we must have that $(T_+)^{2j}\,|j, m_h\rangle \propto |j, m_\ell\rangle$, leading to $m_h - m_\ell = 2j$ or

$$\lambda = j. \tag{4.21}$$

Finally, we can solve the recursion relations for $\lambda_{j,m}^\pm$ to get

$$\lambda_{j,m}^\pm = \sqrt{j(j+1) - m(m \pm 1)}. \tag{4.22}$$

In summary, the states in a generic $(\mathbf{2j+1})$-dimensional representation of SU(2), taken to be eigenstates of $T_3$, can be labelled by two quantum numbers $|j, m\rangle$ where $j = 0, 1/2, 1, 3/2, \dots$ and

$$m = -j, -j+1, \dots, j-1, j. \tag{4.23}$$

The quantum numbers $j$ and $m$ are related to the eigenvalues of $T^2$ and $T_3$ according to $T^2 |j, m\rangle = \hbar^2 j(j+1) |j, m\rangle$ and $T_3 |j, m\rangle = \hbar m |j, m\rangle$. The states can be arranged into a line-segment

$$T_- \longleftarrow \quad\quad\quad\quad\quad\quad\quad \longrightarrow T_+$$

$$\bullet \quad\quad \bullet \quad\quad \bullet \quad \text{-----} \quad \bullet \quad\quad \bullet \quad\quad \bullet$$

$$m = -j \quad\quad\quad\quad\quad\quad\quad\quad\quad m = j$$

There are a total of $2j + 1$ states. The ladder operators $T_\pm$ take us to the state immediately to the right/left in the arrangement above.

**Example:** The angular momentum operators $T_i = J_i$ in quantum mechanics furnish an $\mathfrak{su}(2)$ algebra. Quantum particles transform in an irreducible representation of this SU(2), labelled by the quantum number $j$ associated with the total angular momentum operator $T^2 = J^2$. Independent states of a quantum particle are labelled by the quantum number $m$ representing the $z$-component $T_3 = J_3$ of angular-momentum.

**Example:** The weak force in the Standard Model is associated with an internal SU(2) symmetry. The left-handed quarks and leptons, and the Standard Model Higgs boson transform as weak SU(2) doublets "**2**" ($j = 1/2$), while the weak force gauge bosons and the electromagnetic photon collectively transform as a weak SU(2) triplet "**3**" ($j = 1$) and a singlet "**1**" ($j = 0$). The quantum number $m$ is associated with the electric charge $q = m \pm 1/6$ for left-handed (anti)quarks and $q = m \mp 1/2$ for left-handed (anti)leptons. See sections 6.5 and 7.

**Example:** The Standard Model has an approximate SU(2) isospin symmetry. The light quarks $(u, d)$ and anti-quarks $(\bar{u}, \bar{d})$ make up SU(2) isospin doublets "**2**" ($j = 1/2$). Various hadrons appear as SU(2) isospin singlets "**1**" ($j = 0$; e.g. $\Lambda^0$ baryon and $\eta^0$ meson), doublets "**2**" ($j = 1/2$; e.g. $(p, n)$ baryons and $(K^0, K^+)$ mesons), triplets "**3**" ($j = 1$; e.g. $(\Sigma^-, \Sigma^0, \Sigma^+)$ baryons and $(\pi^-, \pi^0, \pi^+)$ mesons), or quadruplets "**4**" ($j = 3/2$; e.g. $(\Delta^-, \Delta^0, \Delta^+, \Delta^{2+})$ baryons). More details on the classification can be found in the next subsection during our discussion of the SU(3) flavour group.

## 4.4 SU(3)

SU(3) is next simplest special unitary group. It is the group of all $3 \times 3$ unit determinant unitary matrices. The eight generators of the associated Lie algebra $\mathfrak{su}(3)$, i.e. the set of all $3 \times 3$ traceless Hermitian matrices, are given by $T_a$ written out in eq. (2.21). To classify the spectrum of states, it is instead convenient to work with the complexified basis

$$T_\pm = T_1 \pm iT_2, \quad\quad T_3, \quad\quad V_\pm = T_4 \pm iT_5, \quad\quad U_\pm = T_6 \pm iT_7, \quad\quad Y = \frac{2}{\sqrt{3}} T_8. \quad\quad (4.24)$$

These have commutation relations

$$[T_+, T_-] = 2\hbar T_3, \quad\quad [V_+, V_-] = \frac{3\hbar}{2} Y + \hbar T_3, \quad\quad [U_+, U_-] = \frac{3\hbar}{2} Y - \hbar T_3,$$

$$[T_\pm, V_\pm] = 0, \quad\quad [T_\pm, V_\mp] = \mp \hbar U_\mp, \quad\quad [T_\pm, U_\pm] = \pm \hbar V_\pm, \quad\quad [T_\pm, U_\mp] = 0,$$

$$[U_\pm, V_\pm] = 0, \quad\quad [U_\pm, V_\mp,] = \pm \hbar T_\mp, \quad\quad (4.25)$$

along with

$$[T_3, T_\pm] = \pm\hbar T_\pm, \qquad [T_3, V_\pm] = \pm\frac{\hbar}{2}V_\pm, \qquad [T_3, U_\pm] = \mp\frac{\hbar}{2}U_\pm,$$

$$[Y, T_\pm] = 0, \qquad [Y, T_3] = 0, \qquad [Y, V_\pm] = \pm\hbar V_\pm \qquad [Y, U_\pm] = \pm\hbar U_\pm. \qquad (4.26)$$

The Casimir operators are given as $T^2 = T_a T_a$ and $T^3 = d_{abc} T_a T_b T_c$, where $d_{abc} = 2/\hbar^3 \, \text{tr}(\{T_a, T_b\}T_c)$ is the totally symmetric structure constant of $\mathfrak{su}(3)$. These Casimir operators commute with all the generators of the $\mathfrak{su}(3)$ algebra $[T^2, T_a] = [T^3, T_a] = 0$.

The generators $T_\pm$, $T_3$ span a $\mathfrak{su}(2)$ subalgebra of $\mathfrak{su}(3)$, known as the *isospin* subalgebra. However, the total isospin operator $I^2 = T_1^2 + T_2^2 + T_3^2 = \frac{1}{2}\{T_+, T_-\} + T_3^2$ is not a Casimir of the full $\mathfrak{su}(3)$ algebra; to wit

$$[I^2, T_\pm] = 0, \qquad [I^2, T_3] = 0, \qquad [I^2, Y] = 0,$$

$$[I^2, V_\pm] = \pm\frac{\hbar}{2}\{V_\pm, T_3\} \pm \frac{\hbar}{2}\{U_\pm, T_\pm\}, \qquad [I^2, U_\pm] = \mp\frac{\hbar}{2}\{U_\pm, T_3\} \pm \frac{\hbar}{2}\{V_\pm, T_\mp\}. \qquad (4.27)$$

It will, nonetheless, be useful in the classification of the spectrum of states. Similar $\mathfrak{su}(2)$ subalgebras are also spanned by the generators $V_\pm$, $3Y/4 + T_3/2$ and by $U_\pm$, $3Y/4 - T_3/2$.

The irreducible representations of SU(3) can be labelled by two integers $p$ and $q$. In terms of Young tableaux, the representation $D(p, q)$ is one with $p$ number of one-box columns and $q$ number of two-box columns. The representations can be arranged in a 2-dimensional array

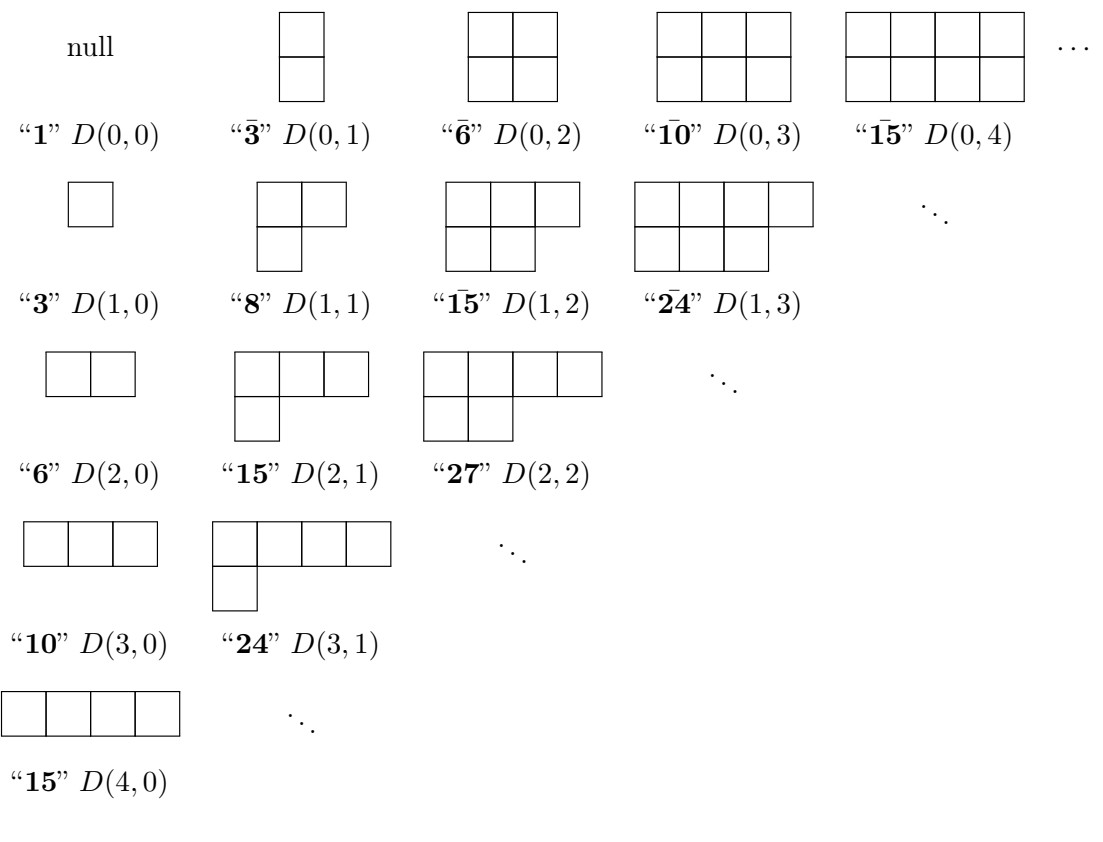

Diagonally opposite representations $D(p,q)$ and $D(q,p)$ are conjugates to one another, while the diagonal representations $D(p,p)$ are self-conjugates. We have noted the dimensions of the representations in bold. Note that, unlike SU(2), the dimensions do not uniquely characterise a representation. It is conventional to denote the dimensions of conjugate representations $D(p,q)$ with $q > p$ with a bar. With the barred notation in place, the dimensions do uniquely characterise the first $4 \times 4$ block of the representations above. These are typically all we need in the Standard Model. The $T^2$ and $T^3$ eigenvalues of the representation $D(p,q)$ are given as $\hbar^2 c_2(p,q)$ and $\hbar^3 c_3(p,q)$ where

$$c_2(p,q) = \frac{1}{3}\left(p^2 + q^2 + 3p + 3q + pq\right), \qquad c_3(p,q) = \frac{1}{18}(p-q)(3+p+2q)(3+q+2p), \quad (4.28)$$

whereas the dimension of the representation is given as

$$\dim D(p,q) = \frac{1}{2}(p+1)(q+1)(p+q+2). \tag{4.29}$$

Let us consider a field $\Psi$ transforming in the $D(p,q)$ representation of SU(3). The independent states of $\Psi$ can be labelled by the eigenvalues of the mutually commuting set of operators $T^2$, $T^3$, $I^2$, $T_3$, and $Y$, i.e. $|p,q;j,m,y\rangle$. Note that the eigenvalues of the Casimir operators $T^2$ and $T^3$ is already fixed in terms of $p$, $q$. To wit, we have

$$T^2 |p,q;j,m,y\rangle = \hbar^2 c_2(p,q) |p,q;j,m,y\rangle, \qquad T^3 |p,q;j,m,y\rangle = \hbar^3 c_3(p,q) |p,q;j,m,y\rangle,$$
$$T_3 |p,q;j,m,y\rangle = \hbar m |p,q;j,m,y\rangle, \qquad Y |p,q;j,m,y\rangle = \hbar y |p,q;j,m,y\rangle,$$
$$I^2 |p,q;j,m,y\rangle = \hbar^2 j(j+1) |p,q;j,m,y\rangle. \tag{4.30}$$

The action of the remaining $\mathfrak{su}(3)$ operators can be derived using the Lie algebra commutation relations, leading to

$$T_\pm |p,q;j,m,y\rangle = \hbar\sqrt{j(j+1) - m(m \pm 1)} \,|p,q;j,m \pm 1,y\rangle$$

$$U_\pm |p,q;j,m,y\rangle = \hbar\sqrt{\frac{(j \mp m + 1)f_\pm(j \pm \frac{y}{2} + 1)}{(j+1)(2j+1)}} \,|p,q;j+\tfrac{1}{2},m \mp \tfrac{1}{2},y \pm 1\rangle$$

$$+ \hbar\sqrt{\frac{(j \pm m)f_\mp(j \mp \frac{y}{2})}{j(2j+1)}} \,|p,q;j-\tfrac{1}{2},m \mp \tfrac{1}{2},y \pm 1\rangle$$

$$V_\pm |p,q;j,m,y\rangle = \pm\hbar\sqrt{\frac{(j \pm m + 1)f_\pm(j \pm \frac{y}{2} + 1)}{(j+1)(2j+1)}} \,|p,q;j+\tfrac{1}{2},m \pm \tfrac{1}{2},y \pm 1\rangle$$

$$\mp \hbar\sqrt{\frac{(j \mp m)f_\mp(j \mp \frac{y}{2})}{j(2j+1)}} \,|p,q;j-\tfrac{1}{2},m \pm \tfrac{1}{2},y \pm 1\rangle, \tag{4.31}$$

where $f_\pm(x) = (c_2 + 1 - x^2)x/2 \pm c_3/3$. These expressions are considerably more complicated than the SU(2) case. Nonetheless, to write down the spectrum of states, we can start with the "highest-weight" state that is annihilated by all the "creation" operators $T_+$, $U_+$, $V_+$, i.e.

$$T_+ |p,q;j_h,m_h,y_h\rangle = U_+ |p,q;j_h,m_h,y_h\rangle = V_+ |p,q;j_h,m_h,y_h\rangle = 0. \tag{4.32}$$

Using eq. (4.31), it is straight-forward to see that this state has $j_h = m_h = p/2$ and $y_h = p/3 + 2q/3$. All the remaining states in the representation can be obtained by repeatedly applying the "annihilation"

operators $T_-$, $U_-$, $V_-$ on the highest weight state. It is convenient to define the quantum number $\ell = (y - y_h)/2 - j + j_h$. In terms of this, we have the complete spectrum of states

$$\ell = 0, 1, \ldots, q,$$
$$2j = \ell, \ell + 1, \ldots, \ell + p,$$
$$m = -j, -j + 1, \ldots, j - 1, j. \tag{4.33}$$

It can be checked that the total number of states add up to $\dim D(p, q)$ given in eq. (4.29). These states can be neatly arranged in the $m - y$ plane into hexagonal or triangular patterns, known as the eightfold-way diagrams proposed by Murray Gell-Mann in 1961 [12]. For example, the eightfold-way diagram representing the states of the representation $D(p, q)$, assuming $p > q$, is given as

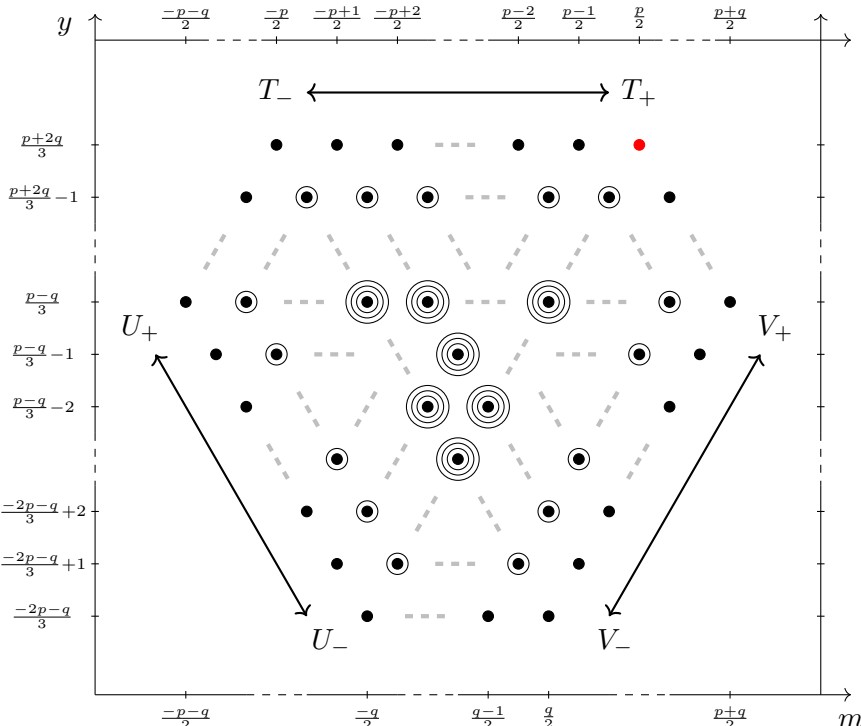

The diagram is a non-regular hexagon with $p + 1$ and $q + 1$ number of states on its alternate sides starting from the top. The overlapping circles denote the multiplicity of states in the total isospin quantum number $j$. The states lying on the outermost hexagon are non-degenerate, those lying on the hexagon immediately inside are doubly degenerate, and so on until we hit a triangle in the center. All the states on and inside this triangle have degeneracy $\min(p, q) + 1$. The highest weight state is denoted in red. The ladder operators $T_\pm$ take us horizontally to a state lying immediately to the right/left, $V_\pm$ diagonally to a linear combination of the states lying immediately to the top-right/bottom-left, while $U_\pm$ diagonally to a linear combination of the states lying immediately to the top-left/bottom-right. The eightfold-way diagram for $D(q, p)$ has the same form, but reflected in the $y$-direction $y \to -y$.

For instance, the states in the "trivial/singlet" "**1**" $D(0, 0)$, "fundamental/triplet" "**3**" $D(1, 0)$, "sextet" "**6**" $D(2, 0)$, "decuplet" "**10**" $D(3, 0)$, and "**15**" $D(4, 0)$ representations, and so on, are

respectively arranged as

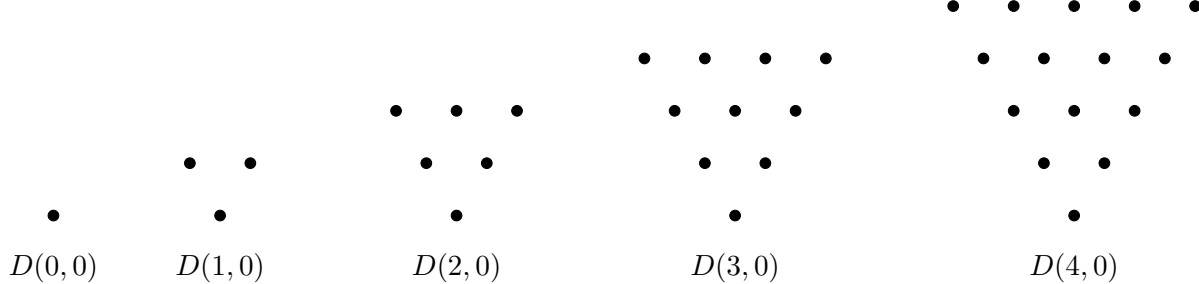

$$D(0,0) \qquad D(1,0) \qquad D(2,0) \qquad D(3,0) \qquad D(4,0)$$

while the states in the conjugate "anti-fundamental/anti-triplet" "$\bar{\mathbf{3}}$" $D(0,1)$, "anti-sextet" "$\bar{\mathbf{6}}$" $D(0,2)$, "anti-decuplet" "$\bar{\mathbf{10}}$" $D(0,3)$, and "$\bar{\mathbf{15}}$" $D(0,4)$ representations, and so on, are arranged as

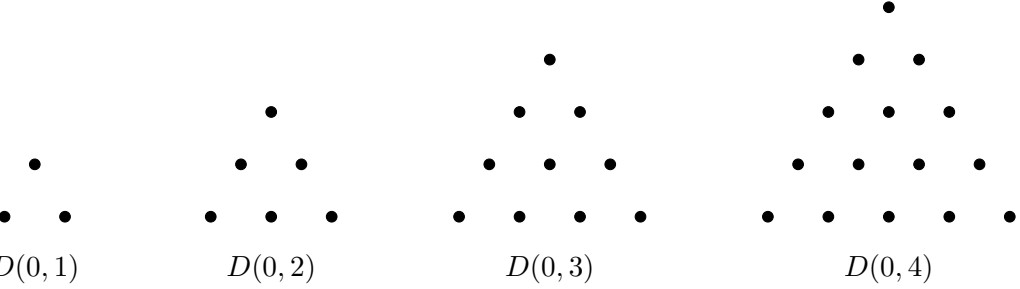

$$D(0,1) \qquad D(0,2) \qquad D(0,3) \qquad D(0,4)$$

The eightfold-way diagrams for the more non-trivial "adjoint/octet" "$\mathbf{8}$" $D(1,1)$, "$\mathbf{15}$" $D(2,1)$, "$\mathbf{24}$" $D(3,1)$, "$\bar{\mathbf{15}}$" $D(1,2)$, "$\bar{\mathbf{24}}$" $D(1,3)$, and "$\mathbf{27}$" $D(2,2)$ representations are given as

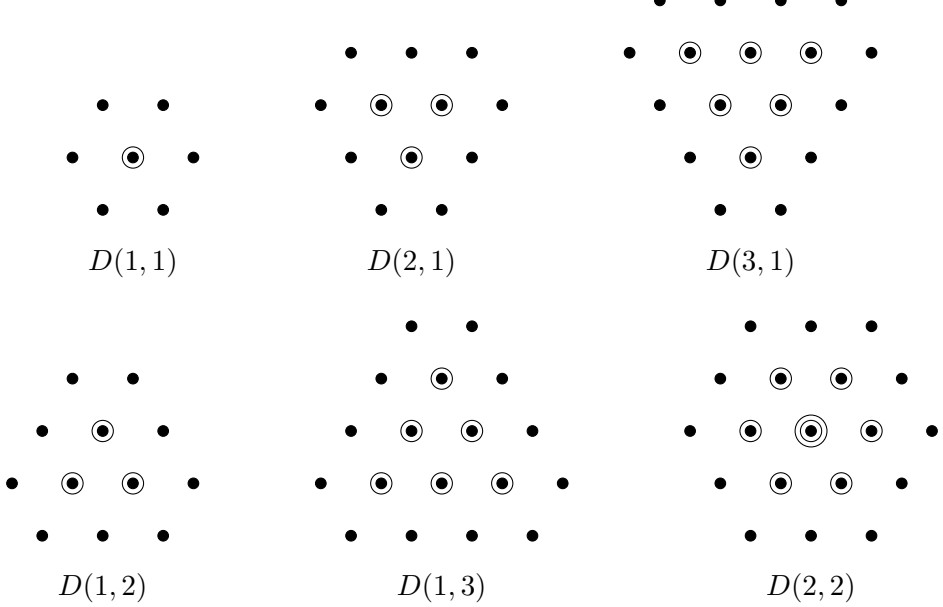

$$D(1,1) \qquad\qquad D(2,1) \qquad\qquad D(3,1)$$

$$D(1,2) \qquad\qquad D(1,3) \qquad\qquad D(2,2)$$

**Example:**   The QCD colour force in the Standard Model is associated with an internal SU(3) symmetry. The quarks and anti-quarks transform in the triplet "$\mathbf{3}$" $D(1,0)$ and anti-triplet "$\bar{\mathbf{3}}$"

$D(0,1)$ representations respectively of the colour SU(3) group, whereas the gluons transform in the adjoint "$\mathbf{8}$" $D(1,1)$ representation. The remaining Standard Model fundamental particles, as well as the naturally appearing hadrons, are colour SU(3) singlets. See sections 6.3 and 7.

**Example:** The Standard Model has an approximate SU(3) flavour symmetry, which has the isospin SU(2) symmetry as a subgroup. The light quarks $(u, d, s)$ and antiquarks $(\bar{u}, \bar{d}, \bar{s})$ make up the triplet "$\mathbf{3}$" $D(1,0)$ and anti-triplet "$\bar{\mathbf{3}}$" $D(0,1)$ representations of the flavour SU(3) group:

$$\bullet d \quad \bullet u \qquad\qquad\qquad \bullet \bar{s}$$

$$\bullet s \qquad\qquad\qquad \bullet \bar{u} \quad \bullet \bar{d}$$

quark triplet           antiquark anti-triplet

The baryons are made of 3 quarks and transform in the octet "$\mathbf{8}$" $D(1,1)$ or decuplet "$\mathbf{10}$" $D(3,0)$ representations of the flavour SU(3) group, while the anti-baryons are made of 3 anti-quarks and transform in the respective octet "$\mathbf{8}$" $D(1,1)$ or anti-decuplet "$\bar{\mathbf{10}}$" $D(0,3)$ representations:

$$\bullet \Delta^- \; \bullet \Delta^0 \; \bullet \Delta^+ \; \bullet \Delta^{2+} \qquad\qquad\qquad \bullet \bar{\Omega}^+$$

$$\bullet n^0 \quad \bullet p^+ \qquad \bullet \Sigma^{*-} \; \bullet \Sigma^{*0} \; \bullet \Sigma^{*+} \qquad \bullet \bar{\Xi}^0 \; \bullet \bar{\Xi}^+ \qquad \bullet \bar{\Xi}^{*0} \; \bullet \bar{\Xi}^{*+}$$

$$\Lambda^0 \qquad\qquad\qquad\qquad \bar{\Lambda}^0$$

$$\bullet \Sigma^- \; \circledcirc \Sigma^0 \; \bullet \Sigma^+ \qquad \bullet \Xi^{*-} \; \bullet \Xi^{*0} \qquad \bullet \bar{\Sigma}^- \; \circledcirc \bar{\Sigma}^0 \; \bullet \bar{\Sigma}^+ \qquad \bullet \bar{\Sigma}^{*-} \; \bullet \bar{\Sigma}^{*0} \; \bullet \bar{\Sigma}^{*+}$$

$$\bullet \Xi^- \; \bullet \Xi^0 \qquad \bullet \Omega^- \qquad \bullet \bar{p}^- \; \bullet \bar{n}^0 \qquad \bullet \bar{\Delta}^{2-} \bullet \bar{\Delta}^- \; \bullet \bar{\Delta}^0 \; \bullet \bar{\Delta}^+$$

baryon octet     baryon decuplet     anti-baryon octet     anti-baryon anti-decuplet

This follows because the tensor decomposition of 3-quark states is $\mathbf{3} \otimes \mathbf{3} \otimes \mathbf{3} = \mathbf{10} \oplus \mathbf{8} \oplus \mathbf{8} \oplus \mathbf{1}$ and that of 3-antiquark states is $\bar{\mathbf{3}} \otimes \bar{\mathbf{3}} \otimes \bar{\mathbf{3}} = \bar{\mathbf{10}} \oplus \mathbf{8} \oplus \mathbf{8} \oplus \mathbf{1}$. Similarly, mesons are made of a quark and an antiquark and transform in the singlet "$\mathbf{1}$" $D(0,0)$ or octet "$\mathbf{8}$" $D(1,1)$ representations of the SU(3) flavour group:

$$\bullet K^0 \quad \bullet K^+$$

$$\eta^0$$

$$\bullet \pi^- \; \circledcirc \pi^0 \; \bullet \pi^+$$

$$\bullet \eta'^0 \qquad\qquad \bullet \bar{K}^- \; \bullet \bar{K}^0$$

meson singlet           meson octet

This is due to the the tensor decomposition of quark-antiquark states $\mathbf{3} \otimes \bar{\mathbf{3}} = \mathbf{8} \oplus \mathbf{1}$. The states appearing in the same horizontal line in the diagrams above make up isospin multiplets. Note that the electric charge of a state is given as $q = m + y/2$. Note also that these diagrams only cover light baryons and mesons, made up of the 3 light quarks $(u, d, s)$ and the respective anti-quarks. The complete set of baryons and mesons, made up of the 6 quarks $(u, d, c, s, t, b)$ and the respective anti-quarks, can be similarly classified using the representations of SU(6).

# 5 | Lorentz and Poincaré groups

Physical theories are required to be invariant under a variety of spacetime symmetries corresponding to the change of inertial reference frame observing the system under consideration. These include spacetime translations, rotations, and boosts, formally making up the Poincaré group. The subgroup of all rotations and boosts is known as the Lorentz group. In this section, we dig into a detailed discussion of the representation theory of Lorentz and Poincaré groups. This also allows us to opportunity to introduce spinor fields as a representation of the Lorentz group, along with other crucial concepts such as chirality, helicity, and spin. Since most of the matter fields occurring in nature are spinorial, these concepts will prove quintessential in setting up the Standard Model of particle physics. The discussion in this section heavily relies on Hugh Osborn's notes on group theory [13].

## 5.1 Lorentz group

Consider a physical theory defined on our $(3 + 1)$-dimensional spacetime manifold with coordinates $x^\mu = (ct, x, y, z)$. The Lorentz transformations form a matrix group given by the coordinate transformations

$$x^\mu \to \Lambda^\mu{}_\nu x^\nu, \tag{5.1}$$

such that the proper distance between a spacetime point and origin remains invariant[3]

$$x^\mu x^\nu \eta_{\mu\nu} \to x^\mu x^\nu \eta_{\mu\nu}, \qquad \eta_{\mu\nu} = \mathrm{diag}(-1, 1, 1, 1). \tag{5.2}$$

Suppressing the indices, we get the Lorentz group O(3,1) as all $4 \times 4$ matrices $\Lambda$ satisfying

$$\Lambda^\mathrm{T} \eta \Lambda = \eta. \tag{5.3}$$

This is a generalisation of the orthogonal group O(4) defined as all $4 \times 4$ matrices $M$ satisfying the relation $M^\mathrm{T} \mathbb{1} M = \mathbb{1}$.

Similar to SO(4), the *proper Lorentz group* SO(3, 1) contains all the orientation preserving Lorentz transformations, defined as all $4 \times 4$ matrices $\Lambda$ satisfying $\Lambda^\mathrm{T} \eta \Lambda = \eta$ and $\det \Lambda = 1$. This excludes the parity operation P : $x^i \to -x^i$ and the time-reversal operation T : $x^0 \to -x^0$, but includes the spacetime parity operation PT : $x^\mu \to -x^\mu$. The simply connected piece of O(3,1) or SO(3,1) further requires

$$\Lambda^\mathrm{T} \eta \Lambda = \eta, \qquad \det \Lambda = 1, \qquad \Lambda^0{}_0 > 0, \tag{5.4}$$

known as SO$^+$(3, 1) or *proper orthochronous Lorentz group*, which excludes all P, T, and PT.

**Lorentz generators:** Let us focus on the connected piece SO$^+$(3, 1). Writing in the exponential parametrisation, we can express an arbitrary element of $\Lambda \in$ SO$^+$(3, 1) in terms of arbitrary parameters $\omega^{\mu\nu} = -\omega^{\nu\mu}$ and generators $M_{\mu\nu} = -M_{\nu\mu}$ spanning the associated Lie algebra $\mathfrak{so}(3, 1)$, leading to

$$\Lambda = \exp\left(\frac{i}{2\hbar} \omega^{\mu\nu} M_{\mu\nu}\right). \tag{5.5}$$

---

[3]We are using the mostly positive convention for the Minkowskian metric, as opposed to the mostly negative convention where $\eta_{\mu\nu} = \mathrm{diag}(1, -1, -1, -1)$.

Here $M_{\mu\nu}$ are six linearly independent $4 \times 4$ matrices satisfying

$$(M_{\mu\nu})^{\mathrm{T}} = -\eta M_{\mu\nu}\eta. \tag{5.6}$$

We can take the generators to be: the rotation generators

$$J_1 \equiv M_{23} = -M_{32} = \hbar \begin{pmatrix} 0 & 0 & 0 & 0 \\ 0 & 0 & 0 & 0 \\ 0 & 0 & 0 & -i \\ 0 & 0 & i & 0 \end{pmatrix}, \qquad J_2 \equiv M_{31} = -M_{13} = \hbar \begin{pmatrix} 0 & 0 & 0 & 0 \\ 0 & 0 & 0 & i \\ 0 & 0 & 0 & 0 \\ 0 & -i & 0 & 0 \end{pmatrix},$$

$$J_3 \equiv M_{12} = -M_{21} = \hbar \begin{pmatrix} 0 & 0 & 0 & 0 \\ 0 & 0 & -i & 0 \\ 0 & i & 0 & 0 \\ 0 & 0 & 0 & 0 \end{pmatrix}, \tag{5.7}$$

and the boost generators

$$K_1 \equiv M_{10} = -M_{01} = \hbar \begin{pmatrix} 0 & i & 0 & 0 \\ i & 0 & 0 & 0 \\ 0 & 0 & 0 & 0 \\ 0 & 0 & 0 & 0 \end{pmatrix}, \qquad K_2 \equiv M_{20} = -M_{02} = \hbar \begin{pmatrix} 0 & 0 & i & 0 \\ 0 & 0 & 0 & 0 \\ i & 0 & 0 & 0 \\ 0 & 0 & 0 & 0 \end{pmatrix},$$

$$K_3 \equiv M_{30} = -M_{03} = \hbar \begin{pmatrix} 0 & 0 & 0 & i \\ 0 & 0 & 0 & 0 \\ 0 & 0 & 0 & 0 \\ i & 0 & 0 & 0 \end{pmatrix}, \tag{5.8}$$

satisfying the commutation relations

$$[J_i, J_j] = i\hbar\epsilon_{ijk}J_k, \qquad [K_i, K_j] = -i\hbar\epsilon_{ijk}J_k, \qquad [J_i, K_j] = i\hbar\epsilon_{ijk}K_k. \tag{5.9}$$

In terms of $M_{\mu\nu}$, these can be covariantly represented as

$$[M_{\mu\nu}, M_{\rho\sigma}] = i\hbar \left( \eta_{\mu\rho}M_{\nu\sigma} - \eta_{\nu\rho}M_{\mu\sigma} - \eta_{\mu\sigma}M_{\nu\rho} + \eta_{\nu\sigma}M_{\mu\rho} \right). \tag{5.10}$$

Note that

$$(M_{\mu\nu})^\rho{}_\sigma = -i\hbar \left( \delta^\rho_\mu \eta_{\nu\sigma} - \delta^\rho_\nu \eta_{\mu\sigma} \right) \qquad \Longrightarrow \qquad \Lambda^\mu{}_\nu = \exp(\omega)^\mu{}_\nu. \tag{5.11}$$

**Double cover:** The connected piece of the Lorentz group $\mathrm{SO}^+(3,1)$ can be mapped to the group $\mathrm{SL}(2,\mathbb{C})$ of all $2 \times 2$ unit determinant complex matrices. Consider rewriting the coordinates $x^\mu$ into a $2 \times 2$ matrix

$$X = \sigma_\mu x^\mu = \begin{pmatrix} x^0 + x^3 & x^1 - ix^2 \\ x^1 + ix^2 & x^0 - x^3 \end{pmatrix}, \qquad X^\dagger = X, \qquad \det X = -x^\mu x^\nu \eta_{\mu\nu}, \tag{5.12}$$

where $\sigma_\mu = (\mathbb{1}, \sigma_i)$, with $\sigma_i$ being the Pauli matrices. Raising/lowering of $\mu, \nu, \ldots$ indices is done by $\eta^{\mu\nu}$ and $\eta_{\mu\nu}$ respectively. A generic $\mathrm{SO}^+(3,1)$ transformation preserves the Hermiticity and determinant of $X$, leading to an $A \in \mathrm{SL}(2,\mathbb{C})$ transformation[4]

$$x^\mu \to \Lambda^\mu{}_\nu x^\nu \implies X \to AXA^\dagger, \qquad \det A = 1. \tag{5.14}$$

Note that an arbitrary $\mathrm{SL}(2,\mathbb{C})$ matrix also has 6 independent real components, similar to Lorentz transformations. Note also that two distinct elements $\pm A \in \mathrm{SL}(2,\mathbb{C})$ correspond to the same Lorentz transformation $\Lambda \in \mathrm{SO}^+(3,1)$. Hence, the group $\mathrm{SL}(2,\mathbb{C})$ is often referred to as the double-cover of the proper orthochronous Lorentz group $\mathrm{SO}^+(3,1)$. This should be contrasted with the double cover $\mathrm{SU}(2)$ of the rotation group $\mathrm{SO}(3)$ we discussed around eq. (2.35).

The Lie algebra $\mathfrak{sl}(2,\mathbb{C})$ of $\mathrm{SL}(2,\mathbb{C})$ is the set of all $2 \times 2$ complex traceless matrices. The respective generators can be taken to be the traceless Hermitian matrices $\tilde{J}_i = \hbar/2\, \sigma_i$ and the traceless anti-Hermitian matrices $\tilde{K}_i = \pm i\hbar/2\, \sigma_i$ with the same commutation relations as eq. (5.9) for either of the signs in $\tilde{K}_i$. Defining

$$X_i^\pm = \frac{1}{2}(\tilde{J}_i \pm i\tilde{K}_i), \tag{5.15}$$

the commutation relations reduce to

$$[X_i^+, X_j^+] = i\hbar\epsilon_{ijk}X_k^+, \qquad [X_i^-, X_j^-] = i\hbar\epsilon_{ijk}X_k^-, \qquad [X_i^+, X_j^-] = 0. \tag{5.16}$$

Note that the generators $X_i^+$ and $X_i^-$ independently span two copies of $\mathfrak{su}(2)$ algebras. It follows that $\mathfrak{sl}(2,\mathbb{C}) \cong \mathfrak{su}(2) \oplus \mathfrak{su}(2)$ or correspondingly $\mathrm{SL}(2,\mathbb{C}) \cong \mathrm{SU}(2) \times \mathrm{SU}(2)$. Taking the sign in $\tilde{K}_i$ to be $+$ or $-$, one sees that either $X_i^+$ or $X_i^-$ is identically zero. This is because $\tilde{K}_i = \pm i\tilde{J}_i$. However, since this property does not need to be satisfied by an arbitrary matrix representation $D$ of $\mathfrak{sl}(2,\mathbb{C})$, both $D(X_i^\pm)$ will be generically nonzero. The exponential parametrisation of $\Lambda \in \mathrm{SO}^+(3,1)$ in eq. (5.5), in this language, leads to[5]

$$\Lambda = \exp\left(\frac{i}{\hbar}\alpha^i J_i + \frac{i}{\hbar}\beta^i K_i\right) \to \exp\left(\frac{i}{\hbar}\omega_+^i X_i^+ + \frac{i}{\hbar}\omega_-^i X_i^-\right) = \exp\left(\frac{i}{\hbar}\omega_+^i X_i^+\right)\exp\left(\frac{i}{\hbar}\omega_-^i X_i^-\right), \tag{5.17}$$

where $\alpha^i = 1/2\, \epsilon^{ijk}\omega_{jk}$ are the rotation parameters and $\beta^i = \omega^{i0}$ are the boost parameters, while $\omega_\pm^i = \alpha^i \mp i\beta^i$ are the parameters of the two copies of $\mathrm{SU}(2)$ respectively. Note that $\omega_\mp^i = (\omega_\pm^i)^*$. The final step above is justified because $[X_i^+, X_j^-] = 0$.

## 5.2 Lorentz representations

A representation $D$ of $\mathrm{SO}^+(3,1)$ is a mapping onto the set of matrices such that

$$D(\mathbb{1}) = \mathbb{1}, \qquad D(\Lambda)D(\Lambda') = D(\Lambda\Lambda'), \tag{5.18}$$

---

[4]The explicit mapping is given by

$$\Lambda^\mu{}_\nu \to A = \mathrm{e}^{i\alpha}\frac{\sigma_\mu\Lambda^\mu{}_\nu\bar{\sigma}^\nu}{2\sqrt{\Lambda^\mu{}_\mu}}, \tag{5.13}$$

where the phase $\mathrm{e}^{i\alpha}$ is determined up to $\pm 1$ by the condition $\det A = 1$.

[5]Note that $M_{ij} = \epsilon_{ijk}J_k$.

for all $\Lambda, \Lambda' \in \mathrm{SO}^+(3,1)$. A field $\Phi(x)$ is said to transform in the representation $D$ of $\mathrm{SO}^+(3,1)$, if $\Lambda \in \mathrm{SO}^+(3,1)$ acts as

$$\Phi(x) \to D(\Lambda)\Phi(\Lambda^{-1}x). \tag{5.19}$$

Note that the symmetry also acts on the coordinate arguments of the field, justifying it to be a "local" spacetime symmetry. Using the exponential parametrisation of $\Lambda \in \mathrm{SO}^+(3,1)$ given in eq. (5.5), the representation $D(\Lambda)$ can be written in terms of the representations of the Lie algebra generators $D(M_{\mu\nu})$, i.e.

$$D(\Lambda) = \exp\left(\frac{i}{2\hbar}\omega^{\mu\nu}D(M_{\mu\nu})\right). \tag{5.20}$$

A representation $D$ of $\mathrm{SO}^+(3,1)$ can also be mapped onto a direct product representation $D_+ \times D_-$ of $\mathrm{SL}(2,\mathbb{C}) \cong \mathrm{SU}(2) \times \mathrm{SU}(2)$, with

$$D^\pm(\Lambda) = \exp\left(\frac{i}{\hbar}\omega_\pm^i D(X_i^\pm)\right), \qquad D(X_i^\pm) = \frac{1}{2}\left(D(J_i) \pm iD(K_i)\right). \tag{5.21}$$

A field $\Phi(x)$, in this representation, transforms as

$$\Phi(x) \to (D^+(\Lambda) \otimes D^-(\Lambda))\Phi(\Lambda^{-1}x). \tag{5.22}$$

The irreducible representations of (the double cover of) $\mathrm{SO}^+(3,1)$ can hence be labelled by the "highest weights" $(j_-, j_+)$ with $j_\pm = 0, 1/2, 1, 3/2, \ldots$ corresponding to the two copies of $\mathrm{SU}(2)$; see section 4.3. The states in an irreducible representation can respectively be labelled by the eigenvalues of $(\vec{X}^\pm)^2$ and $X_3^\pm$ operators, i.e.

$$(\vec{X}^\pm)^2 |j_-, j_+; m_-, m_+\rangle = \hbar^2 j_\pm(j_\pm + 1)|j_-, j_+; m_-, m_+\rangle,$$
$$X_3^\pm |j_-, j_+; m_-, m_+\rangle = \hbar m_\pm |j_-, j_+; m_-, m_+\rangle. \tag{5.23}$$

Few interesting cases of fields transforming under such representations are

- $(0,0)$: scalar fields.
- $(1/2, 0)$, $(0, 1/2)$: left-handed and right-handed Weyl spinor fields.
- $(1/2, 1/2)$: vector fields.

We will look at these in some detail below. It is also convenient to define the *chirality operator*

$$\Gamma = (\vec{X}^+)^2 - (\vec{X}^+)^2 = i\vec{K} \cdot \vec{J} = \frac{i}{8}\epsilon^{\mu\nu\rho\sigma}M_{\mu\nu}M_{\rho\sigma}. \tag{5.24}$$

By definition, $\Gamma$ commutes with all the generators of the Lorentz algebra, and has eigenvalue given by $\hbar^2 j_+(j_+ + 1) - \hbar^2 j_-(j_- + 1)$. For the three examples above, scalars and vectors have chirality $0$, while left-handed and right-handed Weyl spinors have chirality $-3\hbar/4$ and $+3\hbar/4$ respectively. In general, the fields with positive chirality are said to be right-handed (or right-chiral), while the fields with negative chirality are said to be left-handed (or left-chiral).

Given a Lorentz representation $(j_-, j_+)$, the conjugate representation is given by $(j_+, j_-)$. The states in the representation respectively map to $|j_-, j_+; m_-, m_+\rangle \to |j_+, j_-; -m_+, -m_-\rangle$, up to a phase factor.

### 5.2.1   Scalars

A complex field $\phi(x)$ is said to be a scalar if it transforms in the trivial $(0,0)$ representation of the double cover of the connected Lorentz group $\mathrm{SL}(2, \mathbb{C})$, i.e.

$$D_{(0,0)}(X_i^{\pm}) = 0, \qquad D_{(0,0)}(M_{\mu\nu}) = 0, \tag{5.25}$$

or equivalently

$$D_{(0,0)}^{+}(\Lambda) = D_{(0,0)}^{-}(\Lambda) = \mathbb{1}, \qquad D_{(0,0)}(\Lambda) = \mathbb{1}. \tag{5.26}$$

Correspondingly, the transformation rule of a scalar field $\phi(x)$ is given as

$$\phi(x) \to \phi(\Lambda^{-1}x). \tag{5.27}$$

### 5.2.2   Spinors

**Right-handed Weyl spinors:**   A right-handed Weyl spinor $\psi_\alpha(x)$, with $\alpha = 1, 2$, transforms in the $(0, 1/2)$ irreducible representation of $\mathrm{SL}(2, \mathbb{C})$. By definition, this representation is fundamental in the $X_i^+$ generators, while trivial in the $X_i^-$ generators, i.e.

$$D_{(0,1/2)}(X_i^+) = \frac{\hbar}{2}\sigma_i, \qquad D_{(0,1/2)}(X_i^-) = 0$$

$$\implies \quad D_{(0,1/2)}(M_{\mu\nu}) = s_{\mu\nu} = \frac{i\hbar}{2}\sigma_{[\mu}\bar\sigma_{\nu]} = \frac{i\hbar}{4}\left(\sigma_\mu\bar\sigma_\nu - \sigma_\nu\bar\sigma_\mu\right) \in \mathfrak{sl}(2, \mathbb{C}), \tag{5.28}$$

where $\sigma_\mu = (-\mathbb{1}, \sigma_i)$ and $\bar\sigma_\mu = (-\mathbb{1}, -\sigma_i)$. Raising the Lorentz indices, we get $\sigma^\mu = (\mathbb{1}, \sigma_i)$ and $\bar\sigma^\mu = (\mathbb{1}, -\sigma_i)$. This corresponds to $D_{(0,1/2)}(J_i) = \hbar/2\,\sigma_i$ and $D_{(0,1/2)}(K_i) = -i\hbar/2\,\sigma_i$. Note that each component of $\sigma_\mu$ and $\bar\sigma_\mu$ is a $2 \times 2$ complex matrix. It is customary to denote the components in terms of "undotted" and "dotted" indices

$$(\sigma_\mu)_{\alpha\dot\alpha}, \qquad (\bar\sigma_\mu)^{\dot\alpha\alpha}, \tag{5.29}$$

where both the kinds of indices run over $\alpha, \beta, \ldots, \dot\alpha, \dot\beta, \ldots = 1, 2$. In terms of these components, we can denote

$$(s_{\mu\nu})_\alpha{}^\beta = \frac{i\hbar}{4}\left((\sigma_\mu)_{\alpha\dot\alpha}(\bar\sigma_\nu)^{\dot\alpha\beta} - (\sigma_\nu)_{\alpha\dot\alpha}(\bar\sigma_\mu)^{\dot\alpha\beta}\right). \tag{5.30}$$

We can check that $s_{\mu\nu}$ satisfies the Lorentz algebra (5.10). Correspondingly, in the exponential parametrisation, we have

$$D_{(0,1/2)}(\Lambda) = A(\Lambda) = \exp\left(\frac{i}{2\hbar}\omega^{\mu\nu}s_{\mu\nu}\right) \in \mathrm{SL}(2, \mathbb{C}), \tag{5.31}$$

and the right-handed Weyl spinor $\psi_\alpha(x)$ transforms as

$$\psi_\alpha(x) \to A(\Lambda)_\alpha{}^\beta\psi_\beta(\Lambda^{-1}x). \tag{5.32}$$

We can also define the "anti-fundamental" right-handed Weyl spinor by raising the index with the Levi-Civita tensor $\varepsilon^{\alpha\beta}$, with $\varepsilon^{12} = 1$, i.e.

$$\psi^\alpha(x) = \varepsilon^{\alpha\beta}\psi_\beta(x), \qquad \psi^\alpha(x) \to \varepsilon^{\alpha\beta}A(\Lambda)_\beta{}^\gamma\varepsilon_{\gamma\delta}\psi^\delta(\Lambda^{-1}x) = \psi^\beta(\Lambda^{-1}x)(A(\Lambda)^{-1})_\beta{}^\alpha. \tag{5.33}$$

The lowered $\varepsilon_{\alpha\beta}$ is defined such that $\varepsilon_{12} = -1$. Note that for a $2 \times 2$ unit determinant matrix, $(A^{-1})_\beta{}^\alpha = \varepsilon^{\alpha\gamma}A_\gamma{}^\delta\varepsilon_{\delta\beta}$.

**Left-handed Weyl spinors:**   The discussion for a left-handed Weyl spinor $\bar{\psi}^{\dot{\alpha}}(x)$ follows in an analogous manner, transforming in the $(1/2,0)$ irreducible representation of SL(2, $\mathbb{C}$), given by

$$D_{(1/2,0)}(X_i^+) = 0, \qquad D_{(1/2,0)}(X_i^-) = \frac{\hbar}{2}\sigma_i$$

$$\implies \qquad D_{(1/2,0)}(M_{\mu\nu}) = \bar{s}_{\mu\nu} = \frac{i\hbar}{2}\bar{\sigma}_{[\mu}\sigma_{\nu]} = \frac{i\hbar}{4}\left(\bar{\sigma}_\mu\sigma_\nu - \bar{\sigma}_\nu\sigma_\mu\right) \in \mathfrak{sl}(2,\mathbb{C}). \tag{5.34}$$

This still corresponds to $D_{(1/2,0)}(J_i) = \hbar/2\,\sigma_i$, but $D_{(1/2,0)}(K_i) = i\hbar/2\,\sigma_i$. Therefore, the left-handed and right-handed Weyl spinors transform in the same way under rotations, but differently under boosts. In components

$$(\bar{s}_{\mu\nu})^{\dot{\alpha}}{}_{\dot{\beta}} = \frac{i\hbar}{4}\left((\bar{\sigma}_\mu)^{\dot{\alpha}\alpha}(\sigma_\nu)_{\alpha\dot{\beta}} - (\bar{\sigma}_\nu)^{\dot{\alpha}\alpha}(\sigma_\mu)_{\alpha\dot{\beta}}\right). \tag{5.35}$$

It can again be checked that $\bar{s}_{\mu\nu}$ satisfies the Lorentz algebra commutation relations (5.10). Denoting

$$D_{(1/2,0)}(\Lambda) = \bar{A}(\Lambda) = \exp\left(\frac{i}{2\hbar}\omega^{\mu\nu}\bar{s}_{\mu\nu}\right) \in \text{SL}(2,\mathbb{C}), \tag{5.36}$$

the left-handed Weyl spinor $\psi^{\dot{\alpha}}(x)$ transforms as

$$\bar{\psi}^{\dot{\alpha}}(x) \to \bar{A}(\Lambda)^{\dot{\alpha}}{}_{\dot{\beta}}\psi^{\dot{\beta}}(\Lambda^{-1}x). \tag{5.37}$$

In a similar manner, one can define the "anti-fundamental" Weyl spinor as

$$\bar{\psi}_{\dot{\alpha}}(x) = \varepsilon_{\dot{\alpha}\dot{\beta}}\psi^{\dot{\beta}}(x), \qquad \bar{\psi}_{\dot{\alpha}}(x) \to \varepsilon_{\dot{\alpha}\dot{\beta}}\bar{A}(\Lambda)^{\dot{\beta}}{}_{\dot{\gamma}}\varepsilon^{\dot{\gamma}\dot{\delta}}\bar{\psi}_{\dot{\delta}}(\Lambda^{-1}x) = \bar{\psi}_{\dot{\beta}}(\Lambda^{-1}x)(\bar{A}(\Lambda)^{-1})^{\dot{\beta}}{}_{\dot{\alpha}}. \tag{5.38}$$

Here $\varepsilon_{\dot{\alpha}\dot{\beta}}$, $\varepsilon^{\dot{\alpha}\dot{\beta}}$ are again defined so that $\varepsilon^{12} = 1$, $\varepsilon_{12} = -1$.

Note that since $\sigma_\mu^\dagger = \sigma_\mu$ and $\bar{\sigma}_\mu^\dagger = \bar{\sigma}_\mu$, it follows that $\bar{s}_{\mu\nu} = s_{\mu\nu}^\dagger$, and as a consequence, $\bar{A}(\Lambda)^{-1} = A(\Lambda)^\dagger$. Therefore, if $\psi_\alpha(x)$ is a right-handed Weyl spinor, its complex conjugate $(\psi_\alpha)^*$ is a left-handed (anti-fundamental) Weyl spinor, and vice-versa. This is a characteristic feature of $\text{SO}^+(3,1)$ (or SL(2, $\mathbb{C}$) or SU(2) × SU(2)) representations, as opposed to SU(2) representations discussed in section 4.3, where complex conjugation yields equivalent representations.

**Lorentz invariants:**   We would like to observe that the Kronecker delta symbols $\delta^\alpha_\beta$, $\delta^{\dot{\alpha}}_{\dot{\beta}}$, as well as the Levi-Civita symbols $\varepsilon_{\alpha\beta}$, $\varepsilon^{\alpha\beta}$, $\varepsilon_{\dot{\alpha}\dot{\beta}}$, $\varepsilon^{\dot{\alpha}\dot{\beta}}$ are all Lorentz invariants. Furthermore, $(\sigma_\mu)_{\alpha\dot{\alpha}}$ and $(\bar{\sigma}_\mu)^{\dot{\alpha}\alpha}$ are also Lorentz invariants, i.e.

$$\sigma_\mu \to (\Lambda^{-1})^\nu{}_\mu A(\Lambda)\sigma_\nu A(\Lambda)^\dagger = \sigma_\mu, \qquad \bar{\sigma}_\mu \to (\Lambda^{-1})^\nu{}_\mu \bar{A}(\Lambda)\bar{\sigma}_\nu \bar{A}(\Lambda)^\dagger = \bar{\sigma}_\mu, \tag{5.39}$$

or in components

$$(\Lambda^{-1})^\nu{}_\mu A(\Lambda)_\alpha{}^\beta (\sigma_\nu)_{\beta\dot{\beta}} (\bar{A}(\Lambda)^{-1})^{\dot{\beta}}{}_{\dot{\alpha}} = (\sigma_\mu)_{\alpha\dot{\alpha}},$$

$$(\Lambda^{-1})^\nu{}_\mu \bar{A}(\Lambda)^{\dot{\alpha}}{}_{\dot{\beta}} (\bar{\sigma}_\nu)^{\dot{\beta}\beta} (A(\Lambda)^{-1})_\beta{}^\alpha = (\bar{\sigma}_\mu)^{\dot{\alpha}\alpha}. \tag{5.40}$$

This can be proved using the identities

$$s_{\mu\nu}\sigma_\lambda - \sigma_\lambda\bar{s}_{\mu\nu} = i\hbar(\eta_{\mu\lambda}\sigma_\nu - \eta_{\nu\lambda}\sigma_\mu), \qquad \bar{s}_{\mu\nu}\bar{\sigma}_\lambda - \bar{\sigma}_\lambda s_{\mu\nu} = i\hbar(\eta_{\mu\lambda}\bar{\sigma}_\nu - \eta_{\nu\lambda}\bar{\sigma}_\mu). \tag{5.41}$$

Note that $\omega^{\mu\nu}s_{\mu\nu}\sigma_\lambda = \sigma_\lambda\omega^{\mu\nu}\bar{s}_{\mu\nu} - 2i\hbar\sigma_\nu\omega^\nu{}_\lambda$ and $\omega^{\mu\nu}\bar{s}_{\mu\nu}\bar{\sigma}_\lambda = \bar{\sigma}_\lambda\omega^{\mu\nu}s_{\mu\nu} - 2i\hbar\bar{\sigma}_\nu\omega^\nu{}_\lambda$. This leads to, for instance

$$
\begin{aligned}
A(\Lambda)\sigma_\lambda &= \sum_{n=0}^{\infty} \frac{1}{n!}\left(\frac{i}{2\hbar}\omega^{\mu\nu}s_{\mu\nu}\right)^n \sigma_\lambda \\
&= \sum_{n=0}^{\infty} \frac{1}{n!}\sum_{r=0}^{n}\frac{n!}{r!(n-r)!}(\sigma\omega^{n-r})_\lambda\left(\frac{i}{2\hbar}\omega^{\mu\nu}\bar{s}_{\mu\nu}\right)^r \\
&= \sum_{r=0}^{\infty}\frac{1}{r!}\left(\sum_{n=r}^{\infty}\frac{1}{(n-r)!}(\sigma\omega^{n-r})_\lambda\right)\left(\frac{i}{2\hbar}\omega^{\mu\nu}\bar{s}_{\mu\nu}\right)^r \\
&= \sigma_\mu\exp(i\omega^\mu{}_\lambda)\exp\left(\frac{i}{2\hbar}\omega^{\mu\nu}\bar{s}_{\mu\nu}\right) \\
&= \sigma_\mu\Lambda^\mu{}_\lambda\bar{A}(\Lambda),
\end{aligned}
\tag{5.42}
$$

where $(\sigma\omega^m)_\lambda = \sigma_{\mu_1}\omega^{\mu_1}{}_{\mu_2}\omega^{\mu_2}{}_{\mu_3}\ldots\omega^{\mu_m}{}_\lambda$. In the third step, we have exchanged the summations. The other identity follows in a similar manner.

**Dirac spinors:** A Dirac spinor $\Psi^A(x)$, with $A = 1, 2, 3, 4$, is a 4-component spinor field that transforms in the $(1/2, 0) \oplus (0, 1/2)$ *reducible* representation of $\text{SL}(2, \mathbb{C})$, given by

$$
D_{\text{dirac}}(\Lambda) = \begin{pmatrix} D_{(0,1/2)}(\Lambda) & 0 \\ 0 & D_{(1/2,0)}(\Lambda) \end{pmatrix} = \begin{pmatrix} A(\Lambda) & 0 \\ 0 & \bar{A}(\Lambda) \end{pmatrix},
\tag{5.43}
$$

or equivalently

$$
D_{\text{dirac}}(M_{\mu\nu}) = \begin{pmatrix} s_{\mu\nu} & 0 \\ 0 & \bar{s}_{\mu\nu} \end{pmatrix}.
\tag{5.44}
$$

A Dirac spinor $\Psi^A(x)$ can itself be understood as a direct sum of a left-handed and a right-handed Weyl spinor

$$
\Psi^A(x) = \begin{pmatrix} \psi_\alpha(x) \\ \bar{\psi}^{\dot\alpha}(x) \end{pmatrix},
\tag{5.45}
$$

so that

$$
\Psi^A(x) \to D_{\text{dirac}}(\Lambda)^A{}_B\Psi^B(\Lambda^{-1}x).
\tag{5.46}
$$

Dirac representation can also be written in terms of the gamma-matrices (in Weyl basis)

$$
\gamma_\mu = \begin{pmatrix} 0 & \sigma_\mu \\ \bar{\sigma}_\mu & 0 \end{pmatrix}, \qquad \{\gamma_\mu, \gamma_\nu\} = -2\eta_{\mu\nu}, \qquad D_{\text{Dirac}}(M_{\mu\nu}) = \frac{i\hbar}{4}[\gamma_\mu, \gamma_\nu].
\tag{5.47}
$$

Let us define the chiral projection operators

$$
P_\pm = \frac{1}{2}\left(1 \pm \gamma_5\right), \qquad \gamma_5 = \frac{4}{3}D_{\text{Dirac}}(\Gamma) = i\gamma_0\gamma_1\gamma_2\gamma_3 = \begin{pmatrix} \mathbb{1} & 0 \\ 0 & -\mathbb{1} \end{pmatrix}.
\tag{5.48}
$$

Recall that $\Gamma = i/(8\hbar^2)\epsilon^{\mu\nu\rho\sigma}M_{\mu\nu}M_{\rho\sigma}$. Since $\sigma_\mu$ and $\bar{\sigma}_\mu$ are left invariant by Lorentz transformations, so are the gamma-matrices $\gamma_\mu$, and by extension $\gamma_5$ and the projectors $P_\pm$. This allows us to project out irreducible Weyl components of the Dirac spinor as

$$
P_+\Psi(x) = \begin{pmatrix} \psi_\alpha(x) \\ 0 \end{pmatrix}, \qquad P_-\Psi(x) = \begin{pmatrix} 0 \\ \bar{\psi}^{\dot\alpha}(x) \end{pmatrix}.
\tag{5.49}
$$

### 5.2.3   Vectors $(1/2,1/2)$

A vector field $V^\mu(x)$ transforms in the fundamental representation of $\text{SO}^+(3,1)$, given by

$$V^\mu(x) \to \Lambda^\mu{}_\nu V^\nu(\Lambda^{-1}x). \tag{5.50}$$

In terms of the $\text{SL}(2,\mathbb{C})$ classification, vectors can be understood as transforming under the $(1/2,1/2)$ irreducible representation; to wit

$$\sigma_\mu V^\mu(x) \to \sigma_\mu \Lambda^\mu{}_\nu V^\nu(\Lambda^{-1}x) = A(\Lambda)(\sigma_\mu V^\mu(\Lambda^{-1}x))\bar{A}(\Lambda)^{-1}, \tag{5.51}$$

or equivalently

$$\bar{\sigma}_\mu V^\mu(x) \to \sigma_\mu \Lambda^\mu{}_\nu V^\nu(\Lambda^{-1}x) = \bar{A}(\Lambda)(\sigma_\mu V^\mu(\Lambda^{-1}x))A(\Lambda)^{-1}, \tag{5.52}$$

Here we have used eq. (5.39).

## 5.3   Poincaré group

Poincaré transformations are an extension of Lorentz transformations that leave the proper distance between arbitrary spacetime points $x^\mu$, $x'^\mu$ invariant, i.e.

$$(x^\mu - x'^\mu)(x^\nu - x'^\nu)\eta_{\mu\nu} \to (x^\mu - x'^\mu)(x^\nu - x'^\nu)\eta_{\mu\nu}. \tag{5.53}$$

These are given by

$$x^\mu \to \Lambda^\mu{}_\nu x^\nu + a^\mu, \tag{5.54}$$

where $(\Lambda^\mu{}_\nu) \in \text{O}(3,1)$ are Lorentz transformations, and $a^\mu$ are constant spacetime translations. Let us denote an element of the Poincaré group as $(\Lambda, a)$. We can check that these satisfy the group axioms:

- $(\Lambda_1, a_1)(\Lambda_2, a_2) = (\Lambda_1\Lambda_2, \Lambda_1 a_2 + a_1)$.
- $\Big((\Lambda_1, a_1)(\Lambda_2, a_2)\Big)(\Lambda_3, a_3) = (\Lambda_1\Lambda_2\Lambda_3, \Lambda_1\Lambda_2 a_3 + \Lambda_1 a_2 + a_1) = (\Lambda_1, a_1)\Big((\Lambda_2, a_2)(\Lambda_3, a_3)\Big)$.
- $(\mathbb{1}, 0)(\Lambda, a) = (\Lambda, a)(\mathbb{1}, 0) = (\Lambda, a)$.
- $(\Lambda, a)^{-1} = (\Lambda^{-1}, -\Lambda^{-1}a)$.

The Poincaré group is mathematically denoted as $\mathbb{R}^{3,1} \rtimes \text{O}(3,1)$.[6] The simply connected piece of the Poincaré group is accordingly $\mathbb{R}^{3,1} \rtimes \text{SO}^+(3,1)$, and its double cover is $\mathbb{R}^{3,1} \rtimes \text{SL}(2,\mathbb{C})$.

Let us consider a representation $D$ of the simply connected Poincaré group $\mathbb{R}^{3,1} \rtimes \text{SO}^+(3,1)$. The exponential parametrisation of an element $D(\Lambda, a)$ can be written as

$$D(\Lambda, a) = \exp\left(\frac{i}{2\hbar}\omega^{\mu\nu}M_{\mu\nu} + \frac{i}{\hbar}a^\mu P_\mu\right), \tag{5.55}$$

---

[6]The "semi-direct product" symbol $\rtimes$ essentially means that $\mathbb{R}^{3,1}$ is a normal subgroup of the Poincaré group, but $\text{O}(3,1)$ is not.

where $M_{\mu\nu}$ and $P_\mu$ are the Lie algebra generators of Lorentz transformations and spacetime transla­tions respectively, while $\omega^{\mu\nu}$ and $a^\mu$ are the respective parameters. These satisfy the commutation relations

$$[P_\mu, P_\nu] = 0,$$
$$[M_{\mu\nu}, P_\rho] = i\hbar \left(\eta_{\mu\rho}P_\nu - \eta_{\nu\rho}P_\mu\right),$$
$$[M_{\mu\nu}, M_{\rho\sigma}] = i\hbar \left(\eta_{\mu\rho}M_{\nu\sigma} - \eta_{\nu\rho}M_{\mu\sigma} - \eta_{\mu\sigma}M_{\nu\rho} + \eta_{\nu\sigma}M_{\mu\rho}\right). \tag{5.56}$$

In terms of the rotation generators (angular momenta) $J_i = 1/2\,\epsilon_{ijk}M_{jk}$, boost generators $K_i = M_{i0}$, translation generators (momenta) $P_i$, and time-translation generator (Hamiltonian) $H = -cP_0$, where $c$ is the speed of light, the Lie algebra is given as

$$[H, P_i] = 0, \qquad [P_i, P_j] = 0,$$
$$[J_i, H] = 0, \qquad [J_i, P_j] = i\hbar\epsilon_{ijk}P_k, \qquad [K_i, H] = -i\hbar cP_i, \qquad [K_i, P_j] = -\frac{i\hbar}{c}\delta_{ij}H,$$
$$[J_i, J_j] = i\hbar\epsilon_{ijk}J_k, \qquad [K_i, K_j] = -i\hbar\epsilon_{ijk}J_k, \qquad [J_i, K_j] = i\hbar\epsilon_{ijk}K_k. \tag{5.57}$$

At least some of these might be recognisable from quantum mechanics. Note that the subalgebra spanned by $P_i, H$ is an invariant subalgebra of the Poincaré algebra.

## 5.4   Poincaré representations

The irreducible representations of the Poincaré group can be labelled using the Casimir operators $P^2 = \eta_{\mu\nu}P^\mu P^\nu$ and $W^2 = \eta_{\mu\nu}W^\mu W^\nu$, where $W^\mu$ is the Pauli-Lubanski vector

$$W^\mu = \frac{1}{2}\epsilon^{\mu\nu\rho\sigma}M_{\nu\rho}P_\sigma. \tag{5.58}$$

It satisfies

$$[P_\mu, W^\nu] = 0, \qquad [M_{\mu\nu}, W^\rho] = i\hbar\left(\delta^\rho_\mu W_\nu - \delta^\rho_\nu W_\mu\right), \qquad [W^\mu, W^\nu] = -i\hbar\epsilon^{\mu\nu\rho\sigma}P_\rho W_\sigma, \tag{5.59}$$

along with $W^\mu P_\mu = 0$. It can be checked that

$$[P^2, M_{\mu\nu}] = [P^2, P_\mu] = [W^2, M_{\mu\nu}] = [W^2, P_\mu] = 0. \tag{5.60}$$

We can take the states in the Hilbert space to be the eigenstates of the mutually commuting set of operators: four-momenta $P_\mu$ and helicity operator $W_0 = J_i P^i$, in addition to $P^2$ and $W^2$. We can label a state $|m, s; \vec{p}, h\rangle$ in a given representation by its mass $m$, spin $s$, momenta $p_i$, and helicity $h$, defined as

$$W^2 |m, s; \vec{p}, h\rangle = m^2 c^2 \hbar^2 s(s+1) |m, s; \vec{p}, h\rangle,$$
$$P_\mu |m, s; \vec{p}, h\rangle = p_\mu |m, s; \vec{p}, h\rangle,$$
$$W_0 |m, s; \vec{p}, h\rangle = \hbar|\vec{p}|h |m, s; \vec{p}, h\rangle, \tag{5.61}$$

where $p_\mu = (-\sqrt{m^2c^2 + \vec{p}^2}, \vec{p})$ is the four-momentum of the state. Note that these states are automatically eigenstates of the $P^2$ operator as

$$P^2 |m, s; \vec{p}, h\rangle = \eta^{\mu\nu}P_\mu P_\nu |m, s; \vec{p}, h\rangle = -m^2c^2 |m, s; \vec{p}, h\rangle. \tag{5.62}$$

To appreciate the physical meaning of the spin quantum number $s$ and helicity $h$, let us first focus on massive representations with $m > 0$. We can always Lorentz boost arbitrary states in the representation to go to the local rest frame and set $p_\mu = (-mc, \vec{0})$ or $\vec{p} = 0$. For such states

$$W^0|m, s; \vec{0}, h\rangle = 0, \qquad W^i|m, s; \vec{0}, h\rangle = -\frac{1}{c}J^i H|m, s; \vec{0}, h\rangle = -mcJ^i|m, s; \vec{0}, h\rangle. \qquad (5.63)$$

It implies that

$$W^2|m, s; \vec{0}, h\rangle = m^2 c^2 \vec{J}^2|m, s; \vec{0}, h\rangle = m^2 c^2 \hbar^2 s(s+1)|m, s; \vec{0}, h\rangle. \qquad (5.64)$$

Hence, the spin $s$ is the highest-weight quantum number associated with the total angular-momentum $\vec{J}^2$ in the local rest frame of the particle. To interpret $h$, on the other hand, we need to spatially rotate an arbitrary state to set $\vec{p} = p\hat{z} = (0, 0, p)$. We note that

$$W_0|m, s; p\hat{z}, h\rangle = J_i P^i|m, s; p\hat{z}, h\rangle = pJ_3|m, s; p\hat{z}, h\rangle = \hbar ph|m, s; p\hat{z}, h\rangle. \qquad (5.65)$$

Hence $h$ is the $J_3$ eigenvalue of a state where the momenta is aligned along the $z$-direction. It also follows that $s$ and $h$ take half integer values

$$s = 0, \frac{1}{2}, 1, \frac{3}{2}, \dots, \qquad h = -s, -s+1, \dots, s-1, s. \qquad (5.66)$$

For instance, electron (massive Dirac spinor) has $s = 1/2$ and $h = \pm 1/2$, while Z-boson (massive vector) has $s = 1$ and $h = -1, 0, +1$. Massive irreducible Poincare representations (particles) can be labelled by their mass and spin $(m, s)$.

For massless representations $m = 0$, on the other hand, we can imagine going to a Lorentz frame moving arbitrarily close to the speed of light, such that $p_\mu \to 0$. From the above analysis, this implies that both $P^2$ and $W^2$ eigenvalues are 0. Since both these operators are Poincaré invariant, the same must be true in any Lorentz frame. Hence, the spin quantum number $s$ loses all meaning and we can label the states as simply $|h; \vec{p}\rangle$. Now, if we were to perform a rotation to set $p_\mu = (-E/c, 0, 0, E/c)$, where $E$ is the energy of the state, we see that

$$W_0|h; E/c\hat{z}\rangle = \frac{E}{c}J_3|h; E/c\hat{z}\rangle = \frac{\hbar E}{c}h|h; E/c\hat{z}\rangle. \qquad (5.67)$$

Therefore $h = 0, \pm 1/2, \pm 1, \pm 3/2, \dots$ is still the $J_3$ eigenvalue in a state where the momenta is aligned along the $z$-direction. An important distinction between massless and massive particles is that helicity for a massless particle is Poincaré invariant.[7] Hence, massless irreducible representations (particles) can be labelled by just their helicity $(h)$. For instance, photons come in either left-polarised $h = -1$ or right-polarised $h = +1$ varieties. Similarly gravitons come in $h = -2$ and $h = +2$ varieties.

The concept of helicity should be contrasted with the concept of chirality mentioned before. The helicity operator $W_0 = \vec{J} \cdot \vec{P}$ is conserved, i.e. it commutes with the Hamiltonian $[H, W_0] = 0$, but is not generically Lorentz invariant because $[K_i, W_0] = i\hbar W_i$. On the other hand, the chirality operator $\Gamma = i\vec{J} \cdot \vec{K}$ is Lorentz invariant, but is not generically conserved, i.e. $[H, \Gamma] = \hbar c W_0$. The two concepts are equivalent for massless particles.

---

[7]To see this, note that $W_0|\Psi\rangle = \hbar E h/c|\psi\rangle$ and $W_3|\psi\rangle = -\hbar E h/c|\Psi\rangle$, where $|\Psi\rangle = |h; E/c\hat{z}\rangle$. Since $W^2|\psi\rangle = 0$, it follows that $W_2|\psi\rangle = W_3|\psi\rangle = 0$. Furthermore, assuming the Hamiltonian to not admit any zero-energy states, let us define the inverse Hamiltonian operator $H^{-1}$ so that $H^{-1}|\psi\rangle = E^{-1}|\psi\rangle$ and $H^{-1}W_0|\psi\rangle = \hbar h/c|\psi\rangle$. It can be checked that the commutators $[H, H^{-1}W_0] = [P_i, H^{-1}W_0] = [J_i, H^{-1}W_0] = 0$ are identically zero, while

$$[K_i, H^{-1}W_0]|\psi\rangle = [K_i, H^{-1}]W_0|\psi\rangle + H^{-1}[K_i, W_0]|\psi\rangle = i\hbar c H^{-1}P_i H^{-1}W_0|\psi\rangle + i\hbar H^{-1}W_i|\psi\rangle$$

$$= i\hbar H^{-1}\left(P_i\hbar h + W_i\right)|\psi\rangle = 0. \qquad (5.68)$$

Hence, $H^{-1}W_0$ is Poincaré invariant for massless representations.

# 6 | Symmetries in field theory

In previous sections, we have provided an abstract discussion of the symmetry groups that enter the physical theories of particle physics: internal U(1) and SU($N$) symmetries and spacetime Poincaré symmetries. We now proceed to see how these symmetries are implemented in the framework of quantum field theories. In the following, we study various field theoretic topics and techniques such as global and local (gauge) symmetries, their spontaneous breaking, and Yukawa couplings. In the process, we also write down field theoretic models for quantum electrodynamics (QED), quantum chromodynamics (QCD), and the electroweak theory. The discussion here will be quite brief. An elaborate discussion can be found in any of the standard texts on quantum field theory, like [14–16], or those geared towards particle physics, like [17–19].

## 6.1 Global symmetries

Let us consider a field theory characterised by an action $S[\Phi]$, written as a functional of the spacetime field(s) $\Phi(x)$. Assuming the field theory to be local, we can write the action in terms of a local Lagrangian density $\mathcal{L}(\Phi(x), \partial\Phi(x), \ldots)$, written as a function of $\Phi(x)$ and its derivatives $\partial_\mu\Phi(x)$, $\partial_\mu\partial_\nu\Phi(x)$ and so on, i.e.

$$S[\Phi] = \int \mathrm{d}t\mathrm{d}^3x \, \mathcal{L}(\Phi(x), \partial\Phi(x), \ldots). \tag{6.1}$$

The equations of motion for $\Phi(x)$ can be obtained by varying $S[\Phi]$ with respect to $\Phi$, leading to

$$\frac{\delta S[\Phi]}{\delta\Phi(x)} = \frac{\partial\mathcal{L}(\Phi(x), \partial\Phi(x), \ldots)}{\partial\Phi(x)} - \partial_\mu\left(\frac{\partial\mathcal{L}(\Phi(x), \partial\Phi(x), \ldots)}{\partial(\partial_\mu\Phi(x))}\right) + \ldots = 0. \tag{6.2}$$

The middle expression above is known as the Euler-Lagrange derivative of the Lagrangian density $\mathcal{L}$, denoted by $\delta\mathcal{L}/\delta\Phi$. The generalisation to multiple fields is straight-forward.

A field theory is said to be admit a symmetry group $G$ if the action $S[\Phi]$ is invariant under the action of $g \in G$ on the fields $\Phi(x) \xrightarrow{g} \Phi'(x)$, i.e. $S[\Phi(x)] = S[\Phi'(x)]$. A continuous (Lie group) symmetry $G$ is said to be *global*, if the transformation group parameters $g \in G$ are not dependent on spacetime points. Note that spacetime Poincaré symmetries are global, because both the Lorentz matrix $\Lambda^\mu{}_\nu$ and translation parameters $a^\mu$ are constants.

### 6.1.1 Noether's theorem

Given a field theory invariant under spacetime Poincaré symmetries and global internal symmetries $G_{\mathrm{int}}$, the fields $\Phi(x)$ transform as

$$\Phi(x) \to \Phi'(x) = (D(g) \otimes D(\Lambda))\Phi(\Lambda^{-1}x - a), \tag{6.3}$$

for $g \in G_{\mathrm{int}}$, $\Lambda^\mu{}_\nu \in \mathrm{SO}^+(3,1)$, and $a^\mu \in \mathbb{R}^{3,1}$. Due to this transformation being a symmetry of the action, the Lagrangian density must return to itself up to a total derivative term

$$\mathcal{L}(\Phi', \partial\Phi', \ldots) = \mathcal{L}(\Phi, \partial\Phi, \ldots) - \partial_\mu K^\mu, \tag{6.4}$$

for some $K^\mu$. Let us consider an infinitesimal such symmetry transformation, given by

$$D(g) = \mathbb{1} + \frac{i}{\hbar}\theta^a D(T_a) + \dots, \qquad D(\Lambda) = \mathbb{1} + \frac{i}{2\hbar}\omega^{\mu\nu} D(M_{\mu\nu}) + \dots. \tag{6.5}$$

We note that

$$\Phi'(x) = \Phi(x) + \frac{i}{\hbar}\theta^a D(T_a)\Phi(x) + \frac{i}{2\hbar}\omega^{\mu\nu} D(M_{\mu\nu})\Phi(x) - (\omega^\mu{}_\nu x^\nu + a^\mu)\,\partial_\mu\Phi(x) + \dots$$
$$= \Phi(x) + \delta\Phi(x) + \dots. \tag{6.6}$$

It follows that

$$0 = \mathcal{L}(\Phi', \partial\Phi', \dots) - \mathcal{L}(\Phi, \partial\Phi, \dots) + \partial_\mu K^\mu$$
$$= \frac{\partial\mathcal{L}}{\partial\Phi}\delta\Phi + \frac{\partial\mathcal{L}}{\partial(\partial_\mu\Phi)}\partial_\mu\delta\Phi + \frac{\partial\mathcal{L}}{\partial(\partial_\mu\partial_\nu\Phi)}\partial_\mu\partial_\nu\delta\Phi + \dots + \partial_\mu K^\mu$$
$$= \partial_\mu\left(\frac{\partial\mathcal{L}}{\partial(\partial_\mu\Phi)}\delta\Phi + \frac{\partial\mathcal{L}}{\partial(\partial_\mu\partial_\nu\Phi)}\partial_\nu\delta\Phi - \partial_\nu\left(\frac{\partial\mathcal{L}}{\partial(\partial_\nu\partial_\mu\Phi)}\right)\delta\Phi + \dots + K^\mu\right)$$
$$= \partial_\mu\left(\frac{\delta\mathcal{L}}{\delta(\partial_\mu\Phi)}\delta\Phi + \frac{\delta\mathcal{L}}{\delta(\partial_\mu\partial_\nu\Phi)}\partial_\nu\delta\Phi + \dots + K^\mu\right). \tag{6.7}$$

In the third line, we have used the Euler-Lagrange equations of motion for $\Phi$, while in the forth line, we have used the Euler Lagrange derivatives of the Lagrangian with respect to the derivatives of $\Phi$. Assuming the Lagrangian density $\mathcal{L}(x) = \mathcal{L}(\Phi(x), \partial\Phi(x), \dots)$ to be a scalar, and defining $\mathcal{L}'(x) = \mathcal{L}(\Phi'(x), \partial\Phi'(x), \dots)$, it must be true that

$$\mathcal{L}'(x) = \mathcal{L}(\Lambda^{-1}x - a) = \mathcal{L}(x) - (\omega^\mu{}_\nu x^\nu + a^\mu)\partial_\mu\mathcal{L}(x) + \dots = \mathcal{L}(x) - \partial_\mu\Big((\omega^\mu{}_\nu x^\nu + a^\mu)\mathcal{L}(x) + \dots\Big),$$
$$\implies K^\mu = (\omega^\mu{}_\nu x^\nu + a^\mu)\mathcal{L}(x) + \dots. \tag{6.8}$$

With this in place, we note that eq. (6.7) must be satisfied for all values of the infinitesimal transformation parameters $\theta^a$, $\omega^{\mu\nu}$, and $a^\mu$. This results in a set of conserved currents

$$J_a^\mu = -\frac{\delta\mathcal{L}}{\delta(\partial_\mu\Phi)}\frac{i}{\hbar}D(T_a)\Phi + \dots,$$

$$L^\mu{}_{\rho\sigma} = -\frac{\delta\mathcal{L}}{\delta(\partial_\mu\Phi)}\left(\frac{i}{\hbar}D(M_{\rho\sigma})\Phi - 2x_{[\sigma}\partial_{\rho]}\Phi\right) + \dots - 2\delta^\mu_{[\rho}x_{\sigma]}\mathcal{L}$$

$$= -\frac{\delta\mathcal{L}}{\delta(\partial_\mu\Phi)}\frac{i}{\hbar}D(M_{\rho\sigma})\Phi + \dots - 2T_{\text{can}}{}^\mu{}_{[\rho}x_{\sigma]},$$

$$T_{\text{can}}{}^\mu{}_\nu = -\frac{\delta\mathcal{L}}{\delta(\partial_\mu\Phi)}\partial_\nu\Phi + \dots + \delta^\mu_\nu\mathcal{L}. \tag{6.9}$$

For a canonical field theory, where the Lagrangian only depends on $\partial_\mu\Phi$ and not on the higher derivatives, the ellipsis in the expressions above go away. Generalisation to multiple fields is straightforward. These are the internal conserved currents $J_a^\mu$, angular momentum current $L^\mu{}_{\rho\sigma}$, and canonical energy-momentum tensor $T_{\text{can}}{}^\mu{}_\nu$ respectively, satisfying

$$\partial_\mu J_a^\mu = \partial_\mu L^\mu{}_{\rho\sigma} = \partial_\mu T_{\text{can}}{}^\mu{}_\nu = 0. \tag{6.10}$$

This is the statement of the Noether's theorem: *corresponding to every continuous global symmetry, there exists a locally conserved current.*

**Spin current and Belinfante energy-momentum tensor:** The angular-momentum current can be split into orbital angular momentum $(L_{\text{orbital}})^\mu{}_{\rho\sigma}$ and spin angular momentum $S^\mu{}_{\rho\sigma}$ as

$$(L_{\text{orbital}})^\mu{}_{\rho\sigma} = -2T_{\text{can}}{}^\mu{}_{[\rho}x_{\sigma]}, \qquad S^\mu{}_{\rho\sigma} = -\frac{\delta\mathcal{L}}{\delta(\partial_\mu\Phi)}\frac{i}{\hbar}D(M_{\rho\sigma})\Phi(x). \tag{6.11}$$

Only the sum of these components is conserved. In fact, $\partial_\mu S^\mu{}_{\rho\sigma} = -\partial_\mu(L_{\text{orbital}})^\mu{}_{\rho\sigma} = -T_{[\rho\sigma]}^{\text{can}}$. We can also define the symmetric Belinfante energy-momentum tensor

$$T^{\mu\nu} = T_{\text{can}}^{(\mu\nu)} + 2\partial_\lambda S^{(\mu\nu)\lambda}, \tag{6.12}$$

which is conserved $\partial_\mu T^{\mu\nu} = 0$.

**Conserved charges:** Conserved currents imply conserved charges, i.e. quantities that remain constant in time. These are given by[8]

$$\hat{T}_a = \frac{1}{c}\int \mathrm{d}^3x\, J_a^0 = -\int \mathrm{d}^3x\, \frac{\delta\mathcal{L}}{\delta(\partial_t\Phi)}\frac{i}{\hbar}D(T_a)\Phi + \dots,$$

$$\hat{K}_i = \frac{1}{c}\int \mathrm{d}^3x\, L^0{}_{i0} = -\int \mathrm{d}^3x\, \frac{\delta\mathcal{L}}{\delta(\partial_t\Phi)}\left(\frac{i}{\hbar}D(K_i)\Phi + ct\partial_i\Phi + \frac{x_i}{c}\partial_t\Phi\right) + \dots + \frac{1}{c}\int \mathrm{d}^3x\, x_i\mathcal{L},$$

$$\hat{J}_i = \frac{1}{c}\int \mathrm{d}^3x\, \frac{1}{2}\epsilon_{ijk}L^0{}_{jk} = -\int \mathrm{d}^3x\, \frac{\delta\mathcal{L}}{\delta(\partial_t\Phi)}\left(\frac{i}{\hbar}D(J_i)\Phi + \epsilon_{ijk}x_j\partial_k\Phi\right) + \dots,$$

$$\hat{H} = \int \mathrm{d}^3x\, T_{\text{can}}^{00} = \int \mathrm{d}^3x\left[\frac{\delta\mathcal{L}}{\delta(\partial_t\Phi)}\partial_t\Phi + \dots - \mathcal{L}\right],$$

$$\hat{P}^i = \frac{1}{c}\int \mathrm{d}^3x\, T_{\text{can}}^{0i} = -\int \mathrm{d}^3x\, \frac{\delta\mathcal{L}}{\delta(\partial_t\Phi)}\partial^i\Phi + \dots. \tag{6.14}$$

Again, for a canonical field theory, the ellipsis in these expressions go away and $\delta\mathcal{L}/\delta(\partial_t\Phi)$ just becomes the conjugate momentum field $\Pi = \partial\mathcal{L}/\partial(\partial_t\Phi)$. In this case, the definition of the Hamiltonian $\hat{H} = \int \mathrm{d}^3x\,(\Pi\partial_t\Phi - \mathcal{L})$ must be familiar from classical mechanics.

Upon quantisation of fields, the conserved charges get promoted to operators on the Hilbert space of states, and themselves furnish a Hermitian representation of the symmetry algebra. Given as such, the states in the Hilbert space can be parametrised by the eigenvalues of a set of mutually commuting operators. Defining $\hat{P}_0 = -\hat{H}/c$, $\hat{M}_{i0} = \hat{K}_i$, and $\hat{M}_{ij} = \epsilon_{ijk}\hat{J}_k$, in the Poincaré sector, these operators can be taken to be $\hat{P}_\mu$, $\hat{W}^\mu\hat{W}_\mu$, and $\hat{W}_0$, where $\hat{W}^\mu = 1/2\,\epsilon^{\mu\nu\rho\sigma}\hat{M}_{\nu\rho}\hat{P}_\sigma$; see section 5.4 for more details. In the internal symmetry sector, the choice of mutually commuting operators depends on the symmetry group $G_{\text{int}}$: for U(1) it is just $\hat{T} = -\int \mathrm{d}^3x\, i\Pi\Phi$, for SU(2) these can be taken to be $\hat{T}^2$ and $\hat{T}_3$ (see section 4.3), while for SU(3) we have $\hat{T}^2$, $\hat{T}^3$, $\hat{T}_3$, $\hat{Y} = 2/\sqrt{3}\,\hat{T}_8$, and $\hat{I}^2$ (see section 4.4).

---

[8]The conservation trivially follows; for instance for the Hamiltonian we have

$$\frac{\mathrm{d}\hat{H}}{\mathrm{d}t} = c\int \mathrm{d}^3x\, \partial_0 T_{\text{can}}^{00} = -c\int \mathrm{d}^3x\, \partial_i T_{\text{can}}^{i0} = -c\oint \mathrm{d}^2x\, n_i T_{\text{can}}^{i0}, \tag{6.13}$$

where $n_i$ is an outward-pointing unit vector transverse to the boundary of the region. A similar argument follows for all the remaining conserved charges as well. Physically, this implies that the rate of change of a conserved charge in a region is equal to the flux flowing into the region from its boundary.

### 6.1.2 Scalar field theory

Let us consider the example of a complex scalar field $\varphi(x)$ with Lagrangian

$$\mathcal{L}_{\text{scalar}} = -\hbar c\, \partial_\mu \varphi^* \partial^\mu \varphi - \frac{m^2 c^3}{\hbar} \varphi^* \varphi, \tag{6.15}$$

where the constant $m$ represents the mass of the field. We have chosen the coefficients in the Lagrangian such that $\varphi$ has units of $L^{-1}$. This Lagrangian has spacetime Poincaré and an internal U(1) symmetry given by

$$\varphi(x) \to e^{iq\theta} \varphi(\Lambda^{-1} x - a), \qquad \varphi^*(x) \to e^{-iq\theta} \varphi^*(\Lambda^{-1} x - a), \tag{6.16}$$

where $e^{i\theta} \in U(1)$ and $q$ is a dimensionless constant. Note that the scalar fields $\varphi(x)$, $\varphi^*(x)$ transform in the trivial $(0,0)$ representation of the Lorentz group and the $D_q(\alpha) = \alpha^q$, $D_{-q}(\alpha) = \alpha^{-q}$ representations of $\alpha \in U(1)$ respectively. Taking the Lie algebra generator of $\mathfrak{u}(1)$ to be "$\hbar$", the respective representations are given as $D(1 \in \mathfrak{u}(1)) = \hbar q$ and $D(1 \in \mathfrak{u}(1)) = -\hbar q$ respectively. Substituting $\Phi = (\varphi, \varphi^*)$ in the general discussion above, we can find the conserved currents

$$J^\mu = iq\hbar c \left( (\partial^\mu \varphi^*) \varphi - \varphi^* \partial^\mu \varphi \right),$$
$$S^\mu{}_{\rho\sigma} = 0,$$
$$T^{\mu\nu}_{\text{can}} = T^{\mu\nu} = 2\hbar c\, \partial^{(\mu} \varphi^* \partial^{\nu)} \varphi - \eta^{\mu\nu} \left( \hbar c\, \partial_\lambda \varphi^* \partial^\lambda \varphi + \frac{m^2 c^3}{\hbar} \varphi^* \varphi \right). \tag{6.17}$$

Note that the spin current is zero for a scalar field and the canonical energy-momentum tensor is the same as the symmetric Belinfante energy-momentum tensor. We can also add interactions in the theory by adding terms like $-\lambda(\varphi^* \varphi)^2$ to the Lagrangian.

For a non-Abelian example, let us consider a theory of SU($N$) fundamental scalars

$$\mathcal{L}_{\text{scalar}} = -\hbar c\, \partial_\mu \varphi^\dagger \partial^\mu \varphi - \frac{m^2 c^3}{\hbar} \varphi^\dagger \varphi, \tag{6.18}$$

where $\varphi = (\varphi_1, \ldots, \varphi_N)$ is an $N$-component complex scalar field. The theory is invariant under spacetime Poincaré and an internal SU($N$) symmetry[9]

$$\varphi(x) \to U\varphi(\Lambda^{-1} x - a), \qquad \varphi^\dagger \to \varphi^\dagger(\Lambda^{-1} x - a) U^\dagger, \tag{6.19}$$

where $U = \exp(i/\hbar\, \lambda^a T_a) \in SU(N)$. We can find the conserved currents

$$J^\mu_a = ic \left( \partial^\mu \varphi^\dagger T_a \varphi - \varphi^\dagger T_a \partial^\mu \varphi \right),$$
$$S^\mu{}_{\rho\sigma} = 0,$$
$$T^{\mu\nu}_{\text{can}} = T^{\mu\nu} = 2\hbar c\, \partial^{(\mu} \varphi^\dagger \partial^{\nu)} \varphi - \eta^{\mu\nu} \left( \hbar c\, \partial_\lambda \varphi^\dagger \partial^\lambda \varphi + \frac{m^2 c^3}{\hbar} \varphi^\dagger \varphi \right). \tag{6.20}$$

Note that $T_a$'s are $N \times N$ matrices. So, for instance, the non-Abelian charge current is really $J^\mu_a = c(\partial^\mu(\varphi^\dagger)^i (T_a)_i{}^j \varphi_j - (\varphi^\dagger)^i (T_a)_i{}^j \partial^\mu \varphi_j)$, where $(\varphi^\dagger)^i = \varphi_i^*$.

---

[9]This theory also has a U(1)$^N$ symmetry, corresponding to an independent phase change of each of the scalars. We ignore this for now.

### 6.1.3 Spinor field theory

Let us consider the example of right- and left-handed Weyl spinors $\psi(x)$ and $\chi(x)$, and Dirac spinor $\Psi(x)$. The respective Lagrangians are given as[10]

$$\mathcal{L}_{\text{weyl-right}} = \frac{i\hbar c}{2}\left(\psi^\dagger \bar{\sigma}^\mu \partial_\mu \psi - \partial_\mu \psi^\dagger \bar{\sigma}^\mu \psi\right),$$

$$\mathcal{L}_{\text{weyl-left}} = \frac{i\hbar c}{2}\left(\chi^\dagger \sigma^\mu \partial_\mu \chi - \partial_\mu \chi^\dagger \sigma^\mu \chi\right),$$

$$\mathcal{L}_{\text{dirac}} = \frac{i\hbar c}{2}\left(\bar{\Psi}\gamma^\mu \partial_\mu \Psi - \partial_\mu \bar{\Psi}\gamma^\mu \Psi\right) - mc^2 \bar{\Psi}\Psi, \tag{6.22}$$

where $\bar{\Psi} = \Psi^\dagger \gamma^0$. The spinor fields have dimensions $L^{-3/2}$. The Lagrangians have spacetime Poincaré and internal U(1) symmetry given by

$$\psi(x) \to e^{iq\theta} A(\Lambda)\psi(\Lambda^{-1}x - a), \qquad \psi^\dagger(x) \to e^{-iq\theta}\psi^\dagger(\Lambda^{-1}x - a)A^\dagger(\Lambda),$$

$$\chi(x) \to e^{iq\theta} \bar{A}(\Lambda)\chi(\Lambda^{-1}x - a), \qquad \chi^\dagger(x) \to e^{-iq\theta}\chi^\dagger(\Lambda^{-1}x - a)\bar{A}^\dagger(\Lambda),$$

$$\Psi(x) \to e^{iq\theta} D_{\text{dirac}}(\Lambda)\Psi(\Lambda^{-1}x - a), \qquad \bar{\Psi}(x) \to e^{-iq\theta}\bar{\Psi}(\Lambda^{-1}x - a)D_{\text{dirac}}(\Lambda^{-1}), \tag{6.23}$$

where $e^{i\theta} \in U(1)$ and $q$ is a constant representing the charge of the spinor fields. See the definition of the matrices $A(\Lambda)$, $\bar{A}(\Lambda)$, $D_{\text{dirac}}(\Lambda)$ and $\sigma^\mu$, $\bar{\sigma}^\mu$, $\gamma^\mu$ in section 5.2.2. The identities in eq. (5.39) shall be useful in showing the Poincaré invariance of the Lagrangians. A curious feature of Weyl spinors as opposed to Dirac spinors is that Poincaré invariance does not allow a mass term; this will turn out to be crucial in the Standard Model. Substituting $\Phi = (\psi, \psi^\dagger)$ in the general discussion above, we can find the conserved currents for the right-handed Weyl spinor

$$J^\mu = q\hbar c\, \psi^\dagger \bar{\sigma}^\mu \psi,$$

$$S^\mu{}_{\rho\sigma} = \frac{c}{2}\psi^\dagger\left(\bar{\sigma}^\mu s_{\rho\sigma} + \bar{s}_{\rho\sigma}\bar{\sigma}^\mu\right)\psi,$$

$$T^{\mu\nu}_{\text{can}} = -\frac{i\hbar c}{2}\left(\psi^\dagger \bar{\sigma}^\mu \partial^\nu \psi - \partial^\nu \psi^\dagger \bar{\sigma}^\mu \psi\right), \qquad T^{\mu\nu} = -\frac{i\hbar c}{2}\left(\psi^\dagger \bar{\sigma}^{(\mu}\partial^{\nu)}\psi - \partial^{(\nu}\psi^\dagger \bar{\sigma}^{\mu)}\psi\right). \tag{6.24}$$

We have used the equations of motion to simplify the energy-momentum tensor. It can be checked that $\partial_\lambda S^{(\mu\nu)\lambda} = 0$, therefore the Belinfante energy-momentum is merely the symmetric part of the canonical one. Similarly, for left-handed Weyl spinor, we take $\Phi = (\chi, \chi^\dagger)$ and find

$$J^\mu = q\hbar c\, \chi^\dagger \sigma^\mu \chi,$$

$$S^\mu{}_{\rho\sigma} = \frac{c}{2}\chi^\dagger\left(\sigma^\mu \bar{s}_{\rho\sigma} + s_{\rho\sigma}\sigma^\mu\right)\chi,$$

$$T^{\mu\nu}_{\text{can}} = -\frac{i\hbar c}{2}\left(\chi^\dagger \sigma^\mu \partial^\nu \chi - \partial^\nu \chi^\dagger \sigma^\mu \chi\right), \qquad T^{\mu\nu} = -\frac{i\hbar c}{2}\left(\chi^\dagger \sigma^{(\mu}\partial^{\nu)}\chi - \partial^{(\nu}\chi^\dagger \sigma^{\mu)}\chi\right). \tag{6.25}$$

---

[10]Note that, up to a total derivative term, the Dirac Lagrangian can also be expressed into a more popular form

$$\mathcal{L} = i\hbar c\bar{\Psi}\gamma^\mu \partial_\mu \Psi - mc^2 \bar{\Psi}\Psi. \tag{6.21}$$

For Dirac spinor, we take $\Phi = (\Psi, \bar{\Psi})$ and find

$$J^\mu = q\hbar c\, \bar{\Psi}\gamma^\mu\Psi,$$

$$S^\mu{}_{\rho\sigma} = \frac{i\hbar c}{8}\bar{\Psi}\{\gamma^\mu, [\gamma_\rho, \gamma_\sigma]\}\Psi,$$

$$T^{\mu\nu}_{\text{can}} = -\frac{i\hbar c}{2}\left(\bar{\Psi}\gamma^\mu\partial^\nu\Psi - \partial^\nu\bar{\Psi}\gamma^\mu\Psi\right), \qquad T^{\mu\nu} = -\frac{i\hbar c}{2}\left(\bar{\Psi}\gamma^{(\mu}\partial^{\nu)}\Psi - \partial^{(\nu}\bar{\Psi}\gamma^{\mu)}\Psi\right). \tag{6.26}$$

The discussion can be repeated when the spinors are SU($N$)-fundamental. The respective Lagrangians stay notationally the same

$$\mathcal{L}_{\text{weyl-right}} = \frac{i\hbar c}{2}\left(\psi^\dagger\bar{\sigma}^\mu\partial_\mu\psi - \partial_\mu\psi^\dagger\bar{\sigma}^\mu\psi\right),$$

$$\mathcal{L}_{\text{weyl-left}} = \frac{i\hbar c}{2}\left(\chi^\dagger\sigma^\mu\partial_\mu\chi - \partial_\mu\chi^\dagger\sigma^\mu\chi\right),$$

$$\mathcal{L}_{\text{dirac}} = \frac{i\hbar c}{2}\left(\bar{\Psi}\gamma^\mu\partial_\mu\Psi - \partial_\mu\bar{\Psi}\gamma^\mu\Psi\right) - mc^2\bar{\Psi}\Psi, \tag{6.27}$$

but now have spacetime Poincaré and internal SU($N$) symmetry given by

$$\psi(x) \to UA(\Lambda)\psi(\Lambda^{-1}x - a), \qquad \psi^\dagger(x) \to \psi^\dagger(\Lambda^{-1}x - a)A^\dagger(\Lambda)U^\dagger,$$

$$\chi(x) \to U\bar{A}(\Lambda)\chi(\Lambda^{-1}x - a), \qquad \chi^\dagger(x) \to \chi^\dagger(\Lambda^{-1}x - a)\bar{A}^\dagger(\Lambda)U^\dagger,$$

$$\Psi(x) \to UD_{\text{dirac}}(\Lambda)\Psi(\Lambda^{-1}x - a), \qquad \bar{\Psi}(x) \to \bar{\Psi}^j(\Lambda^{-1}x - a)D_{\text{dirac}}(\Lambda^{-1})U^\dagger, \tag{6.28}$$

where $U = \exp(i/\hbar\, \lambda^a T_a) \in$ SU($N$). Note that the SU($N$) and Poincaré symmetries act independently. For example, for the right-handed Weyl spinor, in components we have the transformation rule $\psi_{\alpha i}(x) \to U_i{}^j A(\Lambda)_\alpha{}^\beta\psi_{\beta j}(\Lambda^{-1}x - a)$. The conserved currents for the right-handed Weyl spinor are

$$J_a^\mu = c\psi^\dagger\bar{\sigma}^\mu T_a\psi,$$

$$S^\mu{}_{\rho\sigma} = \frac{c}{2}\psi^\dagger\left(\bar{\sigma}^\mu s_{\rho\sigma} + \bar{s}_{\rho\sigma}\bar{\sigma}^\mu\right)\psi,$$

$$T^{\mu\nu}_{\text{can}} = -\frac{i\hbar c}{2}\left(\psi^\dagger\bar{\sigma}^\mu\partial^\nu\psi - \partial^\nu\psi^\dagger\bar{\sigma}^\mu\psi\right), \qquad T^{\mu\nu} = -\frac{i\hbar c}{2}\left(\psi^\dagger\bar{\sigma}^{(\mu}\partial^{\nu)}\psi - \partial^{(\nu}\psi^\dagger\bar{\sigma}^{\mu)}\psi\right). \tag{6.29}$$

To avoid confusion with indices, we further note that the non-Abelian charge current is expanded as $c(\psi^\dagger)_{\dot{\alpha}i}(\bar{\sigma}^\mu)^{\dot{\alpha}\alpha}(T_a)_i{}^j\psi_{\alpha j}$, but it is neater to drop the indices. For the left-handed Weyl spinor we similarly have

$$J_a^\mu = c\chi^\dagger\sigma^\mu T_a\chi,$$

$$S^\mu{}_{\rho\sigma} = \frac{c}{2}\chi^\dagger\left(\sigma^\mu\bar{s}_{\rho\sigma} + s_{\rho\sigma}\sigma^\mu\right)\chi,$$

$$T^{\mu\nu}_{\text{can}} = -\frac{i\hbar c}{2}\left(\chi^\dagger\sigma^\mu\partial^\nu\chi - \partial^\nu\chi^\dagger\sigma^\mu\chi\right), \qquad T^{\mu\nu} = -\frac{i\hbar c}{2}\left(\chi^\dagger\sigma^{(\mu}\partial^{\nu)}\chi - \partial^{(\nu}\chi^\dagger\sigma^{\mu)}\chi\right). \tag{6.30}$$

And finally for Dirac spinors we get

$$J_a^\mu = c\bar{\Psi}\gamma^\mu T_a\Psi,$$

$$S^\mu{}_{\rho\sigma} = \frac{i\hbar c}{8}\bar{\Psi}\{\gamma^\mu, [\gamma_\rho, \gamma_\sigma]\}\Psi,$$

$$T^{\mu\nu}_{\text{can}} = -\frac{i\hbar c}{2}\left(\bar{\Psi}\gamma^\mu\partial^\nu\Psi - \partial^\nu\bar{\Psi}\gamma^\mu\Psi\right), \qquad T^{\mu\nu} = -\frac{i\hbar c}{2}\left(\bar{\Psi}\gamma^{(\mu}\partial^{\nu)}\Psi - \partial^{(\nu}\bar{\Psi}\gamma^{\mu)}\Psi\right). \tag{6.31}$$

## 6.2   Local (gauge) symmetries

Let us consider a field theory that is invariant under spacetime Poincaré transformations and global internal symmetry group $G_{\text{int}}$, i.e. $U \in G_{\text{int}}$ is not dependent on spacetime. If the field theory happens to be further invariant under arbitrary spacetime-dependent group transformations, i.e. $U(x) \in G_{\text{int}}$ is a smooth function of spacetime, the internal symmetry group $G_{\text{int}}$ is said to be a local symmetry.[11]

### 6.2.1   Gauging

A field theory described by an action $S[\Phi]$, given as

$$S[\Phi] = \int \mathrm{d}t \mathrm{d}^3 x \, \mathcal{L}(\Phi(x), \partial \Phi(x), \dots), \qquad (6.32)$$

that is invariant under a global internal symmetry $G_{\text{int}}$, is typically not invariant under the local version of the symmetry automatically. This is because under a local internal transformation $U(x) \in G_{\text{int}}$, the field $\Phi(x) \to D(U(x))\Phi(x)$ transforms "covariantly", but its derivative does not

$$\partial_\mu \Phi(x) \to D(U(x)) \, \partial_\mu \Phi(x) + \partial_\mu D(U(x)) \, \Phi(x). \qquad (6.33)$$

The second "inhomogeneous" piece in this transformation results from the spacetime dependence of the transformation parameter $U(x)$.

The situation can be remedied by introducing a gauge field $A_\mu \in \mathfrak{g}_{\text{int}}$ valued in the Lie algebra, that transforms as

$$A_\mu(x) \to U(x) \left( A_\mu(x) + \frac{i}{g} \partial_\mu \right) U^{-1}(x). \qquad (6.34)$$

Here $g$ is a dimensionless constant called the interaction strength of the gauge field.[12] Assuming the exponential parametrisation $U(x) = \exp(i/\hbar \, \theta^a(x) T_a)$ and decomposing the gauge field into components $A_\mu(x) = 1/\hbar \, A_\mu^a(x) T_a$, the infinitesimal transformation is given as

$$A_\mu^a(x) \to A_\mu^a(x) - \theta^b(x) A_\mu^c(x) f_{bca} + \frac{1}{g} \partial_\mu \theta^a(x) + \mathcal{O}(\theta^2). \qquad (6.35)$$

The gauge field can be used to define a gauge covariant derivative $\mathrm{D}_\mu \Phi(x)$ of the fields, such that $\mathrm{D}_\mu \Phi(x) \to D(U(x)) \, \mathrm{D}_\mu \Phi(x)$ under the action of the group. For instance, taking $\Phi(x)$ to transform in the fundamental representation, i.e. $\Phi(x) \to U(x)\Phi(x)$, the covariant derivative is defined as

$$\mathrm{D}_\mu \Phi(x) = \partial_\mu \Phi(x) - ig A_\mu(x)\Phi(x). \qquad (6.36)$$

Similarly, for an anti-fundamental field $\Phi(x) \to \Phi(x)U(x)^{-1}$, we have

$$\mathrm{D}_\mu \Phi(x) = \partial_\mu \Phi(x) + ig \, \Phi(x)A_\mu(x). \qquad (6.37)$$

---

[11]Spacetime Poincaré symmetry, $x^\mu \to \Lambda^\mu{}_\nu x^\nu + a^\mu$, can also be promoted to a local symmetry, i.e. invariance under arbitrary coordinate transformations $x^\mu \to \Lambda^\mu{}_\nu(x)x^\nu + a^\mu(x) = x'^\mu(x)$, known as diffeomorphism symmetry. This symmetry is present, for instance, in the general theory of relativity describing Einstein's gravity. Since the Standard Model of particle physics does not describe gravity, we will not focus on these in this course.

[12]With this convention, the dimension of the gauge field is $[A_\mu] = L^{-1}$.

And, for an adjoint field $\Phi(x) \to U(x)\Phi(x)U(x)^{-1}$, we have

$$D_\mu \Phi(x) = \partial_\mu \Phi(x) - ig[A_\mu(x), \Phi(x)]. \tag{6.38}$$

A field theory with global $G_{\text{int}}$ symmetry can typically be turned into a theory with local $G_{\text{int}}$ symmetry by simply converting all partial derivatives $\partial_\mu$ to covariant derivatives $D_\mu$, i.e.

$$S[\Phi] = \int dt d^3x \, \mathcal{L}(\Phi(x), \partial\Phi(x), \ldots) \to S[\Phi; A] = \int dt d^3x \, \mathcal{L}(\Phi(x), D\Phi(x), \ldots). \tag{6.39}$$

**Conserved currents via variation:**   A curious byproduct of the gauging procedure is that the conserved current $J_a^\mu$, associated with global $G_{\text{int}}$ symmetry, can be obtained by varying $S[\Phi]$ with respect to the gauge field components $A_\mu^a$, i.e.

$$J_a^\mu = \frac{1}{g} \frac{\delta S[\Phi, A]}{\delta A_\mu^a}. \tag{6.40}$$

In the limit where $A_\mu^a = 0$, this formula results in the same expression as derived in eq. (6.9). The "conservation equation" follows from requiring the action to be invariant under infinitesimal symmetry transformations $\delta\Phi = i/\hbar\, \theta^a D(T_a)\Phi$ and $\delta A_\mu^a = -\theta^b A_\mu^c f_{bca} + 1/g\, \partial_\mu \theta^a$, leading to

$$0 = \int d^4x \left[ \frac{\delta S[\Phi, A]}{\delta \Phi} \frac{i}{\hbar} \theta^a D(T_a)\Phi + \frac{\delta S[\Phi, A]}{\delta A_\mu^a} \left( -\theta^b A_\mu^c f_{bca} + \frac{1}{g} \partial_\mu \theta^a \right) \right]$$
$$= -\int d^4x \left( g f_{abc} A_\mu^b J_c^\mu + \partial_\mu J_a^\mu \right) \theta^a, \tag{6.41}$$

implying

$$\partial_\mu J_a^\mu + g f_{abc} A_\mu^b J_c^\mu = 0. \tag{6.42}$$

In the second line above, we have used the $\Phi$ equations of motion $\delta S/\delta\Phi = 0$. We see that $J_a^\mu$ is conserved in the absence of gauge field. Defining $J^\mu = 1/\hbar\, J_a^\mu T_a$ and using total anti-symmetry of $f_{abc}$, this can be more compactly denoted as

$$D_\mu J^\mu = \partial_\mu J^\mu - ig[A_\mu, J^\mu] = 0. \tag{6.43}$$

Hence, the covariant derivative of $J^\mu$ is zero.

**Local U(1) symmetry:**   Let us consider complex scalar and spinor field theories from sections 6.1.2 and 6.1.3. The respective Lagrangians get promoted to

$$\mathcal{L}_{\text{scalar}} = -\hbar c \, D_\mu \varphi^* D^\mu \varphi - \frac{m^2 c^3}{\hbar} \varphi^* \varphi,$$
$$\mathcal{L}_{\text{weyl-right}} = \frac{i\hbar c}{2} \left( \psi^\dagger \bar{\sigma}^\mu D_\mu \psi - D_\mu \psi^\dagger \bar{\sigma}^\mu \psi \right),$$
$$\mathcal{L}_{\text{weyl-left}} = \frac{i\hbar c}{2} \left( \chi^\dagger \sigma^\mu D_\mu \chi - D_\mu \chi^\dagger \sigma^\mu \chi \right),$$
$$\mathcal{L}_{\text{dirac}} = \frac{i\hbar c}{2} \left( \bar{\Psi} \gamma^\mu D_\mu \Psi - D_\mu \bar{\Psi} \gamma^\mu \Psi \right) - mc^2 \bar{\Psi}\Psi, \tag{6.44}$$

where the covariant derivatives are defined as

$$
\begin{aligned}
\mathrm{D}_\mu\varphi &= \partial_\mu\varphi - iqeA_\mu\varphi, & \mathrm{D}_\mu\varphi^* &= \partial_\mu\varphi^* + iqe\varphi^*A_\mu, \\
\mathrm{D}_\mu\psi &= \partial_\mu\psi - iqeA_\mu\psi, & \mathrm{D}_\mu\psi^\dagger &= \partial_\mu\psi^\dagger + iqe\psi^\dagger A_\mu, \\
\mathrm{D}_\mu\chi &= \partial_\mu\chi - iqeA_\mu\chi, & \mathrm{D}_\mu\chi^\dagger &= \partial_\mu\chi^\dagger + iqe\chi^\dagger A_\mu, \\
\mathrm{D}_\mu\Psi &= \partial_\mu\Psi - iqeA_\mu\Psi, & \mathrm{D}_\mu\bar{\Psi} &= \partial_\mu\bar{\Psi} + iqe\bar{\Psi}A_\mu,
\end{aligned}
\tag{6.45}
$$

where $e = \sqrt{4\pi\alpha}$ is the fundamental dimensionless electronic charge serving as the U(1) interaction strength, and $\alpha$ is the fine structure constant. These theories are now invariant under local U(1) symmetry in addition to Poincaré symmetries. The transformation of the U(1) gauge field is given as $A_\mu \to A_\mu + 1/e\,\partial_\mu\theta$. Varying the respective Lagrangians with respect to the gauge field $A_\mu$ results in the U(1) conserved currents. For spinors, we get the exact same results as in section 6.1.3, while for scalar field we get

$$
J^\mu_{\text{scalar}} = iq\hbar c\left((\mathrm{D}^\mu\varphi^*)\varphi - \varphi^*\mathrm{D}^\mu\varphi\right),
\tag{6.46}
$$

which reduces to the expression in section 6.1.2 upon setting the gauge field to zero.

**Local SU(N) symmetry:**   Similarly, the SU($N$) scalar and spinors field theories become

$$
\begin{aligned}
\mathcal{L}_{\text{scalar}} &= -\hbar c\,\mathrm{D}_\mu\varphi^\dagger\mathrm{D}^\mu\varphi - \frac{m^2c^3}{\hbar}\varphi^\dagger\varphi, \\
\mathcal{L}_{\text{weyl-right}} &= \frac{i\hbar c}{2}\left(\psi^\dagger\bar{\sigma}^\mu\mathrm{D}_\mu\psi - \mathrm{D}_\mu\psi^\dagger\bar{\sigma}^\mu\psi\right), \\
\mathcal{L}_{\text{weyl-left}} &= \frac{i\hbar c}{2}\left(\chi^\dagger\sigma^\mu\mathrm{D}_\mu\chi - \mathrm{D}_\mu\chi^\dagger\sigma^\mu\chi\right), \\
\mathcal{L}_{\text{dirac}} &= \frac{i\hbar c}{2}\left(\bar{\Psi}\gamma^\mu\mathrm{D}_\mu\Psi - \mathrm{D}_\mu\bar{\Psi}\gamma^\mu\Psi\right) - mc^2\bar{\Psi}\Psi,
\end{aligned}
\tag{6.47}
$$

where the covariant derivatives are now defined as

$$
\begin{aligned}
\mathrm{D}_\mu\varphi &= \partial_\mu\varphi - \frac{ig}{\hbar}A^a_\mu T_a\varphi, & \mathrm{D}_\mu\varphi^\dagger &= \partial_\mu\varphi^\dagger + \frac{ig}{\hbar}\varphi^\dagger T_a A^a_\mu, \\
\mathrm{D}_\mu\psi &= \partial_\mu\psi - \frac{ig}{\hbar}A^a_\mu T_a\psi, & \mathrm{D}_\mu\psi^\dagger &= \partial_\mu\psi^\dagger + \frac{ig}{\hbar}\psi^\dagger T_a A^a_\mu, \\
\mathrm{D}_\mu\chi &= \partial_\mu\chi - \frac{ig}{\hbar}A^a_\mu T_a\chi, & \mathrm{D}_\mu\chi^\dagger &= \partial_\mu\chi^\dagger + \frac{ig}{\hbar}\chi^\dagger T_a A^a_\mu, \\
\mathrm{D}_\mu\Psi &= \partial_\mu\Psi - \frac{ig}{\hbar}A^a_\mu T_a\Psi, & \mathrm{D}_\mu\bar{\Psi} &= \partial_\mu\bar{\Psi} + \frac{ig}{\hbar}\bar{\Psi}T_a A^a_\mu.
\end{aligned}
\tag{6.48}
$$

The resultant theories are now invariant under SU($N$) gauge transformations in addition to Poincaré symmetries. Varying with respect to the gauge field components $A^a_\mu$ results in the conserved currents. Again, the results for spinor fields is the same as in section 6.1.3, while for the scalar fields we get

$$
J^\mu_{a,\text{scalar}} = ic\left(\mathrm{D}^\mu\varphi^\dagger T_a\varphi - \varphi^\dagger T_a\mathrm{D}^\mu\varphi\right),
\tag{6.49}
$$

which reduces to the expression in section 6.1.2 upon setting the gauge field to zero.

### 6.2.2  Dynamical gauge fields

In the gauging procedure above, we introduced gauge fields to promote a global internal symmetry into a local one. However, the gauge fields themselves do not have any dynamics. In other words, gauging procedure only introduces coupling terms of the kind $\Phi^* A_\mu \partial^\mu \Phi$ or $\Phi^* A^\mu A_\mu \Phi$ in the action, but no kinetic term for $A_\mu$. These are known as "background gauge fields". To make gauge the fields dynamical, we need to introduce a kinetic term by hand.

**Field strength:**  The kinetic term we seek must include a $1/2\, \partial_t A_\mu^a \partial_t A_a^\mu$ piece, and must respect all the symmetries of the theory – Poincare symmetry and local internal gauge symmetry. To motivate such a term, let us consider a fundamental field $\Phi$ and compute

$$
\begin{aligned}
[D_\mu, D_\nu]\Phi &= D_\mu D_\nu \Phi - D_\nu D_\mu \Phi \\
&= -ig\left(\partial_\mu A_\nu - \partial_\nu A_\mu - ig[A_\mu, A_\nu]\right)\Phi \\
&= -ig F_{\mu\nu}\Phi,
\end{aligned}
\tag{6.50}
$$

where

$$
F_{\mu\nu} = \partial_\mu A_\nu - \partial_\nu A_\mu - ig[A_\mu, A_\nu],
\tag{6.51}
$$

is the *gauge field strength*. Since $\Phi \to U\Phi$ under a gauge transformation and, by definition, $[D_\mu, D_\nu]\Phi \to U\,[D_\mu, D_\nu]\Phi$, it must be true that

$$
F_{\mu\nu} \to U F_{\mu\nu} U^{-1}.
\tag{6.52}
$$

Hence $F_{\mu\nu}$ transforms in the adjoint representation of the group under local symmetry transformations. In components $F_{\mu\nu} = 1/\hbar\, F_{\mu\nu}^a T_a$, the field strength is given as

$$
F_{\mu\nu}^a = \partial_\mu A_\nu^a - \partial_\nu A_\mu^a + g f_{bca} A_\mu^b A_\nu^c.
\tag{6.53}
$$

The following "Bianchi identities" satisfied by the field strength can be verified

$$
D_\mu D_\nu F^{\mu\nu} = 0, \qquad D_{[\lambda} F_{\mu\nu]} = 0.
\tag{6.54}
$$

Note that $D_\lambda F^{\mu\nu} = \partial_\lambda F_{\mu\nu} - ig[A_\lambda, F_{\mu\nu}]$.

**Yang-Mills action:**  The field strength $F_{\mu\nu}$ has a term linear in $\partial_t A_\mu$, so a natural contender for the kinetic term would be $F^{\mu\nu}F_{\mu\nu}$. However, this is not gauge invariant, because $F^{\mu\nu}F_{\mu\nu} \to U F^{\mu\nu} U^{-1} U F_{\mu\nu} U^{-1} = U F^{\mu\nu} F_{\mu\nu} U^{-1}$. This situation can be remedied by taking a trace, leading to the Yang-Mills action

$$
S_{\text{YM}}[A] = -\frac{\hbar c}{4C} \int dt d^3x\ \text{tr}(F^{\mu\nu} F_{\mu\nu}).
\tag{6.55}
$$

The factor of $\hbar$ is included in the action due to dimensional reasons. Using $\text{tr}(T_a, T_b) = C\hbar^2 \delta_{ab}$, in components we have

$$
\begin{aligned}
S_{\text{YM}}[A] =\ & -\frac{\hbar c}{4} \int dt d^3x\ F_a^{\mu\nu} F_{\mu\nu}^a \\
=\ & -\frac{\hbar c}{2} \int dt d^3x \bigg(\partial^\mu A_a^\nu \partial_\mu A_\nu^a - \partial^\nu A_a^\mu \partial_\mu A_\nu^a \\
& + 2g f_{bca} A_\mu^b A_\nu^c \partial^\mu A_a^\nu + \frac{1}{2} g^2 f_{efa} f_{bca} A_e^\mu A_f^\nu A_\mu^b A_\nu^c\bigg).
\end{aligned}
\tag{6.56}
$$

We get a kinetic term, quadratic in $\partial_\mu A_\nu$, plus two interaction terms: a 3-point self-interaction with coupling constant $g$ and a 4-point self-interaction with coupling constant $g^2$. The ensuing equations of motion can be written as

$$D_\mu F^{\mu\nu} = 0. \tag{6.57}$$

The Bianchi identities (6.54) further results in

$$D_\mu \star F^{\mu\nu} = 0, \tag{6.58}$$

where $\star F^{\mu\nu} = 1/2 \, \epsilon^{\mu\nu\rho\sigma} F_{\rho\sigma}$. Yang-Mills theories describe (non-Abelian) gauge fields in vacuum.

We can bring the Yang-Mills action $S_{\mathrm{YM}}[A]$ and the gauge-invariant matter action $S_m[\Phi; A]$ together to obtain the full theory of dynamical gauge fields coupled to matter

$$\begin{aligned} S[\Phi, A] &= S_{\mathrm{YM}}[A] + S_m[\Phi; A] \\ &= \int \mathrm{dt d}^3 x \left( -\frac{\hbar c}{4C} \operatorname{tr}(F^{\mu\nu} F_{\mu\nu}) + \mathcal{L}_m(\Phi, \mathrm{D}\Phi, \dots) \right). \end{aligned} \tag{6.59}$$

The equations of motion now read

$$D_\mu F^{\mu\nu} = \frac{g}{\hbar c} J^\mu. \tag{6.60}$$

The conservation equation $D_\mu J^\mu = 0$ follows using the Bianchi identities (6.54).

**Maxwell's action:**   A special case of Yang-Mills action is when the gauge symmetry group is U(1). In this case the gauge field strength is simply the Maxwell field strength tensor $F_{\mu\nu} = \partial_\mu A_\nu - \partial_\nu A_\mu$. This results in the Maxwell's action

$$\mathcal{L}_{\mathrm{Maxwell}} = -\frac{\hbar c}{4} \int \mathrm{dt d}^3 x \, F^{\mu\nu} F_{\mu\nu}, \tag{6.61}$$

with associated equations $\partial_\mu F^{\mu\nu} = \partial_\mu \star F^{\mu\nu} = 0$. This theory describes free electromagnetic fields in vacuum. A qualitative difference from the generic non-Abelian case is that this theory is free, i.e. it does not have any interactions. This signals to the physical fact that U(1) electromagnetic photons do not interact among themselves, while, for example, SU(3) gluons do. When coupled to matter, the Lagrangian reads

$$\begin{aligned} S[\Phi, A] &= S_{\mathrm{Maxwell}}[A] + S_m[\Phi; A] \\ &= \int \mathrm{dt d}^3 x \left( -\frac{\hbar c}{4} F^{\mu\nu} F_{\mu\nu} + \mathcal{L}_m(\Phi, \mathrm{D}\Phi, \dots) \right), \end{aligned} \tag{6.62}$$

resulting in the Maxwell's equations[13]

$$\partial_\mu F^{\mu\nu} = \frac{e}{\hbar c} J^\mu. \tag{6.63}$$

The remaining Maxwell's equations are obtained by the Bianchi identity $\partial_\mu \star F^{\mu\nu} = 0$.

---

[13]Few comments are in order on the dimensions. We have take the gauge field $A_\mu$ to have dimensions $L^{-1}$. However, the usual electromagnetic gauge field has dimensions $MLT^{-1}Q^{-1}$, which can be obtained by rescaling $A_\mu \to A_\mu/\sqrt{\hbar c \mu_0}$ where $\mu_0$ is the permeability of free space. This results in the familiar Maxwell's action $-\frac{1}{4\mu_0} \int \mathrm{dt d}^3 x \, F^{\mu\nu} F_{\mu\nu}$. Similarly, the charge current should be rescaled as $J^\mu \to J^\mu \sqrt{\hbar c \mu_0}/e$. The Maxwell's equations then become $\partial_\mu F^{\mu\nu} = \mu_0 J^\nu$.

## 6.3   QED and QCD

Quantum electrodynamics (QED) is a theory with U(1) gauge symmetry. This can be obtained by coupling U(1) matter fields (scalars or spinors) to Maxwell's action. For instance, we have scalar- and Dirac-QED Lagrangians

$$\mathcal{L}_{\text{scalar-QED}} = -\hbar c\, \mathrm{D}_\mu \varphi^* \mathrm{D}^\mu \varphi - \frac{m^2 c^3}{\hbar} \varphi^* \varphi - \frac{\hbar c}{4} F^{\mu\nu} F_{\mu\nu},$$

$$\mathcal{L}_{\text{dirac-QED}} = \frac{i\hbar c}{2} \left( \bar{\Psi} \gamma^\mu \mathrm{D}_\mu \Psi - \mathrm{D}_\mu \bar{\Psi} \gamma^\mu \Psi \right) - mc^2 \bar{\Psi}\Psi - \frac{\hbar c}{4} F^{\mu\nu} F_{\mu\nu}, \tag{6.64}$$

The latter "Dirac-QED" or simply "QED" can describe, for instance, electrons coupled to electromagnetic fields. These are interacting theories. For instance, denoting matter fields with solid and photons by wavy lines, scalar-QED has two interaction vertices

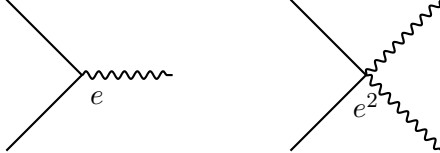

that scale as $e$ and $e^2$ respectively. On the other hand, Dirac-QED only has the first 3-point interaction vertex.

Quantum chromodynamics (QCD), on the other hand, is a theory with SU(3) gauge symmetry. This can be obtained by coupling SU(3) matter fields (scalars or spinors) to Yang-Mills action. The Lagrangians for scalar- and Dirac-QCD are given as

$$\mathcal{L}_{\text{scalar-QCD}} = -\hbar c\, \mathrm{D}_\mu \varphi^\dagger \mathrm{D}^\mu \varphi - \frac{m^2 c^3}{\hbar} \varphi^\dagger \varphi - \frac{\hbar c}{2} \operatorname{tr}(F^{\mu\nu} F_{\mu\nu}),$$

$$\mathcal{L}_{\text{dirac-QCD}} = \frac{i\hbar c}{2} \left( \bar{\Psi} \gamma^\mu \mathrm{D}_\mu \Psi - \mathrm{D}_\mu \bar{\Psi} \gamma^\mu \Psi \right) - mc^2 \bar{\Psi}\Psi - \frac{\hbar c}{2} \operatorname{tr}(F^{\mu\nu} F_{\mu\nu}). \tag{6.65}$$

"Dirac-QCD" or simply "QCD" can describe quarks coupled to gluons. The qualitative difference in QCD compared to QED is that we still have the matter interaction vertices given above

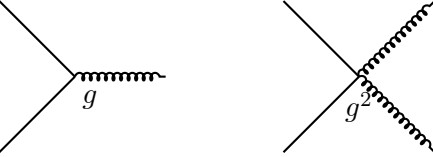

but, in addition, we also have gluons self interaction vertices

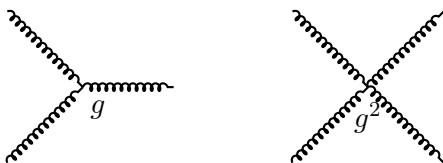

This is the fundamental reason why we see free photons in nature, but we do not see free gluons.

## 6.4   Spontaneous symmetry breaking

It is often the case in field theory that the full action respects a certain symmetry group, but the ground state is not invariant under the symmetry. Therefore, the said symmetry is not apparent in the low-energy spectrum of the theory. This phenomenon is formally known as *spontaneous symmetry breaking.*

### 6.4.1   Global symmetry breaking and Goldstone theorem

Consider a complex scalar field theory living in an arbitrary potential $V(\varphi^*\varphi)$, described by the Lagrangian

$$\mathcal{L} = -\hbar c\, \partial_\mu \varphi^* \partial^\mu \varphi - V(\varphi^*\varphi). \tag{6.66}$$

This Lagrangian admits a global U(1) symmetry given by

$$\varphi \to \exp(iq\theta)\varphi, \qquad \varphi^* \to \varphi^* \exp(-iq\theta), \tag{6.67}$$

for a constant $q$ representing the U(1) charge of $\varphi$, and a constant symmetry parameter $\theta$. Normally, we take the potential to be the simple mass term, with possible interactions, e.g. $V(\varphi^*\varphi) = m^2 c^3/\hbar\, \varphi^*\varphi + \lambda(\varphi^*\varphi)^2$. This potential has a minima at $\varphi = 0$. Notably, the $\varphi = 0$ state is invariant under the action of the U(1) symmetry, and hence the symmetry is *not* spontaneously broken. Consider now the potential

$$V(\varphi^*\varphi) = -\hbar c \mu^2 \varphi^*\varphi + \hbar c \lambda (\varphi^*\varphi)^2, \tag{6.68}$$

colloquially known as the "Mexican-hat potential". This potential has a maxima at $\varphi = 0$, whereas it admits an infinite number of minima parametrised by $\varphi_0 = v/\sqrt{2}\, \exp(iq\vartheta)$ for arbitrary $\vartheta$, where $v = \mu/\sqrt{\lambda}$. These states are no longer invariant under the U(1) symmetry, but rather transform into each other according to $\vartheta \to \vartheta + \theta$. Since all these minima states are equivalent and have exactly the same potential energy, the system arbitrarily picks one as its ground state, spontaneously breaking the U(1) symmetry. Without loss of generality, we can take this state to be $\vartheta = 0$ or $\varphi_0 = v/\sqrt{2}$. Excitations about this ground state no longer respect the original U(1) symmetry of the field theory. See figure 1.

To obtain an effective theory of aforementioned excitations, let us expand $\varphi$ about the ground state $\varphi_0 = v/\sqrt{2}$ and parametrise

$$\varphi(x) = \frac{1}{\sqrt{2}} \exp(iq\pi(x))(v + \eta(x)). \tag{6.69}$$

The real scalar field $\eta$ denotes excitations in the magnitude of $\varphi$, going up and down the valley of the Mexican-hat potential. On the other hand, the real scalar field $\pi$ denotes excitations in the phase of $\varphi$, going around the ring in the potential. As a consequence, $\eta$ is a massive field, while $\pi$ is massless. Generically, we get one massless field, known as a *Goldstone boson*, for every spontaneously broken symmetry generator. This is known as the Goldstone theorem. To be more precise, let us write the Lagrangian in terms of $\pi$ and $\eta$. We get

$$\mathcal{L} = -\frac{\hbar c}{2}\partial_\mu \eta \partial^\mu \eta - \frac{q^2 \hbar c}{2}(v+\eta)^2 \partial_\mu \pi \partial^\mu \pi + \frac{\hbar c \mu^2}{2}(v+\eta)^2 - \frac{\hbar c \lambda}{4}(v+\eta)^4$$

$$= -\frac{\hbar c}{2}\partial_\mu \eta \partial^\mu \eta - \hbar c \mu^2 \eta^2 - \frac{q^2 v^2 \hbar c}{2}\partial_\mu \pi \partial^\mu \pi + \text{interactions.} \tag{6.70}$$

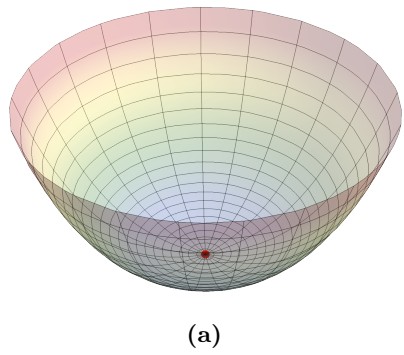
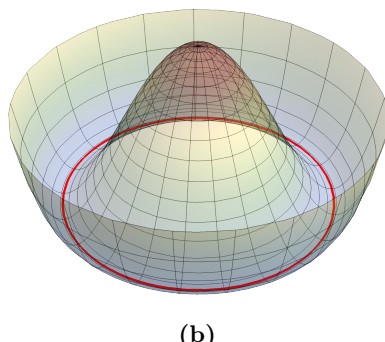

**(a)** **(b)**

**Figure 1:** Potential $V(\varphi^*\varphi)$ plotted against complex $\varphi$ for **(a)** unbroken symmetry and **(b)** spontaneously broken symmetry. In the former case, the potential has a unique minima at $\varphi_0 = 0$, denoted by a red dot, which is invariant under the action of the U(1) symmetry. In the latter case, the potential admits an entire one-parameter family of minima $\varphi_0 = v/\sqrt{2}\exp(i\vartheta)$, denoted by a red ring. Under the action of the U(1) symmetry, the possible minima transform into each other.

In the second step, we have ignored all cubic and higher-order interactions in $\eta$ and $\pi$. We have also ignored a constant piece in the action $\hbar c\mu^4/(4\lambda)$ that only shifts the zero point of the energy and does not affect the equations of motion. We get a massive real scalar field $\eta$, with mass $\sqrt{2}\mu\hbar/c$, and a massless Goldstone field $\pi$.

For a non-Abelian realisation of this idea, consider a field $\varphi$ transforming in some $n$-dimensional unitary representation $D(G)$ of the internal symmetry group $G$,[14] i.e. an element $U \in G$ acts as

$$\varphi \to D(U)\varphi, \qquad \varphi^\dagger \to \varphi^\dagger D(U^{-1}). \tag{6.71}$$

Consider the $G$-invariant Lagrangian

$$\mathcal{L} = -\hbar c\, \partial_\mu\varphi^\dagger \partial^\mu\varphi - V(\varphi^\dagger\varphi). \tag{6.72}$$

where we take the potential to have a similar spontaneous symmetry breaking form

$$V(\varphi^\dagger\varphi) = -\hbar c\mu^2\varphi^\dagger\varphi + \hbar c\lambda(\varphi^\dagger\varphi)^2. \tag{6.73}$$

The potential has a family of minima at $\varphi^\dagger\varphi = v^2/2$ for the system to choose from for the ground state. Without loss of generality, we can take this ground state to be

$$\varphi_0 = \frac{v}{\sqrt{2}}\hat\varphi_0, \qquad \hat\varphi_0 = \begin{pmatrix} 0 \\ \vdots \\ 0 \\ 1 \end{pmatrix}. \tag{6.74}$$

We can parametrise the fluctuations around the ground state as

$$\varphi(x) = \frac{v + \eta(x)}{\sqrt{2}}\exp\left(\frac{i}{\hbar}\pi^a(x)D(T_a)\right)\hat\varphi_0. \tag{6.75}$$

---

[14] A unitary representation $D(G)$ of a group $G$ is a matrix representation such that $D(U^{-1}) = D(U)^\dagger$ for all $U \in G$.

Note that $n$-dimensional complex field $\varphi$ has $2n$ real components. On the other hand, the fields $\eta$ and $\pi^a$ together have $\dim(\mathfrak{g}) + 1$ components. Hence, generically, some components of $\pi^a$ must leave $\hat{\varphi}_0$ invariant, and hence the respective $T_a$ must be unbroken. For example, with $\varphi$ furnishing a 2-dimensional fundamental representation of $G = \mathrm{SU}(2)$, none of the generators are unbroken. On the other hand, with $\varphi$ furnishing a 3-dimensional fundamental representation of $G = \mathrm{SU}(3)$, three generators are unbroken; these are $J_{1,2,3} = 1/2\,\lambda_{1,2,3}$, where $\lambda_i$'s are the Gell-Mann matrices. Putting in the decomposition (6.75) into the Lagrangian, we find

$$
\begin{aligned}
\mathcal{L} &= -\frac{\hbar c}{2}\partial_\mu\eta\partial^\mu\eta - \frac{c}{2\hbar}(v+\eta)^2(\hat{\varphi}_0 D(T_a)D(T_b)\hat{\varphi}_0)\partial_\mu\pi^a\partial^\mu\pi^b + \frac{\hbar c\mu^2}{2}(v+\eta)^2 - \frac{\hbar c\lambda}{4}(v+\eta)^4 \\
&= -\frac{\hbar c}{2}\partial_\mu\eta\partial^\mu\eta - \hbar c\mu^2\eta^2 - \frac{v^2 c}{4\hbar}\hat{\varphi}_0\{D(T_a),D(T_b)\}\hat{\varphi}_0\partial_\mu\pi^a\partial^\mu\pi^b + \text{interactions}.
\end{aligned} \tag{6.76}
$$

For the generators that leave the ground state $\hat{\varphi}_0$ invariant, we have $D(T_a)\hat{\varphi}_0 = 0$, and the associated $\pi^a$ fields do not show up in the Lagrangian. For instance, with $\varphi$ being a $\mathrm{SU}(2)$ fundamental field, all $\pi^{1,2,3}$ are present in the Lagrangian and make up the Goldstone bosons. However, with $\varphi$ being a $\mathrm{SU}(3)$ fundamental field, we only have $\pi^{4,5,6,7,8}$ Goldstone bosons in the Lagrangian, while $\pi^{1,2,3}$ drop out.

### 6.4.2  Local symmetry breaking and Higgs mechanism

In the previous subsection, we considered spontaneous breaking of global symmetries. The discussion is qualitatively different when the broken symmetries are local. Consider, for instance, the complex scalar theory from eq. (6.66), but coupled to $\mathrm{U}(1)$ gauge field $A_\mu$, i.e.

$$
\begin{aligned}
\mathcal{L} &= -\hbar c\,\mathrm{D}_\mu\varphi^*\mathrm{D}^\mu\varphi - V(\varphi^*\varphi) - \frac{\hbar c}{4}F^{\mu\nu}F_{\mu\nu} \\
&= -\hbar c\,\partial_\mu\varphi^*\partial^\mu\varphi - ieq\hbar c\,(\varphi^* A^\mu\partial_\mu\varphi - \partial_\mu\varphi^* A^\mu\varphi) - e^2 q^2\hbar c\,\varphi^*\varphi A_\mu A^\mu \\
&\quad - V(\varphi^*\varphi) - \frac{\hbar c}{4}F^{\mu\nu}F_{\mu\nu},
\end{aligned} \tag{6.77}
$$

with the same potential (6.68). Putting in the decomposition (6.69) for $\varphi$, we get

$$
\begin{aligned}
\mathcal{L} &= -\frac{\hbar c}{2}\partial_\mu\eta\partial^\mu\eta - \hbar c\mu^2\eta^2 - \frac{1}{2}q^2 v^2\hbar c\,\partial_\mu\pi\partial^\mu\pi \\
&\quad + eq^2 v^2\hbar c\,A^\mu\partial_\mu\pi - \frac{1}{2}e^2 q^2 v^2\hbar c\,A_\mu A^\mu - \frac{\hbar c}{4}F^{\mu\nu}F_{\mu\nu} + \text{interactions}.
\end{aligned} \tag{6.78}
$$

As it turns out, in this theory, we can get rid of the Goldstone boson entirely by a redefinition of the gauge field. Let us redefine

$$
A_\mu \to A_\mu + \frac{1}{e}\partial_\mu\pi, \tag{6.79}
$$

leading to

$$
\mathcal{L} = -\frac{\hbar c}{2}\partial_\mu\eta\partial^\mu\eta - \hbar c\mu^2\eta^2 - \frac{\hbar c}{4}F^{\mu\nu}F_{\mu\nu} - \frac{1}{2}\frac{m_A^2 c^3}{\hbar}A_\mu A^\mu + \text{interactions}. \tag{6.80}
$$

where $m_A = eqv\hbar/c$. Importantly, we note that the gauge field $A_\mu$ obtains a mass term that was previously disallowed by the gauge invariance of the action. Colloquially, we say that "the gauge field

eats the Goldstone bosons and becomes massive". Spontaneous symmetry breaking can, hence, be thought of as a mechanism to generate masses for gauge fields. This is called the *Higgs mechanism*, and is vital in particle physics to generate masses for $W_\mu^\pm$ and $Z_\mu$ gauge bosons mediating the weak forces.

The aforementioned idea of Higgs mechanism is more general than mere U(1) symmetries. To appreciate this, let us note that eqs. (6.69) and (6.79) can be together thought of as a local U(1) symmetry transformation with $\theta(x) = \pi(x)$. This is the reason why the "parameter" of this symmetry transformation $\pi$ drops out of the final Lagrangian. The final symmetry-broken Lagrangian is simply given by substituting $\varphi \to (v + \eta)/\sqrt{2}$. More generally, let us consider a field $\varphi$ transforming in some $n$-dimensional unitary representation $D(G)$ of the internal symmetry group $G$. We can couple the Lagrangian (6.72) to a non-Abelian gauge field $A_\mu$ and make it locally $G$-invariant

$$\mathcal{L} = -\hbar c \, \mathrm{D}_\mu \varphi^\dagger \mathrm{D}^\mu \varphi - V(\varphi^\dagger \varphi) - \frac{\hbar c}{4C} \mathrm{tr}(F^{\mu\nu} F_{\mu\nu}), \tag{6.81}$$

where $\mathrm{tr}(T_a T_b) = C\delta_{ab}$. The gauge covariant derivative is defined as $\mathrm{D}_\mu \varphi = \partial_\mu \varphi + ig A_\mu \varphi = \partial_\mu \varphi - ig/\hbar \, A_\mu^a D(T_a)\varphi$, and similarly for the Hermitian conjugate. Substituting the scalar field decomposition (6.75) into the Lagrangian, along with a symmetry transformation of the gauge field

$$A_\mu \to \exp\left(\frac{i}{\hbar}\pi^a T_a\right)\left(A_\mu + \frac{i}{g}\partial_\mu\right)\exp\left(-\frac{i}{\hbar}\pi^a T_a\right), \tag{6.82}$$

we note that the Lagrangian remains invariant except $\varphi \to (v + \eta)/\sqrt{2}\,\hat{\varphi}_0$. We find

$$\begin{aligned}
\mathcal{L} &= -\frac{\hbar c}{2}\partial_\mu \eta \partial^\mu \eta - \frac{g^2 \hbar c}{2}(v+\eta)^2\,\hat{\varphi}_0 A_\mu A^\mu \hat{\varphi}_0 + \frac{\hbar c \mu^2}{2}(v+\eta)^2 - \frac{\hbar c \lambda}{4}(v+\eta)^4 - \frac{\hbar c}{4C}\mathrm{tr}(F^{\mu\nu}F_{\mu\nu}) \\
&= -\frac{\hbar c}{2}\partial_\mu \eta \partial^\mu \eta - \frac{c^3}{2\hbar}\frac{(v+\eta)^2}{v^2}m_{ab}A_\mu^a A^{b\mu} + \frac{\hbar c \mu^2}{2}(v+\eta)^2 - \frac{\hbar c \lambda}{4}(v+\eta)^4 - \frac{\hbar c}{4}F_{\mu\nu}^a F_a^{\mu\nu} \\
&= -\frac{\hbar c}{2}\partial_\mu \eta \partial^\mu \eta - \hbar c \mu^2 \eta^2 - \frac{c^3}{2\hbar}m_{ab}A_\mu^a A^{b\mu} - \frac{\hbar c}{4}F_{\mu\nu}^a F_a^{\mu\nu} + \text{interactions}, 
\end{aligned} \tag{6.83}$$

where

$$m_{ab} = \frac{g^2 v^2}{2c^2}\hat{\varphi}_0 \{D(T_a), D(T_b)\}\hat{\varphi}_0. \tag{6.84}$$

is the gauge field mass squared matrix. Note that not all gauge field components get massive. For the generators that leave the ground state $\hat{\varphi}_0$ invariant, we have $D(T_a)\hat{\varphi}_0 = 0$, and the associated $m_{ab}$ components are zero. For instance, with $\varphi$ being a SU(2) fundamental, the entire $m_{ab} \neq 0$ and all the three gauge field components obtain a mass. However, with $\varphi$ being a SU(3) fundamental, we have $m_{11} = m_{12} = m_{13} = m_{22} = m_{23} = m_{33} = 0$, and only the five components $A_\mu^{4,5,6,7,8}$ obtain a mass. We note that there is a one-to-one correspondence between the "would be" Goldstone bosons, if the symmetry was global, and the massive gauge fields. This is the general Higgs mechanism.

## 6.5   Electroweak theory

The electroweak theory is a field theory describing electromagnetic and weak forces in a unified framework. The theory contains $\mathrm{SU}(2)_L \times \mathrm{U}(1)_Y$ gauge bosons $W_\mu = 1/\hbar\, W_\mu^A T_A$ and $B_\mu$ (with $T_A = \hbar/2\,\sigma_A$, where $\sigma_{A=1,2,3}$ are Pauli matrices generating the $\mathfrak{su}(2)$ algebra), coupled to a left-handed Weyl spinor $\chi_I^{\dot\alpha}$ transforming in the fundamental representation of $\mathrm{SU}(2)_L$, with the indices

$I, J, \ldots = 1, 2$ and a right-handed Weyl spinor $\psi_\alpha$ transforming in the trivial representation of $SU(2)_L$. Generalisation to multiple spinor fields is straightforward. We also introduce a complex scalar "Higgs" field $\varphi_I$ transforming in the fundamental representation of $SU(2)_L$ with a "Mexican-hat potential" $V(\varphi^\dagger \varphi)$. The theory utilises spontaneous symmetry breaking mechanism outlined above to generate masses for the weak force gauge fields and the spinors. The subscript "$L$" on $SU(2)_L$ is to remind ourselves that the symmetry only acts on left-handed particles, while the subscript "$Y$" on $U(1)_Y$ is to distinguish it from the electromagnetic $U(1)$ obtained as the residual symmetry after spontaneous symmetry breaking.

### 6.5.1  Electroweak Lagrangian

The Lagrangian for the electroweak theory is given as (suppressing group and spinor indices)

$$\frac{1}{\hbar c}\mathcal{L} = \frac{i}{2}\left(\psi^\dagger \bar{\sigma}^\mu D_\mu \psi - D_\mu \psi^\dagger \bar{\sigma}^\mu \psi\right) + \frac{i}{2}\left(\chi^\dagger \sigma^\mu D_\mu \chi - D_\mu \chi^\dagger \sigma^\mu \chi\right) - D_\mu \varphi^\dagger D^\mu \varphi - \frac{1}{\hbar c}V(\varphi^\dagger \varphi)$$
$$- \frac{1}{2}\operatorname{tr}(W^{\mu\nu}W_{\mu\nu}) - \frac{1}{4}B^{\mu\nu}B_{\mu\nu}, \tag{6.85}$$

where $B_{\mu\nu} = \partial_\mu B_\nu - \partial_\nu B_\mu$ and $W_{\mu\nu} = \partial_\mu W_\nu - \partial_\nu W_\mu - ig_w[W_\mu, W_\nu]$. The covariant derivatives are defined as

$$D_\mu \psi = \partial_\mu \psi - iq_\psi g_y B_\mu \psi,$$
$$D_\mu \chi = \partial_\mu \chi - iq_\chi g_y B_\mu \chi - ig_w W_\mu \chi,$$
$$D_\mu \varphi = \partial_\mu \varphi - iq_\varphi g_y B_\mu \varphi - ig_w W_\mu \varphi, \tag{6.86}$$

and similarly for the Hermitian conjugates. Here $g_w$ and $g_y$ are $SU(2)_L$ and $U(1)_Y$ coupling constants, while $q_\psi$, $q_\chi$, $q_\varphi$ are the fundamental $U(1)_Y$ "hypercharges" of the respective fields. The theory is invariant under the action of $e^{i\theta} \in U(1)_Y$ and $U \in SU(2)_L$ acting as

$$\psi \to e^{iq_\psi \theta}\,\psi, \qquad \psi^\dagger \to \psi^\dagger e^{-iq_\psi \theta},$$
$$\chi \to e^{iq_\chi \theta}\,U\chi, \qquad \chi^\dagger \to \chi^\dagger\,U^{-1}e^{-iq_\chi \theta},$$
$$\varphi \to e^{iq_\varphi \theta}\,U\varphi, \qquad \varphi^\dagger \to \varphi^\dagger\,U^{-1}e^{-iq_\varphi \theta},$$
$$B_\mu \to B_\mu + \frac{1}{g_y}\partial_\mu \theta, \qquad W_\mu \to UW_\mu U^{-1} + \frac{i}{g_w}U\partial_\mu U^{-1}. \tag{6.87}$$

Finally, the Higgs potential is taken to be

$$V(\varphi^\dagger \varphi) = -\hbar c\mu^2 \varphi^\dagger \varphi + \hbar c\lambda\,(\varphi^\dagger \varphi)^2. \tag{6.88}$$

The Higgs potential has a minima at $\varphi^\dagger \varphi = v^2/2$, where $v = \mu/\sqrt{\lambda}$, and spontaneously breaks $SU(2)_L \times U(1)_Y$ down to a subgroup $U(1)$. The system spontaneously picks from one of the minima for its ground state, which we take to be $\varphi_0 = v/\sqrt{2}\,\hat{\varphi}_0$, where $\hat{\varphi}_0 = (0, 1)$.

### 6.5.2  Electromagnetic U(1)

To inspect the residual $U(1)$ symmetry after spontaneous symmetry breaking, we use the exponential parametrisation of $SU(2)_L$ and identify the subgroup of $SU(2)_L \times U(1)_Y$ transformations that leave

the ground state invariant

$$\exp(iq_\varphi\theta)\exp\left(\frac{i}{\hbar}\theta^A T_A\right)\varphi_0 = \varphi_0 \quad \Longrightarrow \quad \left(q_\varphi\theta\,\mathbb{1} + \frac{1}{\hbar}\theta^A T_A\right)\varphi_0 = 0$$

$$\Longrightarrow \quad \theta^1 = \theta^2 = 0, \qquad \theta^3 = 2q_\varphi\theta. \tag{6.89}$$

Setting these parameters as such, this transformation acts on the remaining fields as

$$\psi \to \exp(iq_\psi\theta)\,\psi, \qquad \psi^\dagger \to \psi^\dagger\exp(-iq_\psi\theta),$$

$$\chi \to \exp\left(i\theta(q_\chi\mathbb{1} + q_\varphi\sigma_3)\right)\chi, \qquad \chi^\dagger \to \chi^\dagger\exp\left(-i\theta(q_\chi\mathbb{1} + q_\varphi\sigma_3)\right),$$

$$\eta \to \eta,$$

$$B_\mu \to B_\mu + \frac{1}{g_y}\partial_\mu\theta, \qquad W_\mu \to \exp(iq_\varphi\theta\sigma_3)W_\mu\exp(-iq_\varphi\theta\sigma_3) + \frac{q_\varphi}{g_w}\partial_\mu\theta\,\sigma_3. \tag{6.90}$$

In components, it results in

$$\psi \to \exp(iq_\psi\theta)\,\psi, \qquad \psi^\dagger \to \psi^\dagger\exp(-iq_\psi\theta),$$

$$\chi_1 \to \exp\left(i\theta(q_\chi + q_\varphi)\right)\chi_1, \qquad \chi_1^\dagger \to \chi_1^\dagger\exp\left(-i\theta(q_\chi + q_\varphi)\right),$$

$$\chi_2 \to \exp\left(i\theta(q_\chi - q_\varphi)\right)\chi_2, \qquad \chi_2^\dagger \to \chi_2^\dagger\exp\left(-i\theta(q_\chi - q_\varphi)\right),$$

$$\eta \to \eta,$$

$$W_\mu^\pm \to \exp\left(\pm 2iq_\varphi\theta\right)W_\mu^\pm, \qquad Z_\mu \to Z_\mu, \qquad A_\mu \to A_\mu + \frac{1}{e}\partial_\mu\theta. \tag{6.91}$$

where we have defined

$$W_\mu^\pm = \frac{1}{\sqrt{2}}\left(W_\mu^1 \mp iW_\mu^2\right), \qquad \begin{pmatrix} A_\mu \\ Z_\mu \end{pmatrix} = \begin{pmatrix} \cos\theta_w & \sin\theta_w \\ -\sin\theta_w & \cos\theta_w \end{pmatrix}\begin{pmatrix} B_\mu \\ W_\mu^3 \end{pmatrix}, \tag{6.92}$$

with $\theta_w = \arctan(2q_\varphi g_y/g_w)$ being the weak mixing angle and $e = g_y\cos\theta_w$ being the fundamental dimensionless electronic charge (electromagnetic coupling constant). We can read out the electronic charges of various fields in terms of the hypercharges

$$q_\psi^e = q_\psi, \qquad q_{\chi_1}^e = q_\chi + q_\varphi, \qquad q_{\chi_2}^e = q_\chi - q_\varphi, \qquad q_\eta^e = 0,$$

$$q_{W^\pm}^e = \pm 2q_\varphi, \qquad q_Z^e = q_A^e = 0, \tag{6.93}$$

while the Hermitian conjugate fields have opposite charges. We note that the residual Higgs field $\eta$, as well as gauge fields $Z_\mu$ and $A_\mu$, are electrically neutral. On the other hand, the $W_\mu^\pm$ gauge fields are electrically charged. The electric charges of the remaining fields follow the constraint

$$q_{\chi_1}^e - q_{\chi_2}^e = q_{W^+}^e = -q_{W^-}^e. \tag{6.94}$$

We shall require the electromagnetic U(1) symmetry to preserve chirality. This means that right- and $(SU(2)_L$ components of) left-handed spinors must either come in pairs of same electric charge, or their respective electric charge must be zero. This implies three possibilities: either there are two right-handed spinors $\psi = \psi_1, \psi_2$ corresponding to the left-handed spinor $\chi_{I=1,2}$ such that

$$q_{\chi_1}^e = q_{\psi_1}^e \quad \Longrightarrow \quad q_{\psi_1} = q_\chi + q_\varphi,$$

$$q_{\chi_2}^e = q_{\psi_2}^e \quad \Longrightarrow \quad q_{\psi_2} = q_\chi - q_\varphi. \tag{6.95}$$

These fields model quarks; see section 7. Or, the field $\chi_1$ has electric charge zero, while the field $\psi$ couples to $\chi_2$ such that

$$
\begin{aligned}
q^e_{\chi_1} = 0 &\implies q_\chi = -q_\varphi, \\
q^e_{\chi_2} = q^e_\psi &\implies q_\psi = q_\chi - q_\varphi = -2q_\varphi.
\end{aligned}
\tag{6.96}
$$

This is the case of leptons; see section 7. Or, finally, the field $\chi_2$ has electric charge zero, while the field $\psi$ couples to $\chi_1$ such that

$$
\begin{aligned}
q^e_{\chi_1} = q^e_\psi &\implies q_\psi = q_\chi + q_\varphi = 2q_\varphi, \\
q^e_{\chi_2} = 0 &\implies q_\chi = q_\varphi.
\end{aligned}
\tag{6.97}
$$

The second and third cases are equivalent up to $q_\varphi \to -q_\varphi$.

### 6.5.3 Higgs mechanism and massive gauge fields

To see the repercussions of spontaneous symmetry breaking on the electroweak Lagrangian, we can proceed in the manner similar to the one employed in section 6.4.2 and decompose the Higgs field as

$$
\varphi(x) = \frac{1}{\sqrt{2}}(v + \eta(x)) \, \exp(iq_\varphi \pi(x)) \exp\left(\frac{i}{\hbar}\pi^A(x)T_A\right) \hat{\varphi}_0, \qquad \hat{\varphi}_0 = \begin{pmatrix} 0 \\ 1 \end{pmatrix},
\tag{6.98}
$$

and perform a field redefinition

$$
\begin{aligned}
\psi(x) &\to \exp\left(iq_\psi \pi(x)\right) \psi(x), \\
\chi(x) &\to \exp\left(iq_\chi \pi(x)\right) \exp\left(\frac{i}{\hbar}\pi^A(x)T_A\right) \chi(x), \\
B_\mu(x) &\to B_\mu(x) + \frac{1}{g_y}\partial_\mu \pi(x), \\
W_\mu(x) &\to \exp\left(\frac{i}{\hbar}\pi^A(x)T_A\right)\left(W_\mu(x) + \frac{i}{g_w}\partial_\mu\right)\exp\left(-\frac{i}{\hbar}\pi^A(x)T_A\right).
\end{aligned}
\tag{6.99}
$$

The amounts to a symmetry transformation of the Lagrangian with $\theta = \pi$ and $U = \exp(i/\hbar\,\pi^A T_A)$, and only has the effect that $\varphi \to (v + \eta)/\sqrt{2}$. We are left with

$$
\begin{aligned}
\frac{1}{\hbar c}\mathcal{L} = {}& \frac{i}{2}\left(\psi^\dagger\bar{\sigma}^\mu D_\mu\psi - D_\mu\psi^\dagger\bar{\sigma}^\mu\psi\right) + \frac{i}{2}\left(\chi^\dagger\sigma^\mu D_\mu\chi - D_\mu\chi^\dagger\sigma^\mu\chi\right) \\
& - \frac{1}{2}\partial_\mu\eta\partial^\mu\eta - \frac{1}{2}(v+\eta)^2\hat{\varphi}_0\left(q_\varphi g_y B_\mu \mathbb{1} + g_w W_\mu\right)\left(q_\varphi g_y B^\mu \mathbb{1} + g_w W^\mu\right)\hat{\varphi}_0 - \frac{1}{\hbar c}V((v+\eta)^2/2) \\
& - \frac{1}{2}\operatorname{tr}(W^{\mu\nu}W_{\mu\nu}) - \frac{1}{4}B^{\mu\nu}B_{\mu\nu},
\end{aligned}
\tag{6.100}
$$

which expands to give

$$
\begin{aligned}
\frac{1}{\hbar c}\mathcal{L} ={}& \frac{i}{2}\left(\psi^\dagger\bar{\sigma}^\mu\tilde{\mathrm{D}}_\mu\psi - \tilde{\mathrm{D}}_\mu\psi^\dagger\bar{\sigma}^\mu\psi\right) + \frac{i}{2}\left(\chi_1^\dagger\sigma^\mu\tilde{\mathrm{D}}_\mu\chi_1 - \tilde{\mathrm{D}}_\mu\chi_1^\dagger\sigma^\mu\chi_1\right) + \frac{i}{2}\left(\chi_2^\dagger\sigma^\mu\tilde{\mathrm{D}}_\mu\chi_2 - \tilde{\mathrm{D}}_\mu\chi_2^\dagger\sigma^\mu\chi_2\right) \\
& - \left(\lambda_\psi\psi^\dagger\bar{\sigma}^\mu\psi + \lambda_{\chi1}\chi_1^\dagger\sigma^\mu\chi_1 + \lambda_{\chi2}\chi_2^\dagger\sigma^\mu\chi_2\right)Z_\mu + \frac{g_w}{\sqrt{2}}\left(W_\mu^+\chi_1^\dagger\sigma^\mu\chi_2 + W_\mu^-\chi_2^\dagger\sigma^\mu\chi_1\right) \\
& - \frac{1}{2}\partial_\mu\eta\partial^\mu\eta + \frac{\mu^2}{2}(v+\eta)^2 - \frac{\lambda}{4}(v+\eta)^4 - \frac{1}{4}g_w^2(v+\eta)^2\left(W_\mu^-W^{+\mu} + \frac{1}{2}\sec^2\theta_w\, Z^\mu Z_\mu\right) \\
& - \frac{1}{2}W_{\mu\nu}^+W^{-\mu\nu} - \frac{1}{4}Z_{\mu\nu}Z^{\mu\nu} - \frac{1}{4}F_{\mu\nu}F^{\mu\nu} \\
& - ig_w\cos\theta_w\left(W^{-\mu\nu}W_\mu^+Z_\nu - W^{+\mu\nu}W_\mu^-Z_\nu - Z^{\mu\nu}W_\mu^+W_\nu^-\right) + ig_w\sin\theta_w F^{\mu\nu}W_\mu^+W_\nu^- \\
& - g_w^2\cos^2\theta_w\left(W_\mu^+W^{-\mu}Z_\nu Z^\nu - W_\nu^+Z^\nu W_\mu^-Z^\mu\right) \\
& + \frac{1}{2}g_w^2\left(W_\mu^+W^{+\mu}W_\nu^-W^{-\nu} - (W_\mu^+W^{-\mu})^2\right),
\end{aligned}
\tag{6.101}
$$

where

$$
\lambda_\psi = q_\psi g_y\sin\theta_w, \qquad \lambda_{\chi1} = q_\chi g_y\sin\theta_w - \frac{1}{2}g_w\cos\theta_w, \qquad \lambda_{\chi2} = q_\chi g_y\sin\theta_w + \frac{1}{2}g_w\cos\theta_w. \tag{6.102}
$$

In eq. (6.101), we have utilised the electromagnetic covariant derivatives of all charged fields

$$
\begin{aligned}
\tilde{\mathrm{D}}_\mu\psi &= \partial_\mu\psi - iq_\psi^e eA_\mu\psi, \\
\tilde{\mathrm{D}}_\mu\chi_1 &= \partial_\mu\chi_1 - iq_{\chi1}^e eA_\mu\chi_1, \\
\tilde{\mathrm{D}}_\mu\chi_2 &= \partial_\mu\chi_2 - iq_{\chi2}^e eA_\mu\chi_2, \\
\tilde{\mathrm{D}}_\mu W_\nu^\pm &= \partial_\mu W_\nu^\pm \mp 2iq_\varphi eA_\mu W_\nu^\pm,
\end{aligned}
\tag{6.103}
$$

along with field strengths $W_{\mu\nu}^\pm = \tilde{\mathrm{D}}_\mu W_\nu^\pm - \tilde{\mathrm{D}}_\nu W_\mu^\pm$, $Z_{\mu\nu} = \partial_\mu Z_\nu - \partial_\nu Z_\mu$, and $F_{\mu\nu} = \partial_\mu A_\nu - \partial_\nu A_\mu$. We have used the following identities

$$
\begin{aligned}
\mathrm{D}_\mu\psi &= \tilde{\mathrm{D}}_\mu\psi + iq_\psi g_y\sin\theta_w Z_\mu\psi, \\
(\mathrm{D}_\mu\chi)_1 &= \tilde{\mathrm{D}}_\mu\chi_1 + \frac{i}{2}\left(2q_\chi g_y\sin\theta_w - g_w\cos\theta_w\right)Z_\mu\chi_1 - \frac{ig_w}{\sqrt{2}}W_\mu^+\chi_2, \\
(\mathrm{D}_\mu\chi)_2 &= \tilde{\mathrm{D}}_\mu\chi_2 + \frac{i}{2}\left(2q_\chi g_y\sin\theta_w + g_w\cos\theta_w\right)Z_\mu\chi_2 - \frac{ig_w}{\sqrt{2}}W_\mu^-\chi_1, \\
W_{\mu\nu}^1 &= \frac{1}{\sqrt{2}}\left(W_{\mu\nu}^+ + W_{\mu\nu}^- + 2ig_w\left(W_{[\mu}^+Z_{\nu]} - W_{[\mu}^-Z_{\nu]}\right)\cos\theta_w\right), \\
W_{\mu\nu}^2 &= \frac{i}{\sqrt{2}}\left(W_{\mu\nu}^+ - W_{\mu\nu}^- + 2ig_w\left(W_{[\mu}^+Z_{\nu]} + W_{[\mu}^-Z_{\nu]}\right)\cos\theta_w\right), \\
W_{\mu\nu}^3 &= Z_{\mu\nu}\cos\theta_w + F_{\mu\nu}\sin\theta_w - 2ig_w W_{[\mu}^+W_{\nu]}^-, \\
B_{\mu\nu} &= F_{\mu\nu}\cos\theta_w - Z_{\mu\nu}\sin\theta_w.
\end{aligned}
\tag{6.104}
$$

The full form of the Lagrangian is not terribly useful. To get some physical intuition, let us

ignore all cubic and quartic interactions, which leads to

$$\frac{1}{\hbar c}\mathcal{L} = \frac{i}{2}\left(\psi^\dagger\bar{\sigma}^\mu\partial_\mu\psi - \partial_\mu\psi^\dagger\bar{\sigma}^\mu\psi\right) + \frac{i}{2}\left(\chi_1^\dagger\sigma^\mu\partial_\mu\chi_1 - \partial_\mu\chi_1^\dagger\sigma^\mu\chi_1\right) + \frac{i}{2}\left(\chi_2^\dagger\sigma^\mu\partial_\mu\chi_2 - \partial_\mu\chi_2^\dagger\sigma^\mu\chi_2\right)$$

$$- \frac{1}{2}W_{\mu\nu}^-W^{+\mu\nu} - \frac{m_W^2 c^2}{\hbar^2}W_\mu^- W^{+\mu} - \frac{1}{4}Z_{\mu\nu}Z^{\mu\nu} - \frac{m_Z^2 c^2}{2\hbar^2}Z^\mu Z_\mu - \frac{1}{4}F_{\mu\nu}F^{\mu\nu}$$

$$- \frac{1}{2}\partial_\mu\eta\partial^\mu\eta - \mu^2\eta^2 + \text{interactions.} \tag{6.105}$$

We observe that the gauge field components $W_\mu^\pm$ and $Z_\mu$ acquire masses

$$m_W = \frac{v\hbar}{2c}g_w, \qquad m_Z = \frac{v\hbar}{2c}\sqrt{g_w^2 + 4q_\varphi^2 g_y^2}, \tag{6.106}$$

while the gauge field $A_\mu$ is massless. The gauge field combinations $W_\mu^\pm$, $Z_\mu$ can be thought of as the carriers of weak force, while $A_\mu$ can be identified with the electromagnetic photon. We are also left with a residual massive real scalar Higgs field $\eta$, with mass $\sqrt{2}\mu\hbar/c$.

### 6.5.4   Yukawa couplings

So far, our model of electroweak forces does not give masses to various spinor fields. We already know that Weyl spinors cannot have their own mass terms. The regular Dirac mass term

$$-mc^2\bar{\Psi}\Psi = -mc^2\left(\chi^\dagger\psi + \psi^\dagger\chi\right), \qquad \text{where} \quad \Psi = \begin{pmatrix}\psi\\\chi\end{pmatrix}, \tag{6.107}$$

is also disallowed by $\text{SU}(2)_L$-invariance, which only acts on $\chi$ and not on $\psi$. As it turns out, such a term can indeed be generated via spontaneous symmetry breaking. To wit, we can add the so-called Yukawa couplings to the Lagrangian

$$\frac{1}{\hbar c}\mathcal{L} = \frac{i}{2}\left(\psi^\dagger\bar{\sigma}^\mu D_\mu\psi - D_\mu\psi^\dagger\bar{\sigma}^\mu\psi\right) + \frac{i}{2}\left(\chi^\dagger\sigma^\mu D_\mu\chi - D_\mu\chi^\dagger\sigma^\mu\chi\right) - D_\mu\varphi^\dagger D^\mu\varphi - \frac{1}{\hbar c}V(\varphi^\dagger\varphi)$$

$$- \frac{1}{2}\text{tr}(W^{\mu\nu}W_{\mu\nu}) - \frac{1}{4}B^{\mu\nu}B_{\mu\nu} - y\left(\chi^\dagger\varphi\psi + \psi^\dagger\varphi^\dagger\chi\right) - y'\left(\chi^\dagger\tilde{\varphi}\psi + \psi^\dagger\tilde{\varphi}^\dagger\chi\right), \tag{6.108}$$

with arbitrary Yukawa coupling constants $y$ and $y'$. Here $\tilde{\varphi}_I = \epsilon_{IJ}(\varphi^\dagger)^J$ is the conjugate Dirac field. It transforms under the $\text{SU}(2)_L \times \text{U}(1)_Y$ transformation as

$$\tilde{\varphi} \to \exp(-iq_\varphi\theta)\,U\tilde{\varphi}, \qquad \tilde{\varphi}^\dagger \to \tilde{\varphi}\,U^{-1}\exp(iq_\varphi\theta). \tag{6.109}$$

It immediately follows that the two Yukawa coupling terms are $\text{SU}(2)_L$-invariant. However, for these to be $\text{U}(1)_Y$-invariant, the respective hypercharges must be constrained

$$y \neq 0 \quad \text{if} \quad q_\psi = q_\chi - q_\varphi, \qquad\qquad y' \neq 0 \quad \text{if} \quad q_\psi = q_\chi + q_\varphi. \tag{6.110}$$

Or in the electromagnetic terms

$$y \neq 0 \quad \text{if} \quad q_\psi^e = q_{\chi_2}^e, \qquad\qquad y' \neq 0 \quad \text{if} \quad q_\psi^e = q_{\chi_1}^e. \tag{6.111}$$

Note that, as long as $q_\varphi \neq 0$, both these Yukawa couplings cannot be turned on at the same time for a given right-handed spinor $\psi$. Note also that the Yukawa couplings force us to set the same electric

charges for the left- and right-handed spinors. After spontaneous symmetry breaking, Yukawa couplings generate masses for the spinor fields

$$\frac{1}{\hbar c}\mathcal{L} = \frac{i}{2}\left(\psi^\dagger\bar{\sigma}^\mu\partial_\mu\psi - \partial_\mu\psi^\dagger\bar{\sigma}^\mu\psi\right) + \frac{i}{2}\left(\chi_1^\dagger\sigma^\mu\partial_\mu\chi_1 - \partial_\mu\chi_1^\dagger\sigma^\mu\chi_1\right) + \frac{i}{2}\left(\chi_2^\dagger\sigma^\mu\partial_\mu\chi_2 - \partial_\mu\chi_2^\dagger\sigma^\mu\chi_2\right)$$
$$- \frac{1}{2}W_{\mu\nu}^-W^{+\mu\nu} - \frac{m_W^2c^2}{\hbar^2}W_\mu^-W^{+\mu} - \frac{1}{4}Z_{\mu\nu}Z^{\mu\nu} - \frac{m_Z^2c^2}{2\hbar^2}Z^\mu Z_\mu - \frac{1}{4}F_{\mu\nu}F^{\mu\nu}$$
$$- \frac{1}{2}\partial_\mu\eta\partial^\mu\eta - \mu^2\eta^2 - \frac{m_{\chi_2}c}{\hbar}\left(\chi_2^\dagger\psi + \psi^\dagger\chi_2\right) - \frac{m_{\chi_1}c}{\hbar}\left(\chi_1^\dagger\psi + \psi^\dagger\chi_1\right) + \text{int.}, \qquad (6.112)$$

where

$$m_{\chi_1} = \frac{vy'\hbar}{\sqrt{2}\,c}, \qquad m_{\chi_2} = \frac{vy\hbar}{\sqrt{2}\,c}. \qquad (6.113)$$

If $y \neq 0$, the Yukawa coupling gives the same mass to $\chi_2$ and $\psi$ spinors, while leaving $\chi_1$ massless. On the other hand, if $y' \neq 0$, the spinors $\chi_1$ and $\psi$ are massive, while the $\chi_2$ is massless. If we intend to give masses to both $\chi_1$ and $\chi_2$, we will need to introduce two right-handed spinors $\psi_1$ and $\psi_2$ with the same respective electric charges.

### 6.5.5   Interactions

The final observation we want to make are the allowed electroweak interactions. Due to spontaneous symmetry breaking, the interaction structure of electroweak forces is far richer than QED or QCD. Looking at the spontaneously broken Lagrangian in eq. (6.101), we first have the electromagnetic and weak interactions of the spinor fields coming from the first and second lines

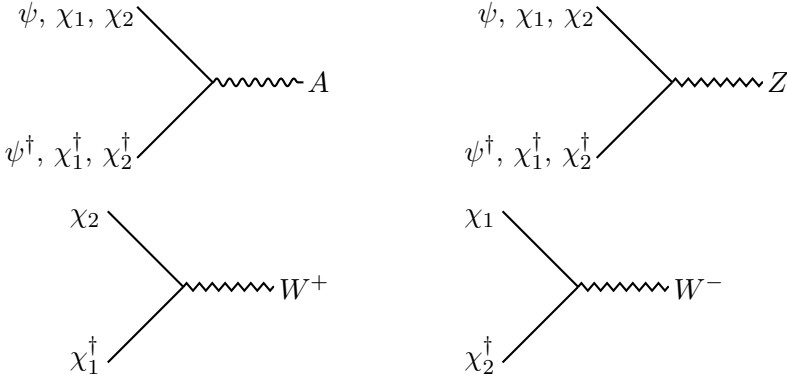

In particular, we note that $\chi_1$ spinors (left-handed neutrino/up-quark) can turn into $\chi_2$ spinors (left-handed electron/down-quark), and vice-versa, by absorbing/emitting charged weak bosons $W_\mu^\pm$.

Then, we have the self and weak interactions of the residual Higgs field coming from the third line

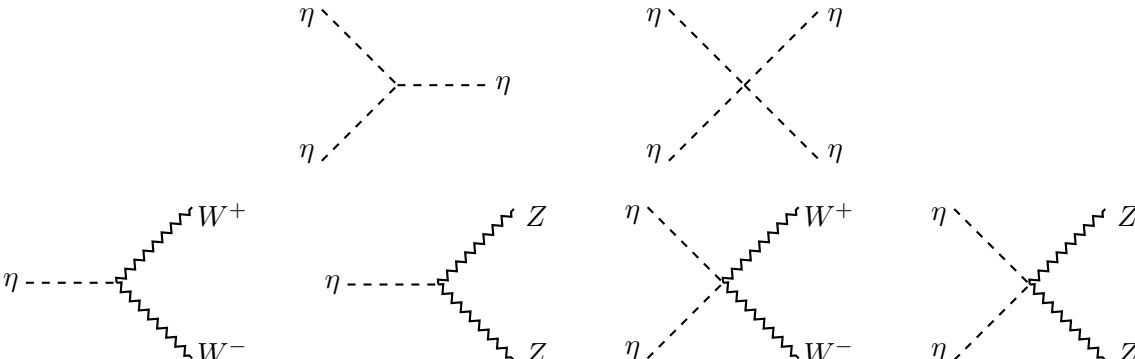

In addition, we have mutual interactions of various gauge fields. We have mixed interactions in the electromagnetic and weak sector coming from the U(1) covariant derivatives in the forth line, and a few terms in the fifth line

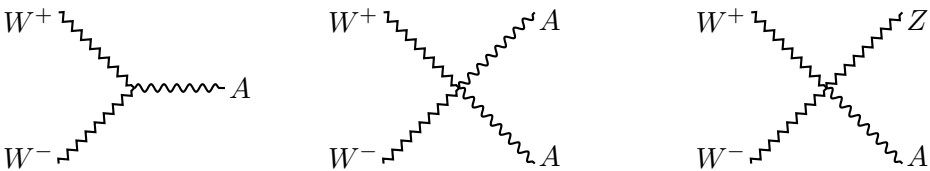

Then, we have weak force self-interactions coming from the last three lines

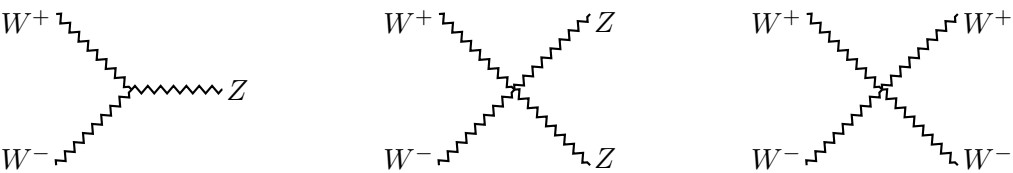

Finally, we have Yukawa interactions between spinors and Higgs field

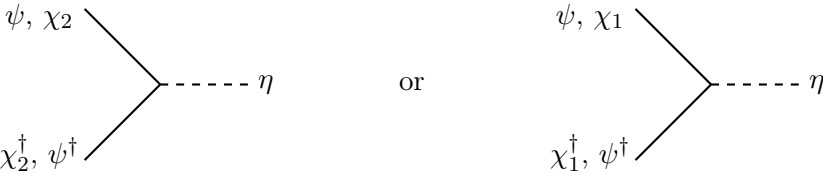

For a given spinor $\psi$, only one of the above interactions can be present in a model, depending on the hypercharge. These Yukawa interactions mix the left- and right-handed spinor sectors and are only present if the respective spinors admit a Yukawa mass term.

# 7 | Standard Model of particle physics

We now have all the requisite tools to write down the Standard Model of particle physics. At the time Standard Model was proposed, neutrinos were known to be massless. For the present discussion, we will assume as such. Discussion of neutrino masses and neutrino flavor oscillations goes somewhat beyond the minimal Standard Model and will not be covered in this course. The discussion in this section follows the standard texts on the Standard Model of particle physics such as [8].

The local internal symmetry group of the Standard Model is a direct product of quantum chromodynamics symmetry and electroweak symmetry, i.e. $SU(3) \times SU(2)_L \times U(1)_Y$. On the other hand, the spacetime symmetry group is still Poincaré. The model employs the Higgs mechanism in the electroweak sector to spontaneously break the symmetry down to $SU(3) \times SU(2)_L \times U(1)_Y \rightarrow SU(3) \times U(1)$, where $U(1)$ corresponds to the residual $U(1)$ electromagnetic transformation. In the process, we generate masses for $W_\mu^\pm$ and $Z_\mu$ gauge bosons mediating weak forces. Prior to symmetry breaking, the QCD symmetry $SU(3)$ preserves chirality, i.e. acts uniformly on left- and right-handed spinors, however the electroweak symmetry $SU(2)_L \times U(1)_Y$ violates it. Interestingly, post symmetry breaking, the residual symmetry group $SU(3) \times U(1)$ preserves chirality.

## 7.1 Field content and symmetries

The field content of Standard Model, pre-symmetry breaking, consists of gauge bosons associated with the local symmetry group $SU(3) \times SU(2)_L \times U(1)_Y$, i.e. $(G_\mu)_i{}^j = 1/\hbar\, G_\mu^a (T_a)_i{}^j$, $(W_\mu)_I{}^J = 1/\hbar\, W_\mu^A (T_A)_I{}^J$, and $B_\mu$ respectively, coupled to matter fields. Here $\mu, \nu, \ldots = 0, 1, 2, 3$ are Lorentz indices, $a, b, \ldots = 1, \ldots, 8$ are $\mathfrak{su}(3)$ indices, $i, j, \ldots = 1, 2, 3$ are $SU(3)$ indices, $A, B, \ldots = 1, 2, 3$ are $\mathfrak{su}(2)_L$ indices, while $I, J, \ldots = 1, 2$ are $SU(2)_L$ indices. Here $T_a = \hbar/2\, \lambda_a$, with $\lambda_a$ being the Gell-Mann matrices, and $T_A = \hbar/2\, \sigma_A$, with $\sigma_A$ being the Pauli matrices.

The matter fields can be broadly classified into three sectors: quarks, leptons, and Higgs. The quark sector contains a left-handed Weyl spinor $(Q_m^L)_{Ii}^{\dot\alpha}$, transforming in the fundamental (triplet) representation of $SU(3)$, fundamental (doublet) representation of $SU(2)_L$, and carrying hypercharge $1/6$ under $U(1)_Y$. We denote the respective $SU(2)_L$ components as $(Q_m^L)_{1i}^{\dot\alpha} = (u_m^L)_i^{\dot\alpha}$ and $(Q_m^L)_{2i}^{\dot\alpha} = (d_m^L)_i^{\dot\alpha}$. In addition, we have two right-handed Weyl spinors $u_{m\alpha i}^R$ and $d_{m\alpha i}^R$ transforming in the fundamental (triplet) representation of $SU(3)$, trivial (singlet) representation of $SU(2)_L$, and carrying $U(1)_Y$ hypercharges $2/3$ and $-1/3$ respectively. Here $\alpha, \beta, \ldots = 1, 2$ and $\dot\alpha, \dot\beta, \ldots = 1, 2$ are right- and left-handed Weyl spinor indices respectively. All the quarks come in 3 *generations*, labelled by the index $m = 1, 2, 3$. Dropping all spinor and group indices, these are identified with $u = u_1$ (up), $d = d_1$ (down), $c = u_2$ (charm), $s = d_2$ (strange), $t = u_3$ (top), and $b = d_3$ (bottom) quarks observed in nature. Anti-quarks are given by the respective Hermitian conjugate fields.

In the lepton sector, we have a a left-handed Weyl spinor $(L_m^L)_I^{\dot\alpha}$, transforming in the trivial (singlet) representation of $SU(3)$, fundamental (doublet) representation of $SU(2)_L$, and carrying hypercharge $-1/2$ under $U(1)_Y$. We denote the respective $SU(2)_L$ components as $(L_m^L)_1^{\dot\alpha} = (\nu_m^L)^{\dot\alpha}$ and $(L_m^L)_2^{\dot\alpha} = (e_m^L)^{\dot\alpha}$. In addition, we have a right-handed Weyl spinor $e_{m\alpha}^R$ transforming in the singlet (trivial) representations of both $SU(3)$ and $SU(2)_L$, and carrying $U(1)_Y$ hypercharge $-1$. Leptons also come in 3 *generations*. Dropping all spinor and group indices, these are identified with $e = e_1$ (electron), $\nu_e = \nu_1$ (electron neutrino), $\mu = e_2$ (muon), $\nu_\mu = \nu_2$ (muon neutrino), $\tau = e_3$ (tau), $\nu_\tau = \nu_3$ (tau neutrino). Importantly, there are no right-handed neutrinos in the Standard Model,

| Particle | SU(3) | SU(2)$_L$ | U(1)$_Y$ | SO$^+$(3,1) |
|:---:|:---:|:---:|:---:|:---:|
| **Gauge sector** | | | | |
| $G_\mu^a$ | **8** | **1** | $0$ | $(\frac{1}{2},\frac{1}{2})$ |
| $W_\mu^A$ | **1** | **3** | $0$ | $(\frac{1}{2},\frac{1}{2})$ |
| $B_\mu$ | **1** | **1** | $0$ | $(\frac{1}{2},\frac{1}{2})$ |
| **Quark sector** | | | | |
| $(Q_m^L)_{Ii}^{\dot\alpha} = \begin{pmatrix}(u_m^L)_i^{\dot\alpha}\\(d_m^L)_i^{\dot\alpha}\end{pmatrix}$ | **3** | **2** | $1/6$ | $(\frac{1}{2},0)$ |
| $(u_m^R)_{\alpha i}$ | **3** | **1** | $2/3$ | $(0,\frac{1}{2})$ |
| $(d_m^R)_{\alpha i}$ | **3** | **1** | $-1/3$ | $(0,\frac{1}{2})$ |
| $(Q_m^{L\dagger})^{\alpha Ii} = \begin{pmatrix}(u_m^{L\dagger})^{\alpha i} & (d_m^{L\dagger})^{\alpha i}\end{pmatrix}$ | **3̄** | **2** | $-1/6$ | $(0,\frac{1}{2})$ |
| $(u_m^{R\dagger})_{\dot\alpha}^i$ | **3̄** | **1** | $-2/3$ | $(\frac{1}{2},0)$ |
| $(d_m^{R\dagger})_{\dot\alpha}^i$ | **3̄** | **1** | $1/3$ | $(\frac{1}{2},0)$ |
| **Lepton sector** | | | | |
| $(L_m^L)_I^{\dot\alpha} = \begin{pmatrix}(\nu_m^L)^{\dot\alpha}\\(e_m^L)^{\dot\alpha}\end{pmatrix}$ | **1** | **2** | $-1/2$ | $(\frac{1}{2},0)$ |
| $(e_m^R)_\alpha$ | **1** | **1** | $-1$ | $(0,\frac{1}{2})$ |
| $(L_m^{L\dagger})^{\alpha I} = \begin{pmatrix}(\nu_m^{L\dagger})^\alpha & (e_m^{L\dagger})^\alpha\end{pmatrix}$ | **1** | **2̄** | $1/2$ | $(0,\frac{1}{2})$ |
| $(e_m^{R\dagger})_{\dot\alpha}$ | **1** | **1** | $1$ | $(\frac{1}{2},0)$ |
| **Higgs sector** | | | | |
| $\varphi_I = \begin{pmatrix}\varphi^+\\\varphi^0\end{pmatrix}$ | **1** | **2** | $1/2$ | $(0,0)$ |
| $(\varphi^\dagger)^I = \begin{pmatrix}\varphi^{+*} & \varphi^{0*}\end{pmatrix}$ | **1** | **2̄** | $-1/2$ | $(0,0)$ |
| $\tilde\varphi_I = \epsilon_{IJ}(\varphi^\dagger)^I = \begin{pmatrix}\varphi^{0*}\\-\varphi^{+*}\end{pmatrix}$ | **1** | **2** | $-1/2$ | $(0,0)$ |
| $(\tilde\varphi^\dagger)^I = \epsilon^{IJ}\varphi_J = \begin{pmatrix}\varphi^0 & -\varphi^+\end{pmatrix}$ | **1** | **2̄** | $1/2$ | $(0,0)$ |

**Table 1:** Field content of the Standard Model of particle physics. The SU(3) and SU(2)$_L$ columns contain the dimensions of the representations respectively, while the U(1)$_Y$ column contains the hypercharges. The SO$^+$(3,1) column contains the highest-weights $(j_-, j_+)$ of the respective fields.

which is a vital assumption that renders the Standard Model neutrinos massless. Anti-leptons are given by the respective Hermitian conjugate fields.

Finally, in the Higgs sector, we have a single scalar Higgs field $\varphi_I$, transforming in the trivial (singlet) representation of SU(3), fundamental (doublet) representation of SU(2)$_L$, and carrying hypercharge 1/2 under U(1)$_Y$. The Higgs field is assumed to have a spontaneous symmetry breaking potential. It is also convenient to define a conjugate Higgs field $\tilde{\varphi}_I = \epsilon_{IJ}(\varphi^\dagger)^I$, which also transforms in the trivial (singlet) representation of SU(3) and fundamental (doublet) representation of SU(2)$_L$, but carries hypercharge $-1/2$ under U(1)$_Y$. Anti-Higgs and conjugate anti-Higgs fields are given by the respective Hermitian conjugates. We have summarised the particle content in table 1.

The symmetry transformations act on various fields according to the representations stated above. The global Poincaré transformations $(\Lambda^\mu{}_\nu, a^\mu)$ act as

$$
\begin{aligned}
G_\mu^a &\to (\Lambda^{-1})^\nu{}_\mu\, G_\nu^a, \qquad W_\mu^A \to (\Lambda^{-1})^\nu{}_\mu\, W_\nu^A, \qquad B_\mu \to (\Lambda^{-1})^\nu{}_\mu\, B_\nu, \\
(Q_m^L)_{Ii}^{\dot\alpha} &\to \bar{A}^{\dot\alpha}{}_{\dot\beta}\,(Q_m^L)_{Ii}^{\dot\beta}, \qquad (Q_m^{L\dagger})^{\alpha Ii} \to (Q_m^{L\dagger})^{\beta Ii}\,(A^{-1})_\beta{}^\alpha, \\
(u_m^R)_{\alpha i} &\to A_\alpha{}^\beta\,(u_m^R)_{\beta i}, \qquad (u_m^{R\dagger})_{\dot\alpha}^i \to (u_m^{R\dagger})_{\dot\beta}^i\,(\bar{A}^{-1})^{\dot\beta}{}_{\dot\alpha}, \\
(d_m^R)_{\alpha i} &\to A_\alpha{}^\beta\,(d_m^R)_{\beta i}, \qquad (d_m^{R\dagger})_{\dot\alpha}^i \to (d_m^{R\dagger})_{\dot\beta}^i\,(\bar{A}^{-1})^{\dot\beta}{}_{\dot\alpha}, \\
(L_m^L)_I^{\dot\alpha} &\to \bar{A}^{\dot\alpha}{}_{\dot\beta}\,(L_m^L)_I^{\dot\beta}, \qquad (L_m^{L\dagger})^{\alpha I} \to (L_m^{L\dagger})^{\beta I}\,(A^{-1})_\beta{}^\alpha, \\
(e_m^R)_\alpha &\to A_\alpha{}^\beta\,(e_m^R)_\beta, \qquad (e_m^{R\dagger})_{\dot\alpha} \to (e_m^{R\dagger})_{\dot\beta}\,(\bar{A}^{-1})^{\dot\beta}{}_{\dot\alpha}, \\
\varphi_I &\to \varphi_I, \qquad (\varphi^\dagger)^I \to (\varphi^\dagger)^I, \qquad \tilde{\varphi}_I \to \tilde{\varphi}_I, \qquad (\tilde{\varphi}^\dagger)^I \to (\tilde{\varphi}^\dagger)^I.
\end{aligned}
\tag{7.1}
$$

Here $A = D_{(1/2,0)}(\Lambda)$ and $\bar{A} = D_{(0,1/2)}(\Lambda)$. The spacetime arguments of all the fields also transform under Poincaré transformations as $x^\mu \to (\Lambda^{-1})^\mu{}_\nu x^\nu - a^\mu$, which we have suppressed in the expressions above for clarity. Note that Hermitian conjugates of (left-) right-handed spinors transform as (right-) left-handed anti-fundamental fermions. We have kept all the indices explicit to avoid any confusion. On the other hand, local $U \in$ SU(3) transformations act on various fields as

$$
\begin{aligned}
G_\mu^a(T_a)_i{}^j &\to U_i{}^k \left( G_\mu^a(T_a)_k{}^l + \frac{i\hbar}{g_s}\delta_k^l\partial_\mu \right) (U^{-1})_l{}^j, \\
(Q_m^L)_{Ii}^{\dot\alpha} &\to U_i{}^j\,(Q_m^L)_{Ij}^{\dot\alpha}, \qquad (Q_m^{L\dagger})^{\alpha Ii} \to (Q_m^{L\dagger})^{\alpha Ij}\,(U^{-1})_j{}^i, \\
(u_m^R)_{\alpha i} &\to U_i{}^j\,(u_m^R)_{\alpha j}, \qquad (u_m^{R\dagger})_{\dot\alpha}^i \to (u_m^{R\dagger})_{\dot\alpha}^j\,(U^{-1})_j{}^i, \\
(d_m^R)_{\alpha i} &\to U_i{}^j\,(d_m^R)_{\alpha j}, \qquad (d_m^{R\dagger})_{\dot\alpha}^i \to (d_m^{R\dagger})_{\dot\alpha}^j\,(U^{-1})_j{}^i,
\end{aligned}
\tag{7.2}
$$

while all other fields are SU(3)-invariant. Here $g_s$ is the strong force coupling constant. Similarly, local $V \in$ SU(2)$_L$ transformations act as

$$
\begin{aligned}
W_\mu^A(T_A)_I{}^J &\to V_I{}^K \left( W_\mu^A(T_A)_K{}^L + \frac{i\hbar}{g_w}\delta_K^L\partial_\mu \right) (V^{-1})_L{}^J, \\
(Q_m^L)_{Ii}^{\dot\alpha} &\to V_I{}^J\,(Q_m^L)_{Ji}^{\dot\alpha}, \qquad (Q_m^L)^{\alpha Ii} \to (Q_m^L)^{\alpha Ji}\,(V^{-1})_J{}^I, \\
(L_m^L)_I^{\dot\alpha}(x) &\to V_I{}^J\,(L_m^L)_J^{\dot\alpha}, \qquad (L_m^L)^{\alpha I} \to (L_m^L)^{\alpha J}\,(V^{-1})_J{}^I, \\
\varphi_I &\to V_I{}^J\,\varphi_J, \qquad (\varphi^\dagger)^I \to (\varphi^\dagger)^J\,(V^{-1})_J{}^I, \\
\tilde{\varphi}_I &\to V_I{}^J\,\tilde{\varphi}_J, \qquad (\tilde{\varphi}^\dagger)^I \to (\tilde{\varphi}^\dagger)^J\,(V^{-1})_J{}^I,
\end{aligned}
\tag{7.3}
$$

where $g_w$ is the weak force coupling constant. Finally, local $\mathrm{e}^{i\theta} \in \mathrm{U}(1)_Y$ transformations act as

$$B_\mu \to B_\mu + \frac{i}{g_y}\mathrm{e}^{i\theta}\partial_\mu \mathrm{e}^{-i\theta} = B_\mu + \frac{1}{g_y}\partial_\mu\theta,$$

$$(Q_m^L)_{Ii}^{\dot\alpha} \to \mathrm{e}^{i\theta/6}\,(Q_m^L)_{Ii}^{\dot\alpha}, \qquad (Q_m^{L\dagger})^{\alpha Ii} \to \mathrm{e}^{-i\theta/6}\,(Q_m^{L\dagger})^{\alpha Ii},$$

$$(u_m^R)_{\alpha i} \to \mathrm{e}^{2i\theta/3}\,(u_m^R)_{\alpha i}, \qquad (u_m^{R\dagger})_{\dot\alpha}^{i} \to \mathrm{e}^{-2i\theta/3}\,(u_m^{R\dagger})_{\dot\alpha}^{i},$$

$$(d_m^R)_{\alpha i} \to \mathrm{e}^{-i\theta/3}\,(d_m^R)_{\alpha i}, \qquad (d_m^{R\dagger})_{\dot\alpha}^{i} \to \mathrm{e}^{i\theta/3}\,(d_m^{R\dagger})_{\dot\alpha}^{i},$$

$$(L_m^L)_I^{\dot\alpha} \to \mathrm{e}^{-i\theta/2}\,(L_m^L)_I^{\dot\alpha}, \qquad (L_m^{L\dagger})^{\alpha I} \to \mathrm{e}^{i\theta/2}\,(L_m^{L\dagger})^{\alpha I},$$

$$(e_m^R)_\alpha \to \mathrm{e}^{-i\theta}\,(e_m^R)_\alpha, \qquad (e_m^{R\dagger})_{\dot\alpha} \to \mathrm{e}^{i\theta}\,(e_m^{R\dagger})_{\dot\alpha},$$

$$\varphi_I \to \mathrm{e}^{i\theta/2}\,\varphi_I, \qquad (\varphi^\dagger)^I \to \mathrm{e}^{-i\theta/2}\,(\varphi^\dagger)^I,$$

$$\tilde\varphi_I \to \mathrm{e}^{-i\theta/2}\,\tilde\varphi_I, \qquad (\tilde\varphi^\dagger)^I \to \mathrm{e}^{i\theta/2}\,(\tilde\varphi^\dagger)^I, \tag{7.4}$$

where $g_y$ is the hypercharge coupling constant. The Standard Model of particle physics is required to be invariant under all of these transformations.

## 7.2 Standard Model Lagrangian

We will now write down the Lagrangian for the Standard Model based on our discussion in the course. The Lagrangian will be required to be invariant under all the spacetime and internal symmetries mentioned above. The construction, pretty much, follows from our discussion in sections 6.3 and 6.5.

### 7.2.1 Gauge sector

The most straight-forward contribution to the Lagrangian comes from the gauge fields. We simply get the respective Yang-Mills terms

$$\begin{aligned}
\frac{1}{\hbar c}\mathcal{L}_{\text{gauge}} &= -\frac{1}{2}\operatorname{tr}(G_{\mu\nu}G^{\mu\nu}) - \frac{1}{2}\operatorname{tr}(W_{\mu\nu}W^{\mu\nu}) - \frac{1}{4}B_{\mu\nu}B^{\mu\nu} \\
&= -\frac{1}{4}G_{\mu\nu}^a G_a^{\mu\nu} - \frac{1}{4}W_{\mu\nu}^A W_A^{\mu\nu} - \frac{1}{4}B_{\mu\nu}B^{\mu\nu}.
\end{aligned} \tag{7.5}$$

Here, the field strengths are defined as

$$\begin{aligned}
G_{\mu\nu} &= \partial_\mu G_\nu - \partial_\nu G_\mu - ig_s[G_\mu, G_\nu], \\
W_{\mu\nu} &= \partial_\mu W_\nu - \partial_\nu W_\mu - ig_w[W_\mu, W_\nu], \\
B_{\mu\nu} &= \partial_\mu B_\nu - \partial_\nu B_\mu.
\end{aligned} \tag{7.6}$$

To avoid confusion, we also note these in components

$$\begin{aligned}
G_{\mu\nu}^a &= \partial_\mu G_\nu^a - \partial_\nu G_\mu^a + g_s G_\mu^b G_\nu^c f_{bca}, \\
W_{\mu\nu}^A &= \partial_\mu W_\nu^A - \partial_\nu W_\mu^A + g_w W_\mu^B W_\nu^C \epsilon_{BCA}, \\
B_{\mu\nu} &= \partial_\mu B_\nu - \partial_\nu B_\mu,
\end{aligned} \tag{7.7}$$

where $f_{abc}$ are the SU(3) structure constants, while the Levi-Civita $\epsilon_{ABC}$ functions as the structure constants of $\mathrm{SU}(2)_L$.

### 7.2.2   Higgs sector

The Lagrangian for the Higgs field comprises of the kinetic term with a SSB potential

$$\frac{1}{\hbar c}\mathcal{L}_{\text{higgs}} = -D_\mu\varphi^\dagger D^\mu\varphi + \mu^2\varphi^\dagger\varphi - \lambda(\varphi^\dagger\varphi)^2$$
$$= -(D_\mu\varphi^\dagger)^I(D^\mu\varphi)_I + \mu^2(\varphi^\dagger)^I\varphi_I - \lambda\left((\varphi^\dagger)^I\varphi_I\right)^2, \tag{7.8}$$

where $\mu$ and $\lambda$ are constants. The covariant derivatives are defined as

$$D_\mu\varphi \to \partial_\mu\varphi - ig_w W_\mu\varphi - \frac{i}{2}g_y B_\mu\varphi,$$
$$D_\mu\varphi^\dagger \to \partial_\mu\varphi^\dagger + ig_w\varphi^\dagger W_\mu + \frac{i}{2}g_y\varphi^\dagger B_\mu, \tag{7.9}$$

or in components

$$(D_\mu\varphi)_I \to \partial_\mu\varphi_I - \frac{ig_w}{\hbar}W_\mu^A(T_A)_I{}^J\varphi_J - \frac{i}{2}g_y B_\mu\varphi_I,$$
$$(D_\mu\varphi^\dagger)^I \to \partial_\mu(\varphi^\dagger)^I + \frac{ig_w}{\hbar}(\varphi^\dagger)^J(T_A)_J{}^I W_\mu^A + \frac{i}{2}g_y(\varphi^\dagger)^I B_\mu. \tag{7.10}$$

### 7.2.3   Quark sector

Next, we have the quark sector. The Lagrangian is given as

$$\frac{1}{\hbar c}\mathcal{L}_{\text{quark}} = \frac{i}{2}\left(Q_m^{L\dagger}\sigma^\mu D_\mu Q_m^L - D_\mu Q_m^{L\dagger}\sigma^\mu Q_m^L\right)$$
$$+ \frac{i}{2}\left(u_m^{R\dagger}\bar\sigma^\mu D_\mu u_m^R - D_\mu u_m^{R\dagger}\bar\sigma^\mu u_m^R\right) + \frac{i}{2}\left(d_m^{R\dagger}\bar\sigma^\mu D_\mu d_m^R - D_\mu d_m^{R\dagger}\bar\sigma^\mu d_m^R\right)$$
$$- Y_{mn}^d\left(Q_m^{L\dagger}\varphi\, d_n^R + d_n^{R\dagger}\varphi^\dagger Q_m^L\right) - Y_{mn}^u\left(Q_m^{L\dagger}\tilde\varphi\, u_n^R + u_n^{R\dagger}\tilde\varphi^\dagger Q_m^L\right)$$
$$= \frac{i}{2}\left((Q_m^{L\dagger})^{\alpha Ii}\sigma_{\alpha\dot\alpha}^\mu(D_\mu Q_m^L)_{Ii}^{\dot\alpha} - (D_\mu Q_m^{L\dagger})^{\alpha Ii}\sigma_{\alpha\dot\alpha}^\mu(Q_m^L)_{Ii}^{\dot\alpha}\right)$$
$$+ \frac{i}{2}\left((u_m^{R\dagger})_{\dot\alpha}^i\bar\sigma^{\mu\dot\alpha\alpha}(D_\mu u_m^R)_{\alpha i} - (D_\mu u_m^{R\dagger})_{\dot\alpha}^i\bar\sigma^{\mu\dot\alpha\alpha}(u_m^R)_{\alpha i}\right)$$
$$+ \frac{i}{2}\left((d_m^{R\dagger})_{\dot\alpha}^i\bar\sigma^{\mu\dot\alpha\alpha}(D_\mu d_m^R)_{\alpha i} - (D_\mu d_m^{R\dagger})_{\dot\alpha}^i\bar\sigma^{\mu\dot\alpha\alpha}(d_m^R)_{\alpha i}\right)$$
$$- \left(Y_{mn}^d(Q_m^{L\dagger})^{\alpha Ii}\varphi_I(d_n^R)_{\alpha i} + Y_{mn}^{d*}(d_n^{R\dagger})_{\dot\alpha}^i(\varphi^\dagger)^I(Q_m^L)_{Ii}^{\dot\alpha}\right)$$
$$- \left(Y_{mn}^u(Q_m^{L\dagger})^{\alpha Ii}\tilde\varphi_I(u_n^R)_{\alpha i} + Y_{mn}^{u*}(u_n^{R\dagger})_{\dot\alpha}^i(\tilde\varphi^\dagger)^I(Q_m^L)_{Ii}^{\dot\alpha}\right). \tag{7.11}$$

Sum over the generation index $m$ is understood. Here $Y_{mn}^d$ and $Y_{mn}^u$ are arbitrary $3\times 3$ Yukawa coupling complex matrices that will determine the quark masses. Note that the regular Dirac mass terms for quarks, e.g. $u_m^{L\dagger}u_n^R + u_n^{R\dagger}u_m^L$, are not permitted by $SU(2)_L$ invariance, which only rotates the left-handed quarks and leaves the right-handed quarks invariant.

The covariant derivatives of quarks are defined according to their transformation properties

$$
D_\mu Q_m^L \to \partial_\mu Q_m^L - ig_s G_\mu Q_m^L - ig_w W_\mu Q_m^L - \frac{i}{6} g_y B_\mu Q_m^L,
$$

$$
D_\mu Q_m^{L\dagger} \to \partial_\mu Q_m^{L\dagger} + ig_s Q_m^{L\dagger} G_\mu + ig_w Q_m^{L\dagger} W_\mu + \frac{i}{6} g_y Q_m^{L\dagger} B_\mu,
$$

$$
D_\mu u_m^R \to \partial_\mu u_m^R - ig_s G_\mu u_m^R - \frac{2i}{3} g_y B_\mu u_m^R,
$$

$$
D_\mu u_m^{R\dagger} \to \partial_\mu u_m^{R\dagger} + ig_s u_m^{R\dagger} G_\mu + \frac{2i}{3} g_y u_m^{R\dagger} B_\mu,
$$

$$
D_\mu d_m^R \to \partial_\mu d_m^R - ig_s G_\mu d_m^R + \frac{i}{3} g_y B_\mu d_m^R,
$$

$$
D_\mu d_m^{R\dagger} \to \partial_\mu d_m^{R\dagger} + ig_s d_m^{R\dagger} G_\mu - \frac{i}{3} g_y d_m^{R\dagger} B_\mu, \tag{7.12}
$$

or in components

$$
(D_\mu Q_m^L)_{Ii}^{\dot\alpha} \to \partial_\mu (Q_m^L)_{Ii}^{\dot\alpha} - \frac{ig_s}{\hbar} G_\mu^a (T_a)_i{}^j (Q_m^L)_{Ij}^{\dot\alpha} - \frac{ig_w}{\hbar} W_\mu^A (T_A)_I{}^J (Q_m^L)_{Ji}^{\dot\alpha} - \frac{i}{6} g_y B_\mu (Q_m^L)_{Ii}^{\dot\alpha},
$$

$$
(D_\mu Q_m^{L\dagger})^{\alpha Ii} \to \partial_\mu (Q_m^{L\dagger})^{\alpha Ii} + \frac{ig_s}{\hbar} (Q_m^{L\dagger})^{\alpha Ij} (T_a)_j{}^i G_\mu^a + \frac{ig_w}{\hbar} (Q_m^{L\dagger})^{\alpha Ji} (T_A)_J{}^I W_\mu^A + \frac{i}{6} g_y (Q_m^{L\dagger})^{\alpha Ii} B_\mu,
$$

$$
(D_\mu u_m^R)_{\alpha i} \to \partial_\mu (u_m^R)_{\alpha i} - \frac{ig_s}{\hbar} G_\mu^a (T_a)_i{}^j (u_m^R)_{\alpha j} - \frac{2i}{3} g_y B_\mu (u_m^R)_{\alpha i},
$$

$$
(D_\mu u_m^{R\dagger})_{\dot\alpha}^i \to \partial_\mu (u_m^{R\dagger})_{\dot\alpha}^i + \frac{ig_s}{\hbar} (u_m^{R\dagger})_{\dot\alpha}^j (T_a)_j{}^i G_\mu^a + \frac{2i}{3} g_y (u_m^{R\dagger})_{\dot\alpha}^i B_\mu,
$$

$$
(D_\mu d_m^R)_{\alpha i} \to \partial_\mu (d_m^R)_{\alpha i} - \frac{ig_s}{\hbar} G_\mu^a (T_a)_i{}^j (d_m^R)_{\alpha j} + \frac{i}{3} g_y B_\mu (d_m^R)_{\alpha i},
$$

$$
(D_\mu d_m^{R\dagger})_{\dot\alpha}^i \to \partial_\mu (u_m^{R\dagger})_{\dot\alpha}^i + \frac{ig_s}{\hbar} (d_m^{R\dagger})_{\dot\alpha}^j (T_a)_j{}^i G_\mu^a - \frac{i}{3} g_y (d_m^{R\dagger})_{\dot\alpha}^i B_\mu. \tag{7.13}
$$

It can explicitly be checked that the Lagrangian is invariant under all the symmetries.

### 7.2.4  Lepton sector

Next, we have the lepton sector. The Lagrangian is given as

$$
\begin{aligned}
\frac{1}{\hbar c} \mathcal{L}_{\text{lepton}} &= \frac{i}{2} \left( L_m^{L\dagger} \sigma^\mu D_\mu L_m^L - D_\mu L_m^{L\dagger} \sigma^\mu L_m^L \right) + \frac{i}{2} \left( e_m^{R\dagger} \bar\sigma^\mu D_\mu e_m^R - D_\mu e_m^{R\dagger} \bar\sigma^\mu e_m^R \right) \\
&\quad - Y_{mn}^e \left( L_m^{L\dagger} \varphi\, e_n^R + e_n^{R\dagger} \varphi^\dagger L_m^L \right) \\
&= \frac{i}{2} \left( (L_m^{L\dagger})^{\alpha I} \sigma^\mu_{\alpha\dot\alpha} (D_\mu L_m^L)_I^{\dot\alpha} - (D_\mu L_m^{L\dagger})^{\alpha I} \sigma^\mu_{\alpha\dot\alpha} (L_m^L)_I^{\dot\alpha} \right) \\
&\quad + \frac{i}{2} \left( (e_m^{R\dagger})_{\dot\alpha} \bar\sigma^{\mu\dot\alpha\alpha} (D_\mu e_m^R)_\alpha - (D_\mu e_m^{R\dagger})_{\dot\alpha} \bar\sigma^{\mu\dot\alpha\alpha} (e_m^R)_\alpha \right) \\
&\quad - \left( Y_{mn}^e (L_m^{L\dagger})^{\alpha I} \varphi_I (e_n^R)_\alpha + Y_{mn}^{e*} (e_n^{R\dagger})_{\dot\alpha} (\varphi^\dagger)^I (L_m^L)_I^{\dot\alpha} \right). \tag{7.14}
\end{aligned}
$$

Sum over the generation index $m$ is understood. Here $Y_{mn}^e$ is an arbitrary $3 \times 3$ Yukawa coupling complex matrix that determines the mass of electron, muon, and taon. A similar Yukawa mass term for neutrinos is not admitted by the Standard Model due the absence of right-handed neutrinos.

The covariant derivatives appearing above are defined as

$$
\begin{aligned}
\mathrm{D}_\mu L_m^L &\to \partial_\mu L_m^L - i g_w W_\mu L_m^L + \frac{i}{2} g_y B_\mu L_m^L, \\
\mathrm{D}_\mu L_m^{L\dagger} &\to \partial_\mu L_m^{L\dagger} + i g_w L_m^{L\dagger} W_\mu - \frac{i}{2} g_y L_m^{L\dagger} B_\mu, \\
\mathrm{D}_\mu e_m^R &\to \partial_\mu e_m^R + i g_y B_\mu e_m^R, \\
\mathrm{D}_\mu e_m^{R\dagger} &\to \partial_\mu e_m^{R\dagger} - i g_y e_m^{R\dagger} B_\mu,
\end{aligned}
\tag{7.15}
$$

or in components

$$
\begin{aligned}
(\mathrm{D}_\mu L_m^L)_I^{\dot\alpha} &\to \partial_\mu (L_m^L)_I^{\dot\alpha} - \frac{i g_w}{\hbar} W_\mu^A (T_A)_I{}^J (L_m^L)_J^{\dot\alpha} + \frac{i}{2} g_y B_\mu (L_m^L)_I^{\dot\alpha}, \\
(\mathrm{D}_\mu L_m^{L\dagger})^{\alpha I} &\to \partial_\mu (L_m^{L\dagger})^{\alpha I} + \frac{i g_w}{\hbar} (L_m^{L\dagger})^{\alpha J} (T_A)_J{}^I W_\mu^A - \frac{i}{2} g_y (L_m^{L\dagger})^{\alpha I} B_\mu, \\
(\mathrm{D}_\mu e_m^R)_\alpha &\to \partial_\mu (e_m^R)_\alpha + i g_y B_\mu (e_m^R)_\alpha, \\
(\mathrm{D}_\mu e_m^{R\dagger})_{\dot\alpha} &\to \partial_\mu (e_m^{R\dagger})_{\dot\alpha} - i g_y (e_m^{R\dagger})_{\dot\alpha} B_\mu.
\end{aligned}
\tag{7.16}
$$

It can explicitly be checked that the Lagrangian is invariant under all the symmetries.

## 7.3 Higgs mechanism and mass generation

Following our discussion from the electroweak theory, the Higgs potential has a minima at $\varphi^\dagger \varphi = v^2/2$, where $v = \mu/\sqrt{\lambda}$. Let us spontaneously choose the minima $\varphi_{0I} = v/\sqrt{2}\,\hat\varphi_{0I}$, where $\hat\varphi_{0I} = (0, 1)$. We can expand the Standard Model Lagrangian around this state according to

$$
\varphi_I = \frac{1}{\sqrt{2}} (v + \eta) \, e^{i/2\,\pi} (V_\pi)_I{}^J \hat\varphi_{0J},
\tag{7.17}
$$

where $V_\pi = \exp(i/\hbar\,\pi^A T_A)$, with $\pi^A$ and $\pi$ being the plausible Goldstone fields capturing the fluctuations in $\mathrm{SU}(2)_L \times \mathrm{U}(1)_Y$ phase of $\varphi$, while $\eta$ being the fluctuation in the magnitude. We follow this by a redefinition of the remaining fields

$$
\begin{aligned}
G_\mu^a \to G_\mu^a, \qquad W_\mu^A (T_A)_I{}^J &\to (V_\pi)_I{}^K \left( W_\mu^A (T_A)_K{}^L + \frac{i\hbar}{g_w} \delta_K^L \partial_\mu \right) (V_\pi^{-1})_L{}^J, \qquad B_\mu \to B_\mu + \frac{1}{g_y} \partial_\mu \pi, \\
(Q_m^L)_{Ii}^{\dot\alpha} &\to e^{i\pi/6} (V_\pi)_I{}^J (Q_m^L)_{Ji}^{\dot\alpha}, \qquad (u_m^R)_{\alpha i} \to e^{2i\pi/3} (u_m^R)_{\alpha i}, \qquad (d_m^R)_{\alpha i} \to e^{-i\pi/3} (d_m^R)_{\alpha i}, \\
(L_m^L)_I^{\dot\alpha} &\to e^{-i\pi/2} (V_\pi)_I{}^J (L_m^L)_J^{\dot\alpha}, \qquad (e_m^R)_\alpha \to e^{-i\pi} (e_m^R)_\alpha,
\end{aligned}
\tag{7.18}
$$

that, being a symmetry transformation, renders the Lagrangian independent of the $\pi^A$ and $\pi$ fields. The only effect of the procedure above on the Lagrangian is to the Higgs field $\varphi_I \to 1/\sqrt{2}(v + \eta)\hat\varphi_{0I}$. Note that this sets the conjugate Higgs to $\tilde\varphi_I \to 1/\sqrt{2}(v+\eta)\hat{\tilde\varphi}_{0I}$, where $\hat{\tilde\varphi}_{0I} = (0, 1)$. In the following, we shall explore the repercussions of this on the Standard Model Lagrangian.

### 7.3.1 Higgs and gauge sector

The Higgs and gauge sector Lagrangian after spontaneous symmetry breaking follows directly from our electroweak discussion. We get

$$
\frac{1}{\hbar c}\left(\mathcal{L}_{\text{higgs}}^{\text{SSB}} + \mathcal{L}_{\text{gauge}}^{\text{SSB}}\right)
$$

$$
= -\frac{1}{2}\partial_\mu\eta\partial^\mu\eta + \frac{1}{2}\mu^2(v+\eta)^2 - \frac{1}{4}\lambda(v+\eta)^4 - \frac{1}{4}(v+\eta)^2 g_w^2\left(W_\mu^- W^{+\mu} + \frac{1}{2}\sec^2\theta_w\, Z^\mu Z_\mu\right)
$$

$$
- \frac{1}{4}G_{\mu\nu}^a G_a^{\mu\nu} - \frac{1}{2}W_{\mu\nu}^+ W^{-\mu\nu} - \frac{1}{4}Z_{\mu\nu}Z^{\mu\nu} - \frac{1}{4}F_{\mu\nu}F^{\mu\nu}
$$

$$
- ig_w\cos\theta_w\left(W^{-\mu\nu}W_\mu^+ Z_\nu - W^{+\mu\nu}W_\mu^- Z_\nu - Z^{\mu\nu}W_\mu^+ W_\nu^-\right) + ig_w\sin\theta_w F^{\mu\nu}W_\mu^+ W_\nu^-
$$

$$
- g_w^2\cos^2\theta_w\left(W_\mu^+ W^{-\mu}Z_\nu Z^\nu - W_\nu^+ Z^\nu W_\mu^- Z^\mu\right)
$$

$$
+ \frac{1}{2}g_w^2\left(W_\mu^+ W^{+\mu}W_\nu^- W^{-\nu} - (W_\mu^+ W^{-\mu})^2\right). \tag{7.19}
$$

Focusing on the quadratic sector and ignoring interactions, we get

$$
\frac{1}{\hbar c}\left(\mathcal{L}_{\text{higgs}}^{\text{SSB}} + \mathcal{L}_{\text{gauge}}^{\text{SSB}}\right)
$$

$$
= -\frac{1}{4}G_{\mu\nu}^a G_a^{\mu\nu} - \frac{1}{2}W_{\mu\nu}^- W^{+\mu\nu} - \frac{m_W^2 c^2}{\hbar^2}W_\mu^- W^{+\mu} - \frac{1}{4}Z_{\mu\nu}Z^{\mu\nu} - \frac{m_Z^2 c^2}{2\hbar^2}Z^\mu Z_\mu - \frac{1}{4}F_{\mu\nu}F^{\mu\nu}
$$

$$
- \frac{1}{2}\partial_\mu\eta\partial^\mu\eta - \mu^2\eta^2 + \text{interactions.} \tag{7.20}
$$

We get a massive residual Higgs field $\eta$ with mass $\sqrt{2}\mu\hbar/c$, three massive weak force gauge fields $W_\mu^\pm$, $Z_\mu$ with respective masses $m_W = v/2\,g_w\hbar/c$, $m_Z = m_W\sec\theta_w$, one massless electromagnetic gauge field (photon) $A_\mu$, and eight massless strong force gauge fields (gluons) $G_\mu^a$.

The interactions in this sector are also the same as the electroweak interactions in section 6.5.5, but with additional gluon self interactions. We have the self and weak interactions of the Higgs field:

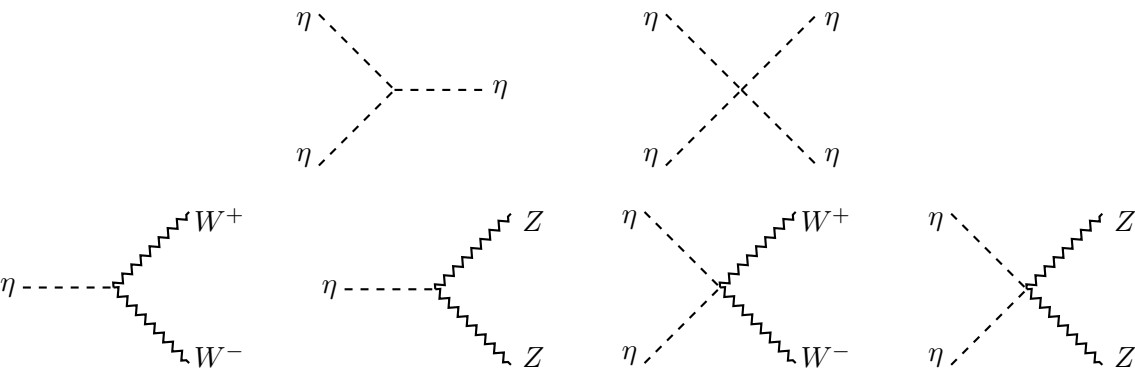

mixed interactions between photon and weak bosons:

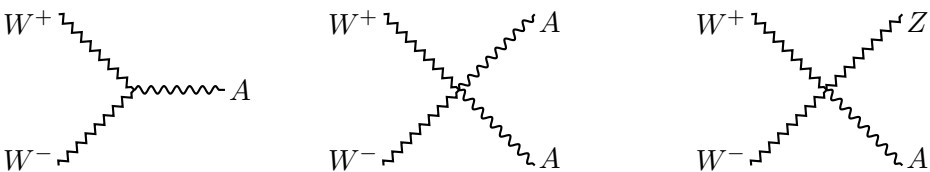

self interactions of weak bosons:

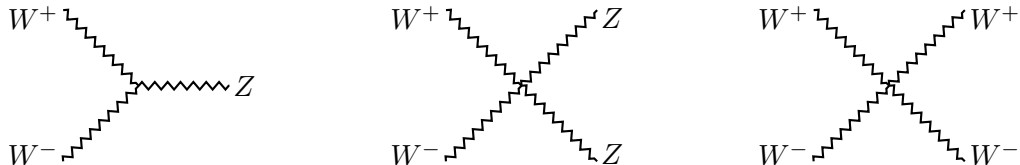

and self interactions of gluons

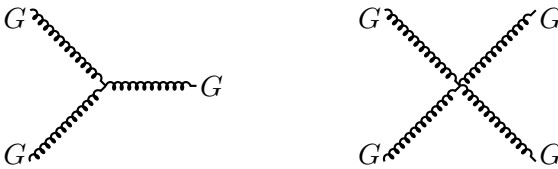

Note that the residual Higgs field is uncharged under the residual $SU(3) \times U(1)$ symmetry, hence it does not interact with photons or gluons.

### 7.3.2 Quark sector

Spontaneous symmetry breaking of the quark sector Lagrangian follows from the generic discussion in section 6.5.3, with $\chi = Q_m^L$ and $\psi = u_m^R$, $d_m^R$

$$
\begin{aligned}
\frac{1}{\hbar c}\mathcal{L}_{\text{quark}}^{\text{SSB}} = {} & \frac{i}{2}\left(u_m^{R\dagger}\bar{\sigma}^\mu \tilde{D}_\mu u_m^R - \tilde{D}_\mu u_m^{R\dagger}\bar{\sigma}^\mu u_m^R\right) + \frac{i}{2}\left(d_m^{R\dagger}\bar{\sigma}^\mu \tilde{D}_\mu d_m^R - \tilde{D}_\mu d_m^{R\dagger}\bar{\sigma}^\mu d_m^R\right) \\
& + \frac{i}{2}\left(u_m^{L\dagger}\sigma^\mu \tilde{D}_\mu u_m^L - \tilde{D}_\mu u_m^{L\dagger}\sigma^\mu u_m^L\right) + \frac{i}{2}\left(d_m^{L\dagger}\sigma^\mu \tilde{D}_\mu d_m^L - \tilde{D}_\mu d_m^{L\dagger}\sigma^\mu d_m^L\right) \\
& - \left(\lambda_u^R\, u_m^{R\dagger}\bar{\sigma}^\mu u_m^R + \lambda_d^R\, d_m^{R\dagger}\bar{\sigma}^\mu d_m^R + \lambda_u^L\, u_m^{L\dagger}\sigma^\mu u_m^L + \lambda_d^L\, d_m^{L\dagger}\sigma^\mu d_m^L\right) Z_\mu \\
& + \frac{g_w}{\sqrt{2}}\left(W_\mu^+ u_m^{L\dagger}\sigma^\mu d_m^L + W_\mu^- d_m^{L\dagger}\sigma^\mu u_m^L\right) \\
& - \frac{v+\eta}{\sqrt{2}}\left(Y_{mn}^u u_m^{L\dagger} u_n^R + Y_{mn}^{u*} u_n^{R\dagger} u_m^L\right) - \frac{v+\eta}{\sqrt{2}}\left(Y_{mn}^d d_m^{L\dagger} d_n^R + Y_{mn}^{d*} d_n^{R\dagger} d_m^L\right), \quad (7.21)
\end{aligned}
$$

where

$$
\lambda_u^R = \frac{2}{3}g_y \sin\theta_w, \qquad \lambda_d^R = -\frac{1}{3}g_y \sin\theta_w,
$$

$$
\lambda_u^L = \frac{1}{6}g_y \sin\theta_w - \frac{1}{2}g_w \cos\theta_w, \qquad \lambda_d^L = \frac{1}{6}g_y \sin\theta_w + \frac{1}{2}g_w \cos\theta_w, \quad (7.22)
$$

and the residual $SU(3) \times U(1)$ covariant derivatives are defined as

$$
\tilde{D}_\mu u_m^{L,R} = \partial_\mu u_m^{L,R} - ig_s G_\mu u_m^{L,R} - \frac{2i}{3}eA_\mu u_m^{L,R},
$$

$$
\tilde{D}_\mu d_m^{L,R} = \partial_\mu d_m^{L,R} - ig_s G_\mu d_m^{L,R} + \frac{i}{3}eA_\mu d_m^{L,R}. \quad (7.23)
$$

Focusing on the quadratic sector of the Lagrangian, we get

$$\frac{1}{\hbar c}\mathcal{L}^{\text{SSB}}_{\text{quark}} = \frac{i}{2}\left(u_m^{R\dagger}\bar{\sigma}^\mu\partial_\mu u_m^R - \partial_\mu u_m^{R\dagger}\bar{\sigma}^\mu u_m^R\right) + \frac{i}{2}\left(d_m^{R\dagger}\bar{\sigma}^\mu\partial_\mu d_m^R - \partial_\mu d_m^{R\dagger}\bar{\sigma}^\mu d_m^R\right)$$

$$+ \frac{i}{2}\left(u_m^{L\dagger}\sigma^\mu\partial_\mu u_m^L - \partial_\mu u_m^{L\dagger}\sigma^\mu u_m^L\right) + \frac{i}{2}\left(d_m^{L\dagger}\sigma^\mu\partial_\mu d_m^L - \partial_\mu d_m^{L\dagger}\sigma^\mu d_m^L\right)$$

$$- \frac{v}{\sqrt{2}}\left(Y^u_{mn}u_m^{L\dagger}u_n^R + Y^{u*}_{mn}u_n^{R\dagger}u_m^L\right) - \frac{v}{\sqrt{2}}\left(Y^d_{mn}d_m^{L\dagger}d_n^R + Y^{d*}_{mn}d_n^{R\dagger}d_m^L\right) + \text{int.}, \quad (7.24)$$

To convert this into a standard form, we can perform a unitary transformation in the flavour (generation) space

$$u_m^L \to \alpha^{uL}_{mn}u_n^L, \qquad u_m^R \to \alpha^{uR}_{mn}u_n^R, \qquad d_m^L \to \alpha^{dL}_{mn}d_n^L, \qquad d_m^R \to \alpha^{dR}_{mn}d_n^R, \quad (7.25)$$

for arbitrary unitary matrices $\alpha^{uL}_{mn}$, $\alpha^{uR}_{mn}$, $\alpha^{dL}_{mn}$, and $\alpha^{dR}_{mn}$, such that the transformed Yukawa coupling matrix is diagonal with positive eigenvalues[15]

$$\frac{v\hbar}{\sqrt{2}c}\alpha^{uL\dagger}_{mr}Y^u_{rs}\alpha^{uR}_{sn} = m^u_{mn}, \qquad \frac{v\hbar}{\sqrt{2}c}\alpha^{dL\dagger}_{mr}Y^d_{rs}\alpha^{dR}_{sn} = m^d_{mn}, \quad (7.26)$$

where $m^{u,d}_{mn}$ are diagonal matrices with positive entries. The transformed free Lagrangian can be represented in the Dirac representation by defining 4-component Dirac spinors

$$u_m = \begin{pmatrix} u_m^R \\ u_m^L \end{pmatrix}, \qquad d_m = \begin{pmatrix} d_m^R \\ d_m^L \end{pmatrix}, \quad (7.27)$$

leading to

$$\mathcal{L}^{\text{SSB}}_{\text{quark}} = \frac{i\hbar c}{2}\left(\bar{u}_m\gamma^\mu\partial_\mu u_m - \partial_\mu\bar{u}_m\gamma^\mu u_m\right) - m^u_{mn}c^2\bar{u}_m u_n$$

$$+ \frac{i\hbar c}{2}\left(\bar{d}_m\gamma^\mu\partial_\mu d_m - \partial_\mu\bar{d}_m\gamma^\mu d_m\right) - m^d_{mn}c^2\bar{d}_m d_n + \text{interactions}, \quad (7.28)$$

In this basis, we get the mass $m^{u,d}_{11}$ for up and down quarks, $m^{u,d}_{22}$ for charm and strange quarks, and $m^{u,d}_{33}$ for top and bottom quarks.

Shifting to the "mass basis", however, has non-trivial implications for interactions. Since the weak interactions in the forth line mix $u_m^L$ and $d_m^L$ quarks, which have been redefined independently, these interactions are not invariant under the said redefinition. To wit, the full quark sector Lagrangian becomes

$$\frac{1}{\hbar c}\mathcal{L}^{\text{SSB}}_{\text{quark}} = \frac{i}{2}\left(u_m^{R\dagger}\bar{\sigma}^\mu\tilde{\text{D}}_\mu u_m^R - \tilde{\text{D}}_\mu u_m^{R\dagger}\bar{\sigma}^\mu u_m^R\right) + \frac{i}{2}\left(d_m^{R\dagger}\bar{\sigma}^\mu\tilde{\text{D}}_\mu d_m^R - \tilde{\text{D}}_\mu d_m^{R\dagger}\bar{\sigma}^\mu d_m^R\right)$$

$$+ \frac{i}{2}\left(u_m^{L\dagger}\sigma^\mu\tilde{\text{D}}_\mu u_m^L - \tilde{\text{D}}_\mu u_m^{L\dagger}\sigma^\mu u_m^L\right) + \frac{i}{2}\left(d_m^{L\dagger}\sigma^\mu\tilde{\text{D}}_\mu d_m^L - \tilde{\text{D}}_\mu d_m^{L\dagger}\sigma^\mu d_m^L\right)$$

$$- \left(\lambda^R_u u_m^{R\dagger}\bar{\sigma}^\mu u_m^R + \lambda^R_d d_m^{R\dagger}\bar{\sigma}^\mu d_m^R + \lambda^L_u u_m^{L\dagger}\sigma^\mu u_m^L + \lambda^L_d d_m^{L\dagger}\sigma^\mu d_m^L\right)Z_\mu$$

$$+ \frac{g_w}{\sqrt{2}}\left(V_{mn}W^+_\mu u_m^{L\dagger}\sigma^\mu d_n^L + V^\dagger_{mn}W^-_\mu d_m^{L\dagger}\sigma^\mu u_n^L\right)$$

$$- \frac{v+\eta}{v\hbar/c}m^u_{mn}\left(u_m^{L\dagger}u_n^R + u_n^{R\dagger}u_m^L\right) - \frac{v+\eta}{v\hbar/c}\tilde{m}^d_{mn}\left(d_m^{L\dagger}d_n^R + d_n^{R\dagger}d_m^L\right), \quad (7.29)$$

---

[15]This can be done generically for any non-singular complex matrix $M$. Note that the Hermitian matrix $M^\dagger M$ can always be diagonalised by some unitary matrix $U$ as $U^\dagger M^\dagger M U = D^2$, where $D^2$ is a matrix with positive eigenvalues and $D$ is a matrix with its positive square-roots as diagonal entries. Then, we can check that $V^\dagger M U = D$ where $V = MUD^{-1}$ is also a unitary matrix.

where the CKM (Cabibbo-Kobayashi-Maskawa) matrix $V_{mn}$ is defined as the unitary matrix

$$V_{mn} = \alpha_{mr}^{uL\dagger}\alpha_{rn}^{dL}. \tag{7.30}$$

Since the CKM matrix is generally non-diagonal, the weak interactions mediated by $W_\mu^\pm$ bosons can switch the generation of quarks.[16] This is to say that, during interactions, up quarks can turn into charm or top quarks, while down quarks can turn into strange or bottom quarks, and vice-versa. These are known as quark flavor oscillations. Experimentally, the CKM matrix is found to be extremely close to diagonal, meaning that such quark flavor changing interactions are extremely rare. All in all, quarks can interact via electromagnetic forces

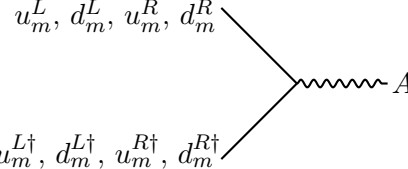

weak forces

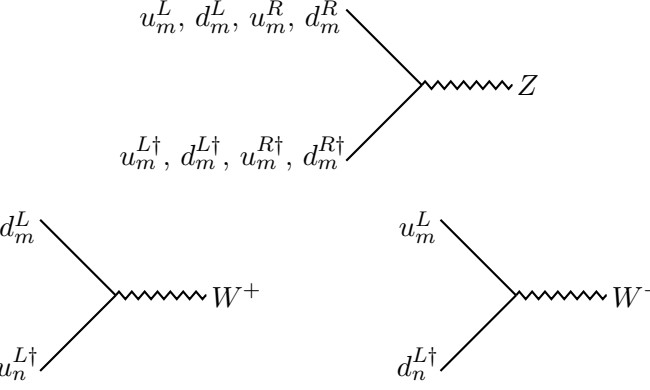

strong forces

and the residual Higgs field

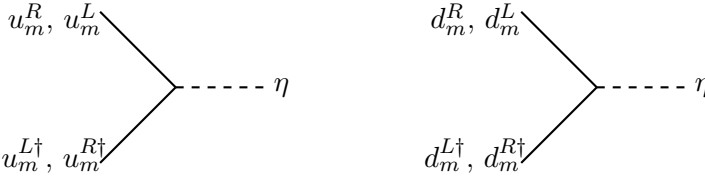

---

[16]A unitary matrix can generically be parametrised by 9 independent real components. However, we can absorb 5 of these in the CKM matrix into a redefinition of the left-handed quarks $u_n^L$, $d_n^L$ with relative phases. This leaves us with 4 independent parameters: three Euler angles and a complex "CP-violating" phase. See section 8 for a discussion on CP symmetry and its violation.

Importantly, the $W_\mu^\pm$ weak interactions only act on the left-handed quarks, while the strong, electromagnetic, and Higgs interactions act democratically on the left- and right-handed quarks. Note also that the Higgs interactions can change the handedness, while $W_\mu^\pm$ interactions can change the generation of quarks.

### 7.3.3 Lepton sector

Spontaneous symmetry breaking of the lepton sector Lagrangian follows from the generic discussion in section 6.5.3, with $\chi = L_m^L$ and $\psi = e_m^R$

$$\frac{1}{\hbar c}\mathcal{L}_{\text{lepton}}^{\text{SSB}} = \frac{i}{2}\left(e_m^{R\dagger}\bar{\sigma}^\mu\tilde{D}_\mu e_m^R - \tilde{D}_\mu e_m^{R\dagger}\bar{\sigma}^\mu e_m^R\right)$$
$$+ \frac{i}{2}\left(\nu_m^{L\dagger}\sigma^\mu\partial_\mu\nu_m^L - \partial_\mu\nu_m^{L\dagger}\sigma^\mu\nu_m^L\right) + \frac{i}{2}\left(e_m^{L\dagger}\sigma^\mu\tilde{D}_\mu e_m^L - \tilde{D}_\mu e_m^{L\dagger}\sigma^\mu e_m^L\right)$$
$$- \left(\lambda_e^R\, e_m^{R\dagger}\bar{\sigma}^\mu e_m^R + \lambda_\nu^L\, \nu_m^{L\dagger}\sigma^\mu\nu_m^L + \lambda_e^L\, e_m^{L\dagger}\sigma^\mu e_m^L\right)Z_\mu$$
$$+ \frac{g_w}{\sqrt{2}}\left(W_\mu^+\nu_m^{L\dagger}\sigma^\mu e_m^L + W_\mu^- e_m^{L\dagger}\sigma^\mu\nu_m^L\right) - \frac{v+\eta}{\sqrt{2}}\left(Y_{mn}^e e_m^{L\dagger}e_n^R + Y_{mn}^{e*}e_n^{R\dagger}e_m^L\right), \quad (7.31)$$

where

$$\lambda_e^R = -g_y\sin\theta_w,$$
$$\lambda_\nu^L = -\frac{1}{2}g_y\sin\theta_w - \frac{1}{2}g_w\cos\theta_w, \qquad \lambda_e^L = -\frac{1}{2}g_y\sin\theta_w + \frac{1}{2}g_w\cos\theta_w. \quad (7.32)$$

and the residual $SU(3) \times U(1)$ transformations act as

$$\tilde{D}_\mu e_m^{L,R} \to \partial_\mu e_m^R + ieA_\mu e_m^R. \quad (7.33)$$

Note that neutrinos are uncharged under strong and electromagnetic forces $SU(3) \times U(1)$.

Focusing on the quadratic sector of the Lagrangian, we obtain

$$\frac{1}{\hbar c}\mathcal{L}_{\text{lepton}}^{\text{SSB}} = \frac{i}{2}\left(e_m^{R\dagger}\bar{\sigma}^\mu\partial_\mu e_m^R - \partial_\mu e_m^{R\dagger}\bar{\sigma}^\mu e_m^R\right) + \frac{i}{2}\left(e_m^{L\dagger}\sigma^\mu\partial_\mu e_m^L - \partial_\mu e_m^{L\dagger}\sigma^\mu e_m^L\right)$$
$$+ \frac{i}{2}\left(\nu_m^{L\dagger}\sigma^\mu\partial_\mu\nu_m^L - \partial_\mu\nu_m^{L\dagger}\sigma^\mu\nu_m^L\right) - \frac{v}{\sqrt{2}}\left(Y_{mn}^e e_m^{L\dagger}e_n^R + Y_{mn}^{e*}e_n^{R\dagger}e_m^L\right), \quad (7.34)$$

Similar to the quark sector, we can perform a unitary transformation on the leptons in the flavour (generation) space

$$e_m^L \to \alpha_{mn}^{eL}e_n^L, \qquad e_m^R \to \alpha_{mn}^{eR}e_n^R, \qquad \nu_m^L \to \alpha_{mn}^{eL}\nu_n^L, \qquad \nu_m^R \to \alpha_{mn}^{eR}\nu_n^R, \quad (7.35)$$

for arbitrary unitary matrices $\alpha_{mn}^{eL}$ and $\alpha_{mn}^{eR}$, so that the Yukawa coupling matrix diagonalises to

$$\frac{v\hbar}{\sqrt{2}c}\alpha_{mr}^{eL\dagger}Y_{rs}^e\,\alpha_{sn}^{eR} = m_{mn}^e, \quad (7.36)$$

where $m_{mn}^e$ is a diagonal matrix with positive entries. Note that we transformed both $e_m^{L,R}$ and $\nu_m^{L,R}$ with the same matrix $\alpha_{mn}^{eL,R}$, which is fine because there is no "neutrino mass matrix" to

diagonalise. The transformed free Lagrangian can be represented in the Dirac representation by defining 4-component Dirac spinors

$$\nu_m = \begin{pmatrix} 0 \\ \nu_m^L \end{pmatrix}, \qquad e_m = \begin{pmatrix} e_m^R \\ e_m^L \end{pmatrix}, \tag{7.37}$$

leading to

$$\begin{aligned}
\mathcal{L}_{\text{lepton}} = {} & \frac{i\hbar c}{2} \left( \bar{\nu}_m \gamma^\mu \partial_\mu \nu_m - \partial_\mu \bar{\nu}_m \gamma^\mu \nu_m \right) \\
& + \frac{i\hbar c}{2} \left( \bar{e}_m \gamma^\mu \partial_\mu e_m - \partial_\mu \bar{e}_m \gamma^\mu e_m \right) - m_{mn}^e c^2 \bar{e}_m e_n + \text{interactions}.
\end{aligned} \tag{7.38}$$

In this basis, we get the mass $m_{11}^e$ for electrons, $m_{22}^e$ for muons, and $m_{33}^e$ for tau leptons.

Since we have transformed both $u_m$ and $\nu_m$ with the same matrix, we do not get an analogue of the CKM matrix for leptons. To wit, the full Lagrangian takes the form

$$\begin{aligned}
\frac{1}{\hbar c} \mathcal{L}_{\text{lepton}}^{\text{SSB}} = {} & -\frac{i}{2} \left( e_m^{R\dagger} \bar{\sigma}^\mu \tilde{\mathrm{D}}_\mu e_m^R - \tilde{\mathrm{D}}_\mu e_m^{R\dagger} \bar{\sigma}^\mu e_m^R \right) \\
& - \frac{i}{2} \left( \nu_m^{L\dagger} \sigma^\mu \partial_\mu \nu_m^L - \partial_\mu \nu_m^{L\dagger} \sigma^\mu \nu_m^L \right) - \frac{i}{2} \left( e_m^{L\dagger} \sigma^\mu \tilde{\mathrm{D}}_\mu e_m^L - \tilde{\mathrm{D}}_\mu e_m^{L\dagger} \sigma^\mu e_m^L \right) \\
& + \left( \lambda_e^R \, e_m^{R\dagger} \bar{\sigma}^\mu e_m^R + \lambda_\nu^L \, \nu_m^{L\dagger} \sigma^\mu \nu_m^L + \lambda_e^L \, e_m^{L\dagger} \sigma^\mu e_m^L \right) Z_\mu \\
& - \frac{g_w}{\sqrt{2}} \left( W_\mu^+ \nu_m^{L\dagger} \sigma^\mu e_m^L + W_\mu^- e_m^{L\dagger} \sigma^\mu \nu_m^L \right) - \frac{v+\eta}{v\hbar/c} m_{mn}^e \left( e_m^{L\dagger} e_n^R + e_n^{R\dagger} e_m^L \right).
\end{aligned} \tag{7.39}$$

The (generations of) electrons can interact via electromagnetic forces

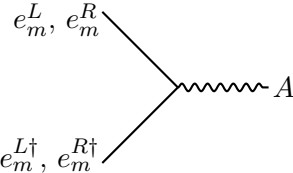

while both electrons and neutrons can interact via weak forces

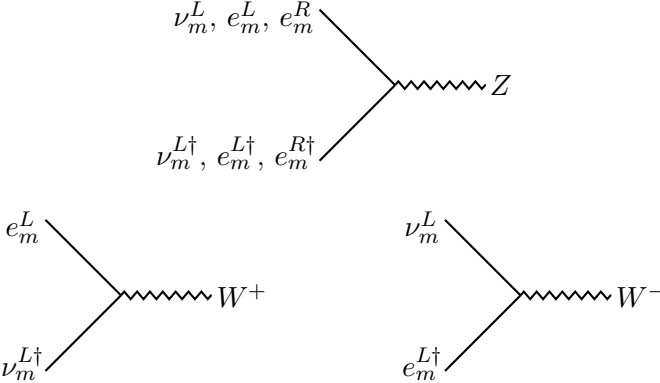

Only the electrons interact with the residual Higgs field

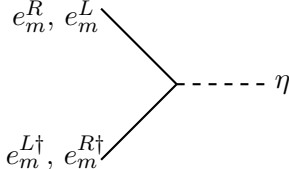

Leptons do not talk to the strong force gluons. Notably, there are no flavor (generation) changing interactions in this model.

   We could extend the Standard Model to include right-handed neutrinos. However, at the time the Standard Model was proposed, neutrinos were known to be massless, i.e. they do not admit a Yukawa interaction. As such, we know that neutrinos are uncharged under electromagnetic and strong forces, while the right-handed neutrinos will also be uncharged under weak forces. In the absence of a Yukawa coupling, the right handed neutrinos will not talk to the residual Higgs field either. Consequently, right-handed neutrinos completely decouple from the rest of the Standard Model and leave no physically distinguishable signatures. However, now we have experimental evidence that neutrinos do have a nonzero mass. It implies that we can indeed add right-handed neutrinos with non-trivial Yukawa interactions. These right-handed neutrinos talk to the rest of the Standard Model via Higgs interactions. In the process of diagonalising the neutrino mass matrix, we will encounter a non-diagonal PMNS (Pontecorvo-Maki-Nakagawa-Sakata) matrix, similar to the CKM matrix for quarks, leading to neutrino and electron flavor oscillations. Unlike the CKM matrix, the PMNS matrix is experimentally observed to be quite far from diagonal. Hence, neutrino oscillations are more readily observed in nature.

# 8 | Discrete symmetries

Let us rewind back to the spacetime Poincaré symmetries discussed in section 5. We noted that the Poincaré group $\mathbb{R}^{3,1} \rtimes O(3,1)$ (or its Lorentz subgroup $O(3,1)$) is not connected. The connected piece is the proper orthochronous Poincaré group $\mathbb{R}^{3,1} \rtimes SO^+(3,1)$, which excludes the discrete transformations: parity P and time-reversal T, as well as their combination PT. Then we quickly specialised to just the connected piece of the Poincaré group and ignored the discrete transformations altogether. The reason is that physical theories, including the Standard Model of particle physics, are not generically invariant under P, T, or PT. As it turns out, physical theories are invariant under CPT symmetry, including a charge conjugation transformation C that complex conjugates all the fields and converts all particles to anti-particles and vice-versa. There is, in fact, a *CPT theorem* that states that all Lorentz-invariant local field theories with a Hermitian Lagrangian (Hamiltonian) is CPT-invariant.

**Symmetry operators and algebra:** The discrete transformations C, P, and T individually form cyclic $\mathbb{Z}_2$ groups, i.e. operating twice with any of these operations returns to the identity operation: $C^2, P^2, T^2 \propto \mathbb{1}$. All the three discrete transformations mutually commute. Their action on the Poincaré generators is given as

$$
\begin{aligned}
CHC^{-1} &= H, & CP_iC^{-1} &= P_i, & CJ_iC^{-1} &= J_i, & CK_iC^{-1} &= K_i, \\
PHP^{-1} &= H, & PP_iP^{-1} &= -P_i, & PJ_iP^{-1} &= J_i, & PK_iP^{-1} &= -K_i, \\
THT^{-1} &= H, & TP_iT^{-1} &= -P_i, & TJ_iT^{-1} &= -J_i, & TK_iT^{-1} &= K_i.
\end{aligned} \tag{8.1}
$$

Note that all these transformations leave the Hamiltonian invariant, so that the energy of a state is not changed. As a consequence, the time-reversal operator T is anti-unitary and implements a complex conjugation $TiT^{-1} = -i$.[17] Note that the charge conjugation operator C leaves all the Poincaré generators invariant. However, given an internal symmetry Lie group $G$ (such as the QCD group $SU(3)$ or the electroweak group $SU(2) \times U(1)$), both P and T commute with the respective generators $T_a$, while C flips them, i.e.

$$
CT_aC^{-1} = -T_a, \qquad PT_aP^{-1} = T_a, \qquad TT_aT^{-1} = T_a. \tag{8.3}
$$

The combined CPT symmetry acts on the generators as

$$
\begin{aligned}
(CPT)H(CPT)^{-1} &= H, & (CPT)P_i(CPT)^{-1} &= P_i, \\
(CPT)J_i(CPT)^{-1} &= -J_i, & (CPT)K_i(CPT)^{-1} &= -K_i, \\
(CPT)T_a(CPT)^{-1} &= -T_a,
\end{aligned} \tag{8.4}
$$

or covariantly

$$
\begin{aligned}
(CPT)P_\mu(CPT)^{-1} &= P_\mu, & (CPT)M_{\mu\nu}(CPT)^{-1} &= -M_{\mu\nu}, \\
(CPT)T_a(CPT)^{-1} &= -T_a.
\end{aligned} \tag{8.5}
$$

---

[17] This curious feature follows from the observation that

$$
Ti\hbar c P_i T^{-1} = T[H, K_i]T^{-1} = [THT^{-1}, TK_iT^{-1}] = [H, K_i] = i\hbar c P_i. \tag{8.2}
$$

However, since $TP_iT^{-1} = -P_i$, it must be true that $TiT^{-1} = -i$. This is to say that T is an "anti-unitary operator" that also implements a complex conjugation.

**Action on fields:** Let us go back to the Lorentz representations, which were classified in section 5.2 using the eigenvalues $(\vec{X}^{\pm})^2$ operators, where $X_i^{\pm} = 1/2(J_i \pm iK_i)$. Under parity transformation, $\mathrm{P}: X_i^+ \leftrightarrow X_i^-$, while under time-reversal, $\mathrm{T}: X_i^{\pm} \to -X_i^{\pm}$. Hence, a representation $(j_-, j_+)$ goes to $(j_+, j_-)$ under parity, but stays invariant under time-reversal. We see that the parity operation turns a left-chiral particle into a right-chiral one and vice-versa. This also follows from the action of discrete symmetries on the chirality operator $\Gamma$, i.e. $\mathrm{P}^{-1}\Gamma\mathrm{P} = -\Gamma$ and $\mathrm{T}^{-1}\Gamma\mathrm{T} = \Gamma$. As a consequence, parity-invariant or *achiral* theories must either include particles in the representations $(j, j)$, like vectors, or $(j_-, j_+) \oplus (j_+, j_-)$ for $j_+ \neq j_-$, like Dirac spinors. The charge conjugation operator C, on the other hand, leaves the Lorentz representation invariant but converts particles to anti-particles and vice-versa. To wit, the action of C, P, T on a scalar field $\varphi(t, \vec{x})$ is defined as

$$\begin{aligned}
\mathrm{C} &: \varphi(t, \vec{x}) \to \eta_C \varphi^*(t, \vec{x}), \\
\mathrm{P} &: \varphi(t, \vec{x}) \to \eta_P \varphi(t, -\vec{x}), \\
\mathrm{T} &: \varphi(t, \vec{x}) \to \eta_T \varphi(-t, \vec{x}).
\end{aligned} \tag{8.6}$$

Here $\eta_C, \eta_P, \eta_T$ are arbitrary phases, with $|\eta_C|^2 = |\eta_P|^2 = |\eta_T|^2 = 1$, which are intrinsic properties of the field in question. One can check that all of these are independently symmetries of the scalar field theory from section 6.1.2. For a pair of left- and right-handed Weyl spinors $\psi(t, \vec{x})$, $\chi(t, \vec{x})$, we instead have

$$\begin{aligned}
\mathrm{C} &: \psi(t, \vec{x}) \to -i\eta_C \sigma_2 \chi^*(t, \vec{x}), &\quad \mathrm{C} &: \chi(t, \vec{x}) \to i\eta_C \sigma_2 \psi^*(t, \vec{x}), \\
\mathrm{P} &: \psi(t, \vec{x}) \to \eta_P \chi(t, -\vec{x}), &\quad \mathrm{P} &: \chi(t, \vec{x}) \to \eta_P \psi(t, -\vec{x}), \\
\mathrm{T} &: \psi(t, \vec{x}) \to i\eta_T \sigma_2 \psi(-t, \vec{x}), &\quad \mathrm{T} &: \chi(t, \vec{x}) \to i\eta_T \sigma_2 \chi(-t, \vec{x}).
\end{aligned} \tag{8.7}$$

C and P individually are *not* symmetries of the Weyl spinor field theories from section 6.1.3, whereas T and the combined transformation CP are symmetries.[18] Note that T transformation is also supposed to change all $i \to -i$ in the Lagrangian. For a Dirac spinor $\Psi(t, x)$, the transformation rules imply

$$\begin{aligned}
\mathrm{C} &: \Psi(t, \vec{x}) \to i\eta_C \gamma^0 \gamma^2 \bar{\Psi}^{\mathrm{T}}(t, \vec{x}), \\
\mathrm{P} &: \Psi(t, \vec{x}) \to \eta_P \gamma^0 \Psi(t, -\vec{x}), \\
\mathrm{T} &: \Psi(t, \vec{x}) \to i\eta_T \gamma_5 \gamma^2 \gamma^0 \Psi(-t, \vec{x}),
\end{aligned} \tag{8.8}$$

Interestingly, all of these can be individually checked to be symmetries of the Dirac spinor field theory from section 6.1.3 up to an overall sign in the action. For a vector field $V^\mu(t, \vec{x})$, we instead have

$$\begin{aligned}
\mathrm{C} &: V^\mu(t, \vec{x}) \to \eta_C V^{*\mu}(t, \vec{x}), \\
\mathrm{P} &: V^\mu(t, \vec{x}) \to -\eta_P \mathbb{P}^\mu_\nu V^\nu(t, -\vec{x}), \\
\mathrm{T} &: V^\mu(t, \vec{x}) \to \eta_T \mathbb{P}^\mu_\nu V^\nu(-t, \vec{x}),
\end{aligned} \tag{8.9}$$

where $\mathbb{P}^\mu_\nu = \mathrm{diag}(1, -1, -1, -1)$.

Moving on to Poincaré representations, it is easy to see that all the discrete symmetries leave the Casimir operators $P^2$ and $W^2$ invariant, whereas the helicity operator $W_0$ flips sign under

---

[18]Note that fermionic fields anti-commute. So, for instance, $\bar{\Psi}^*\Psi^* = \Psi^{\mathrm{T}}\bar{\Psi}^{\mathrm{T}} = -\bar{\Psi}\Psi$. This property is vital to illustrate invariance of spinor field theories under CP.

parity. As a consequence, massive particles transforming in some representation $(m, s)$ are invariant under C, P, and T; that is to say that the mass and spin of a particle are invariant. On the other hand, massless particles transforming in some representation $(h)$ turn into $(-h)$ under parity, but remain invariant under time-reversal and charge conjugation. As a consequence, parity-invariant massless particles must either transform in the representation $(0)$, like massless scalars, or in the representation $(h) \oplus (-h)$ for $h \neq 0$, like photons and gravitons.

**CPT theorem:** Let us consider the combined CPT operation. On various kinds of fields mentioned above, the action of CPT is given as

$$\text{CPT} : \varphi(t, \vec{x}) \to \eta_C \eta_P \eta_T \, \varphi^*(-t, -\vec{x}),$$
$$\text{CPT} : \psi(t, \vec{x}) \to -\eta_C \eta_P \eta_T \, \psi^*(-t, -\vec{x}),$$
$$\text{CPT} : \chi(t, \vec{x}) \to \eta_C \eta_P \eta_T \, \chi^*(-t, -\vec{x}),$$
$$\text{CPT} : \Psi(t, \vec{x}) \to -\eta_C \eta_P \eta_T \, \gamma_5 \Psi^*(-t, -\vec{x}),$$
$$\text{CPT} : V^\mu(t, \vec{x}) \to -\eta_C \eta_P \eta_T \, V^{*\mu}(-t, -\vec{x}), \tag{8.10}$$

and so on for higher-spin fields. The CPT theorem states that for any Hermitian local Lorentz-invariant field theory, we can individually pick the phases $\eta_C, \eta_P, \eta_T$ of the fields so that the action of the theory is CPT-invariant.

Note that CPT transformation is anti-unitary, i.e. it flips all $i \to -i$ in the Lagrangian. If the Lagrangian is Hermitian, i.e. $\mathcal{L}^\dagger = \mathcal{L}$, we can undo this flip by a subsequent complex conjugation of the entire Lagrangian. The coordinate flip $x^\mu \to -x^\mu$ in the arguments of the fields can also be undone by a redefinition of the integration variables $x^\mu \to -x^\mu$ in the action $S = \int \mathrm{d}^4 x \, \mathcal{L}(x) \to \int \mathrm{d}^4 x \, \mathcal{L}(-x)$. Consequently, we get the action of the CPT transformation on the Lagrangian as

$$\text{CPT}^* : \varphi(t, \vec{x}) \to \eta_C^* \eta_P^* \eta_T^* \, \varphi(t, \vec{x}),$$
$$\text{CPT}^* : \psi(t, \vec{x}) \to -\eta_C^* \eta_P^* \eta_T^* \, \psi(t, \vec{x}),$$
$$\text{CPT}^* : \chi(t, \vec{x}) \to \eta_C^* \eta_P^* \eta_T^* \, \chi(t, \vec{x}),$$
$$\text{CPT}^* : \Psi(t, \vec{x}) \to -\eta_C^* \eta_P^* \eta_T^* \, \gamma_5 \Psi(t, \vec{x}),$$
$$\text{CPT}^* : V^\mu(t, \vec{x}) \to -\eta_C^* \eta_P^* \eta_T^* \, V^\mu(t, \vec{x}), \tag{8.11}$$

The bi-product of this procedure is that the spacetime derivatives in the Lagrangian flip sign

$$\text{CPT}^* : \partial_\mu \to -\partial_\mu, \tag{8.12}$$

and the order of fields is flipped, which has consequences for anti-commuting spinor fields. Lorentz invariance requires that all Lorentz indices $\mu, \nu, \dots$ appear in contracted pairs in the Lagrangian. Similarly, all the spinors appear as bilinears, such as $\bar{\Psi}\Psi$. Therefore, for invariance under CPT$^*$, and hence under CPT, it is sufficient to require that the phases multiply to unity $\eta_C \eta_P \eta_T = 1$ for all fields.[19] Hence the CPT theorem. Note that this is only a sufficient condition, not necessary. There can indeed be CPT-invariant theories where the phases of fields do not multiply to unity.

---

[19]The argument of spinor bilinears is slightly subtle due to their anti-commuting nature. For instance

$$\bar{\Psi}\Psi = \Psi^\dagger \gamma^0 \Psi \xrightarrow{\text{CPT}^*} \left(\text{CPT}^*(\Psi)^\dagger \gamma^0 \text{CPT}^*(\Psi)\right)^{\mathrm{T}} = \left(\Psi^\dagger \gamma_5 \gamma^0 \gamma_5 \Psi\right)^{\mathrm{T}} = -(\bar{\Psi}\Psi)^{\mathrm{T}} = \bar{\Psi}\Psi. \tag{8.13}$$

Note the change of order of fields implemented by the CPT$^*$ operation via the transpose. In the last step, we have used the anti-commuting nature of $\Psi$ fields.

CPT theorem also implies that the following discrete transformations are equivalent for a Hermitian local Lorentz-invariant field theory: $CP \approx T$, $PT \approx C$, and $CT \approx P$.

**Discrete symmetries of the Standard Model:** The electroweak symmetry in the Standard Model distinguishes between left- and right-handed spinors. It immediately follows that C and P are not symmetries of the Standard Model Lagrangian. The Lagrangian does have an "approximate" CP or T symmetry, provided that we take $\eta_C\eta_P = 1$ and $\eta_T = 1$ for the gauge fields $B_\mu$, $W_\mu$, $G_\mu$ (and consequently also for $W_\mu^{\pm}$, $Z_\mu$, and $A_\mu$). The only violation of the CP symmetry in the Standard Model arises from the flavour-changing weak interactions in the quark sector given in the second-last line of eq. (7.29). To wit, CP symmetry requires that the CKM matrix $V_{mn}$ be real, which is experimentally known not to be the case and was awarded by the 1980 Nobel Prize in physics [20]. Introducing right-handed neutrinos in the Standard Model, similar CP-violation can also occur in the lepton sector via the PMNS matrix. However, the scientific community has still not reached a consensus on whether the flavour-changing weak interactions in the lepton sector violate CP [21]. Finally, due to the CPT theorem, the Standard Model does have CPT symmetry.

# 9 | Outlook

Physicists really dislike freedom when it comes to model building. Good physical models are usually those that survive the Occam's razor, i.e. reliably explain all the known observational evidence with the least number of free parameters. Lesser the number of free parameters a model has, lesser fine-tuning it requires to reconcile with the known observations and more is its predictive power for future experiments. Symmetries serve as a crucial tool to this end. Requiring that a model respects the symmetries of the physical system it aims to describe, or at times postulating entirely new symmetries with the hope that they are realised by the system in question, dramatically brings down the number of free parameters admissible by the model. For instance, the residual U(1) symmetry in the Standard Model does not allow for any interactions in the Lagrangian that do not conserve the total electric charge – a fact that has been experimentally scrutinised without a shadow of doubt.

This was a course on the mathematical foundations of symmetries and their implementation in quantum field theories. We discussed how the symmetries exhibited by a physical system, their action on the underlying field content, and their spontaneous breaking pattern can enable us to construct field theoretic models describing these systems. We started with the foundations of group theory – the mathematical language for symmetries – and provided a detailed analysis of U(1), SU($N$), and Poincaré groups that enter the discussion of particle physics. We discussed various representations of these symmetry groups, providing us with a classification of fields on the basis of their transformation properties under the action of these symmetries. We formalised how these symmetries can be implemented in field theories – globally or locally – and inspected the consequences of the ground state not respecting (a part of) the symmetries of the full theory. We then implemented these ideas to write down the Lagrangian for the Standard Model of particle physics starting from the global Poincaré symmetries, local internal SU(3) × SU(2)$_L$ × U(1)$_Y$ symmetries, their action on the field content: gauge fields, quarks, leptons, and the Higgs field, and the spontaneous breaking of the internal symmetry down to SU(3) × U(1) by the Higgs potential.

It is generally agreed that the Standard Model is not enough. There is the obvious issue of neutrinos being massless in the Standard Model, however this can be fixed relatively easily by introducing right-handed neutrinos. The Standard Model also has a number of inconsistencies with the ΛCDM model of cosmology, including explanations for the amount of cold dark matter and dark energy in the universe, the matter-antimatter inhomogeneity, and the mechanism for cosmic inflation. All of these are still active topics of research in the physics community. Then there is the issue of gravity: the Standard Model of particle physics only describes three out of four fundamental forces, excluding gravity. The reconciliation of the Standard Model with our present understanding of gravity – Einstein's general theory of relativity – has proven to be incredibly challenging and is still an open question in physics.

There are also a number of "naturalness" issues with the Standard Model. The weak force happens to be $10^{24}$ times stronger than the weakest fundamental force in nature – gravity. This is known as the hierarchy problem, for which the Standard Model offers no viable explanation. A possible resolution is offered by "supersymmetry", which extends the spacetime Poincaré symmetries to include fermionic generators. However, to date, there has been no conclusive experimental evidence for supersymmetry. A somewhat different hierarchy problem also arises due to the vastly different masses across the generations of quarks and leptons. Then there is the "strong CP problem" which asks why the weak force violate the CP symmetry but not the strong force. The Standard Model also does not provide any explanation for why the electric charges of fundamental fields are quantised

in multiples of $e/3$. Furthermore, there are a total of 18 free parameters in the Standard Model (6 quark masses, 3 lepton masses, Higgs mass, Higgs vev, 3 gauge coupling constants, and 4 elements of the CKM matrix). In fact, if we were to also include right-handed neutrinos, there would be 7 additional free parameters (3 neutrino masses and 4 elements of the PMNS matrix). Although not a problem as such, it would be preferable if we can reduce this number down, i.e. find physical reasons for why the parameters in the Standard Model are what they are.

As far as reducing the free parameters goes, it is natural to speculate that there is a stronger physical symmetry which further constrains the Standard Model Lagrangian. This lore goes by the name of *grand unified theories*. Another motivation for grand unified theories is that the three gauge couplings in the Standard Model $g_y, g_w, g_s$ seem to converge at around $10^{15}$ GeV under renormalisation group flow. It is speculated that at this scale, the three fundamental forces combine into a single grand unified force described by a simple group with a single coupling constant. The simplest simple group that seems to work at a technical level is SU(5) that admits the Standard Model group $SU(3) \times SU(2)_L \times U(1)_Y$ as a subgroup. As a bonus, the SU(5) grand unified model also explains the quantisation of electric charges. Unfortunately, it has been practically ruled out by the bounds coming from the lifetime of proton decay. There are higher groups that circumvent this problem like SO(10) and $E_6$. However, given that the present energy scale at the LHC is around $10^4$ GeV, testing grand unified theories at $10^{15}$ GeV is still a fantasy. The situation is worse for the "theories of everything" that aim to describe gravity as well, such as string theory, where the unification is only expected to happen at the Planck's scale of around $10^{19}$ GeV.

Our focus has only been on the model building part in this course; we have not really concerned ourselves with any of the features or physical predictions of the Standard Model. Entire textbooks have been written on the subject that go well beyond the scope of this introductory course [8, 18, 19]. Certain advanced symmetry-related concepts have also been left out of the discussion. For instance, we have mostly ignored the quantum aspects of symmetries. In section 6.1.1, we mentioned in passing that the conserved charges furnish a representation of the symmetry algebra on the Hilbert space. However, the quantisation process is considerably more subtle. It is possible for a symmetry that is respected by the classical action of a theory to get violated by quantum fluctuations. Such symmetries are said to be anomalous. Anomalies are particularly problematic when they occur in gauge symmetries. This is because gauge symmetries are really just redundancies in the choice of fundamental fields and can typically be fixed by making a suitable gauge choice. Having an anomalous gauge symmetry, therefore, signals an inconsistency in the choice of fields at the quantum level. As it turns out, the gauge symmetries in the Standard Model are indeed anomaly-free; see [16].

In this course, we have mainly focused on the implementation of symmetries in particle physics. However, the techniques we developed are equally vital in other areas of physics as well. As we indicated in the introduction, phases of condensed matter systems can often be classified based on the symmetries they respect. Fluids (liquids or gases) are phases of matter that are invariant under spatial translations and rotations, while solids have these symmetries broken. There are also a number of phases with an intermediate level of symmetry breaking in between, commonly referred to as liquid crystals. Fluids often also exhibit a global U(1) symmetry associated with the particle number conservation, which can also be spontaneously broken to give rise to superfluids. The ensuing massless Goldstone bosons are responsible for the superfluidity features such as zero-viscosity flows. Similarly solids with a spontaneously broken global U(1) symmetry lead to supersolid phases of matter. Superconductors, on the other hand, have a spontaneously broken local U(1) symmetry. The consequent "massive gauge fields" can explain the physical phenomenon in superconductivity such as

the Meissner effect. In cosmology, the universe at large scales is assumed to be homogeneous and isotropic, which is reflected as translational and rotational invariance in the models of the universe. At smaller scales, this symmetry gets broken by the clusters of galaxies.

Near second-order phase transitions, physical systems can attain scale invariance. In group theoretic terms, this is realised via an enhancement of the spacetime Poincaré symmetries to include scale transformations (dilatations) and often also the abstract special conformal transformation, together known as the conformal group. Field theories that realise the conformal symmetry group are called conformal field theories. Conformal symmetry is often strong enough to completely fix all the two- and three-point correlation functions in a field theory and even impose strong constraints on the higher-point interactions; see [22]. In string theory, the worldsheet of the fundamental string has conformal invariance. Conformal field theories also show up in the context of the AdS/CFT correspondence: it is believed that certain conformal field theories are dual to certain quantum gravitational theories living in one-higher dimension. For most physical systems, conformal symmetry is too restrictive, but it does serve as a spherical cow approximation to gain qualitative insights into otherwise highly intractable physical systems.

Another spacetime symmetry that we have not had the scope to discuss are diffeomorphisms. We discussed how global internal symmetries can be made into local spacetime-dependent symmetries by introducing gauge fields. A spiritually similar procedure can be implemented for spacetime Poincaré symmetries as well. To wit, invariance of a field theory under spacetime Poincaré transformations $x^\mu \to \Lambda^\mu_{\ \nu} x^\nu + a^\mu$ can be promoted to arbitrary diffeomorphisms $x^\mu \to \Lambda^\mu_{\ \nu}(x) x^\nu + a^\mu(x) = x'^\mu(x)$, by introducing a spacetime metric $g_{\mu\nu}(x)$ and an affine connection $\Gamma^\lambda_{\mu\nu}(x)$. Invariance of physics under arbitrary diffeomorphisms is the defining feature of Einstein's general theory of relativity that describes classical gravity. In principle, this procedure can also be implemented to make the Standard Model Lagrangian invariant under diffeomorphisms and obtain a model describing all four fundamental forces. However, one runs into technical issues with renormalisability while trying to quantise the theory.

This is where we draw this course to a close. Our journey has undoubtedly been high on the technical side. We covered a lot of ground, starting from the mathematical definition of a group all the way to the Standard Model Lagrangian. I would not blame the reader if they failed to recall most of the intricacies of the discussion in a span of a few months. However, if there is one key message from the course that should stay with the reader, it would be that symmetries are powerful. In the words of the German mathematician and theoretical physicist Hermann Weyl [23], "symmetry, as wide or as narrow as you may define its meaning, is one idea by which man through the ages has tried to comprehend and create order, beauty and perfection."

## Acknowledgements

I am immensely thankful to Adam Ritz and Pavel Kovtun for their guidance and aid throughout the preparation of these notes and the teaching of this course. I would like to extend my sincerest gratitude to Adam Ritz for sharing his handwritten notes on the subject that served as the backbone for this course to expand and elaborate upon. I am also thankful to the University of Victoria for allowing me the opportunity to teach this course and, of course, the graduate students who suffered through it.

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
