# Peer review of "Notes on symmetries in particle physics"

_SciPost Physics Lecture Notes_

## Round 1 · Referee Report · Ben Gripaios (Referee 2) · 2022-1-24

Report
On p. 8, a better definition of a group would perhaps be to replace
"A group $G$ is a set of elements with a product rule, such that \dots "
with
"A group $G$ is a set of elements with a product rule, namely a function $G \times G \rightarrow G$, such that \dots "
This removes any ambiguity over what "a product rule" is, along with the need for the first axiom of closure. (It also has the advantage that it mimics precisely the definition of a Lie algebra that follows later.)
The author might also like to remark there that the given axioms are redundant, in that it's enough to assume the existence of a left identity $e$ (not necessarily unique) and existence (again not necessarily unique) of a left inverse $g^{-1}$ for that left identity.
The author doesn't appear to have given a definition of a homomorphism at any point. It is hard to imagine doing anything in group theory without this notion. I would suggest doing so just before the definition of isomorphic groups on p. 9. So a homomorphism is a map between groups that preserves products (which, one might remark, implies it preserves the identity and inverses as well) and an isomorphism is a pair of such maps that are mutual inverses of one another. One can then define two groups to be isomorphic if an isomorphism exists.
On p. 10 the author writes "NB: continuous symmetry transformations, such as rotations of a circle, have order infinite
and are known as infinite dimensional groups." The word "dimensional" is incorrect and should be deleted here and elsewhere (in particular, the given example of rotations of a circle is an infinite group, but it has dimension one, as a Lie group).
On p. 10, the definition of a representation can be shortened to: "a homomorphism from G to GL(N)" if a homomorphism is defined as suggested above. Moreover, we should be told that GL(N) is the group of invertible matrices with elements in some field (assumed, I presume, arbitrary, since reference is later made to both real and complex representations). I suggest using an arbitrary field F, since it will help later with some of the confusion that arises regarding Lie algebras.
This is followed on p.11 with a definition of a unitary representation, but this only makes sense for complex representations, which should thus be specified.
On p. 11, the definition of a faithful representation is incorrect. Such a representation is one-to-one, but not necessarily onto, so not necessarily "invertible" as claimed.
On p. 11, the trivial representation is defined as one for which every group element is sent to the $N\times N$ identity matrix. In fact this is the direct sum of N copies of the trivial representation as usually defined, which sends every group element to the $1\times 1$ identity matrix.
On p. 13, in eq (2.15) a definition is given of the structure constants of a Lie algebra over an arbitrary field F. Unfortunately, neither "$i$" (which is an element of $\mathbb{C}$) nor "$\hbar$" (which is an element of $\mathbb{R}$) have meaning in that context.
Eq. (2.16) that follows is problematic, because neither the product $T_a T_b$ nor its trace have been defined (as things stand, $T_a$ is a vector in some vector space over some field $F$.) Even if they were defined, "$C$" would not be some "real number", as claimed.
I suggest dealing with all this by restricting first (as in the examples) to matrix algebras with complex entries, regarded as Lie algebras over the reals, with Lie bracket given by the commutator of matrix products.
One might also explain, with regard to footnote 1, that the reason for the appearance of the factor $i$ in the physicist's conventions is to make the matrices $T_a$ hermitian, rather than anti-hermitian.
On p. 15, it would again surely be better to define a homomorphism between two Lie algebras (both over the same field $F$) and then an isomorphism as a special case. A virtue of doing so is that one can then define a representation as a homomorphism of Lie algebras $\mathfrak{g} \rightarrow \mathfrak{gl} (N)$ (over some common $F$). Whilst slick, this definition also makes it clear that one needs to be careful about what field one is working over (cf. the examples mentioned above, which are matrix algebras with complex entries being regarded as Lie algebras over the reals). This sort of confusion as to complex vs. real Lie algebras and their representations is commonplace in the physics literature, and this seems to be a good opportunity for the author to sort it out, or at least make clear that this is where it originates!
On p. 17, a better definition of a Lie group would be "A Lie group $G$ is a manifold endowed with the structure of a group, such that the maps $G \times G \rightarrow G : (g,g^\prime) \mapsto gg^\prime$ and $G\rightarrow G : g \mapsto g^{-1}$ (corresponding to multiplication and inversion) are both smooth. (Note in particular that a Lie group is not "infinite-dimensional" as claimed, and nor is it even infinite, in general (every finite group defines a Lie group, for example).
On p. 17, in the definition of a connected Lie group, I'd replace "i.e. there exists a parametrisation \dots" with "i.e. for every $g \in G$, there exists a path $g(\alpha) \in G$ such that $g(0)=e$ and $g(1)=g$.
Immediately following, I'd replace "connected part" with "component connected to the identity", since this are standard. One could also use simply "identity component".
Strengths
Weaknesses
Report
Requested changes
See report. I attached the report in the Section remarks for the Editors only, but in fact it is the report to the author. Since it is a long report of 11 pages I prefer to send a pdf file.

---

## Editorial Decision

unknown